# Innate immune responses against the fungal pathogen *Candida auris*

Yuanyuan Wang[1,2,3,17], Yun Zou[1,2,3,17], Xiaoqing Chen[1,2], Hao Li[4], Zhe Yin[5], Baocai Zhang[6], Yongbin Xu[7], Yiquan Zhang[8], Rulin Zhang[9], Xinhua Huang[1], Wenhui Yang[5], Chaoyue Xu[1,3,10], Tong Jiang[1,2], Qinyu Tang[11], Zili Zhou[1], Ying Ji[1,2], Yingqi Liu[12], Lingfei Hu[5], Jia Zhou[13], Yao Zhou[14], Jingjun Zhao[11], Ningning Liu[13], Guanghua Huang[15], Haishuang Chang[16], Wenxia Fang[14], Changbin Chen[1,3✉] & Dongsheng Zhou[5✉]

*Candida auris* is a multidrug-resistant human fungal pathogen responsible for nosocomial outbreaks worldwide. Although considerable progress has increased our understanding of the biological and clinical aspects of *C. auris*, its interaction with the host immune system is only now beginning to be investigated in-depth. Here, we compare the innate immune responses induced by *C. auris* BJCA001 and *Candida albicans* SC5314 in vitro and in vivo. Our results indicate that *C. auris* BJCA001 appears to be less immunoinflammatory than *C. albicans* SC5314, and this differential response correlates with structural features of the cell wall.

[1] The Center for Microbes, Development and Health, CAS Key Laboratory of Molecular Virology and Immunology, Institut Pasteur of Shanghai, Chinese Academy of Sciences, Shanghai 200031, China. [2] University of Chinese Academy of Sciences, Beijing, China. [3] Nanjing Advanced Academy of Life and Health, Nanjing 211135, China. [4] Department of General Surgery, Shanghai General Hospital, Shanghai Jiao Tong University School of Medicine, Shanghai 200080, China. [5] State Key Laboratory of Pathogen and Biosecurity, Beijing Institute of Microbiology and Epidemiology, Beijing 100071, China. [6] State Key Laboratory of Plant Genomics, Institute of Genetics and Developmental Biology, Chinese Academy of Sciences, Beijing 100101, China. [7] Institute of Chinese Materia Medica, Shanghai University of Traditional Chinese Medicine, Shanghai 201203, China. [8] Wuxi School of Medicine, Jiangnan University, Wuxi 214122 Jiangsu, China. [9] Department of Laboratory Medicine, Shanghai General Hospital, Shanghai Jiao Tong University School of Medicine, Shanghai 20008, China. [10] College of Life Science, Shanghai University, Shanghai, China. [11] Department of Dermatology, Tongji Hospital, Tongji University School of Medicine, Shanghai 200065, China. [12] School of Basic Medicine, Gannan Medical University, Ganzhou, China. [13] Center for Single-Cell Omics, School of Public Health, Shanghai Jiaotong University School of Medicine, Shanghai, China. [14] National Engineering Research Center for Non-Food Biorefinery, Guangxi Academy of Sciences, Nanning 530007 Guangxi, China. [15] Department of Infectious Disease, Huashan Hospital and State Key Laboratory of Genetic Engineering, School of Life Sciences, Fudan University, Shanghai, China. [16] Shanghai Institute of Precision Medicine, Shanghai Ninth People's Hospital, Shanghai Jiaotong University School of Medicine, Shanghai, China. [17] These authors contributed equally: Yuanyuan Wang, Yun Zou. ✉email: cbchen@ips.ac.cn; zhouds@bmi.ac.cn

The frequency of invasive disease due to opportunistic fungal pathogens has increased significantly during the last two decades. Although most of time fungal species cause rare invasive, life-threatening diseases, the incidence of human fungal infections is increasing at an alarming rate, bringing an enormous challenge in human beings, due to the growing population of immunocompromised individuals[1,2]. A good example is the recent nosocomial outbreak of *Candida auris*, a fungal pathogen that has recently emerged as a global threat to public health. This yeast-like fungus was first discovered in Japan in 2009 after it was isolated from the external ear canal of a patient[3]. Since then, it has been spreading rapidly throughout the world with hundreds of thousands of individual cases or outbreaks[4]. One of the most important features about *C. auris* is that 90% of clinically isolated strains exhibit resistance to fluconazole, and some strains are resistant to all currently available antifungals, causing a staggering mortality rate of up to 60%[5]. *C. auris* has been recognized as an urgent threat to public health due to its drug-resistance characteristics and association with long-term colonization and extensive environmental contamination (https://www.cdc.gov/drugresistance/biggest-threats.html)[6].

The pathogenicity of *C. auris* has been investigated using a variety of animal models. A previous study in an invertebrate *Galleria mellonella* infection model found that the nonaggregating forms of isolates have the capacity of forming robust biofilms[7] and exhibit comparable virulence to *C. albicans* whereas the aggregate-forming isolates display greatly reduced virulence, suggesting that the virulence attributes of *C. auris* appear to be strain-dependent[8]. This notion was further verified by testing the pathogenicity of *C. auris* isolates from geographically and phylogenetically distinct clades. For example, a recent systematic study using 17 clinical isolates of the four prevalent clades (South Asian, East Asian, South African, and South American) found that isolates of the same clade did show pathogenic differences in a neutropenic murine bloodstream infection model and *C. auris*, regardless of clade, was less virulent than *C. albicans*[9]. Although the reasons for these differences are not fully understood, it is possible that *C. auris* may survive in the host by withstanding the host immune responses, in addition to its capacity of driving genetic variation within strains. Indeed, Ben-Ami et al. tested the virulence of *C. auris* isolates and found that yeast cells recovered from kidneys of immunosuppressed BALB/c mice form aggregates, implying a possible immune evasion mode present in this specific form[10]. Moreover, immunocompetent C57BL/6 mice also exhibited resistance to the infection of *C. auris* but not *C. albicans*[11]. Interestingly, Johnson et al.[12] found that neutrophils exert a strong antifungal response to *C. albicans* whereas they fail to engage and phagocytose yeast cells and form extracellular traps when interacting with *C. auris*. The results were further recapitulated in a zebrafish model of invasive candidiasis; compared to *C. albicans*, ~50% less neutrophils were recruited in response to *C. auris* infection. A growth advantage of *C. auris* in immunocompetent animal models suggests that *C. auris* may escape from innate immune sensing[13]. However, Bruno et al. recently reported that *C. auris* triggers a stronger innate host defense response compared with *C. albicans*[14]. In summary, previous studies showed differing results on the ability of *C. auris* to induce innate immune responses.

In 2018, Wang et al. reported the identification of the first clinical isolate of *C. auris* in China (BJCA001), from the bronchoalveolar lavage fluid of a female patient[15]. Unlike most drug-resistant strains isolated in other countries, this isolate was found to display susceptibility to antifungals like fluconazole during early incubation, however, rapidly established acquired resistance in a medium containing increased concentrations of fluconazole[16]. Recent work further identified that *C. auris*

BJCA001 was able to undergo a tri-stable phenotypic switch between typical yeast, filamentation-competent (FC) yeast and filamentous cells, a morphological change similar to the classical yeast-to-hyphae switch identified in most *Candida* species like *C. albicans*[17].

Here, we use *C. auris* BJCA001 and other clinical isolates from different geographic regions, to provide in vitro and in vivo evidence that *C. auris* isolates tend to induce a less potent proinflammatory innate immune response than *C. albicans*. Furthermore, their different immunostimulatory capacities correlate with structural differences in the cell wall.

## Results

**Persistent high fungal loads in immunocompetent mice following intravenous infection of *C. auris*.** Accumulating evidence suggests that the increasing global prevalence of *C. auris* infection could be influenced by host immune status, since clinical strains are most frequently isolated from immunosuppressed patients, particularly the ones in the ICU[18,19]. Moreover, studies in experimental mouse models showed that the immunologically competent host exhibits marked resistance to systemic *C. auris* infection[11,15,20], raising a possibility that infected *C. auris* cells might be efficiently eliminated or neutralized by host innate immune defenses. To test this, we established a mouse model of disseminated *C. auris* infection by which groups of immunocompetent, female 6–8-week-old C57BL/6 mice were intravenously inoculated with two different dosages of *C. auris* BJCA001 yeast cells ($2 \times 10^7$ and $1 \times 10^6$ CFUs, respectively), and fungal burdens in kidney, spleen and brain were measured daily by CFU determination. As shown in Supplementary Fig. 1a, b, we unexpectedly found that regardless of a high or low inocula, fungal loads in kidney, spleen and brain were maintained at a constantly high level over the first 7-day experimental period, which normally represents a time window for induction of innate immunity[21]. Notably, infection with pathogenic *C. albicans* SC5314 yeast cells ($5 \times 10^4$ CFUs) continuously yielded high tissue fungal burdens during the course of experiments (Supplementary Fig. 1c), evidence on disease causality. Moreover, histopathologic examination of Periodic acid-Schiff (PAS) and hematoxylin and eosin (H&E)-stained sections from *C. auris*-infected mice confirmed abundant tissue colonization of aggregate yeast cells with no pseudohyphae and filaments, however, showed less severe inflammation and tissue damage and had much smaller multifocal areas of abscess formation than did *C. albicans*-infected mice (Supplementary Fig. 1d–f). Taken together, these results highly suggest that after infection, *C. auris* cells were able to persist in the host and avoid to be recognized and cleared by the host innate immune system.

**In vitro cell culture studies of host innate immune responses against *C. auris*.** The observation that high levels of *C. auris* cells were constantly detected in host organs albeit innate immune activation prompted us to hypothesize that this fungus may evolve an uncharacterized immune evasion strategy to combat the innate immune killing. To test this hypothesis, we first carried out in vitro cell culture studies to examine whether or not *C. auris* cells could evade host antifungal innate immunity, in comparison with the commonly encountered human fungal pathogen *C. albicans*.

The innate immune system represents the first line of defense against systemic fungal infections and innate immune phagocytes, including macrophages and neutrophils, act like a primary infection determinant[22,23]. In an effort to describe the potential ability of *C. auris* to evade host innate immune response, we examined the expression patterns of a list of pro-inflammatory

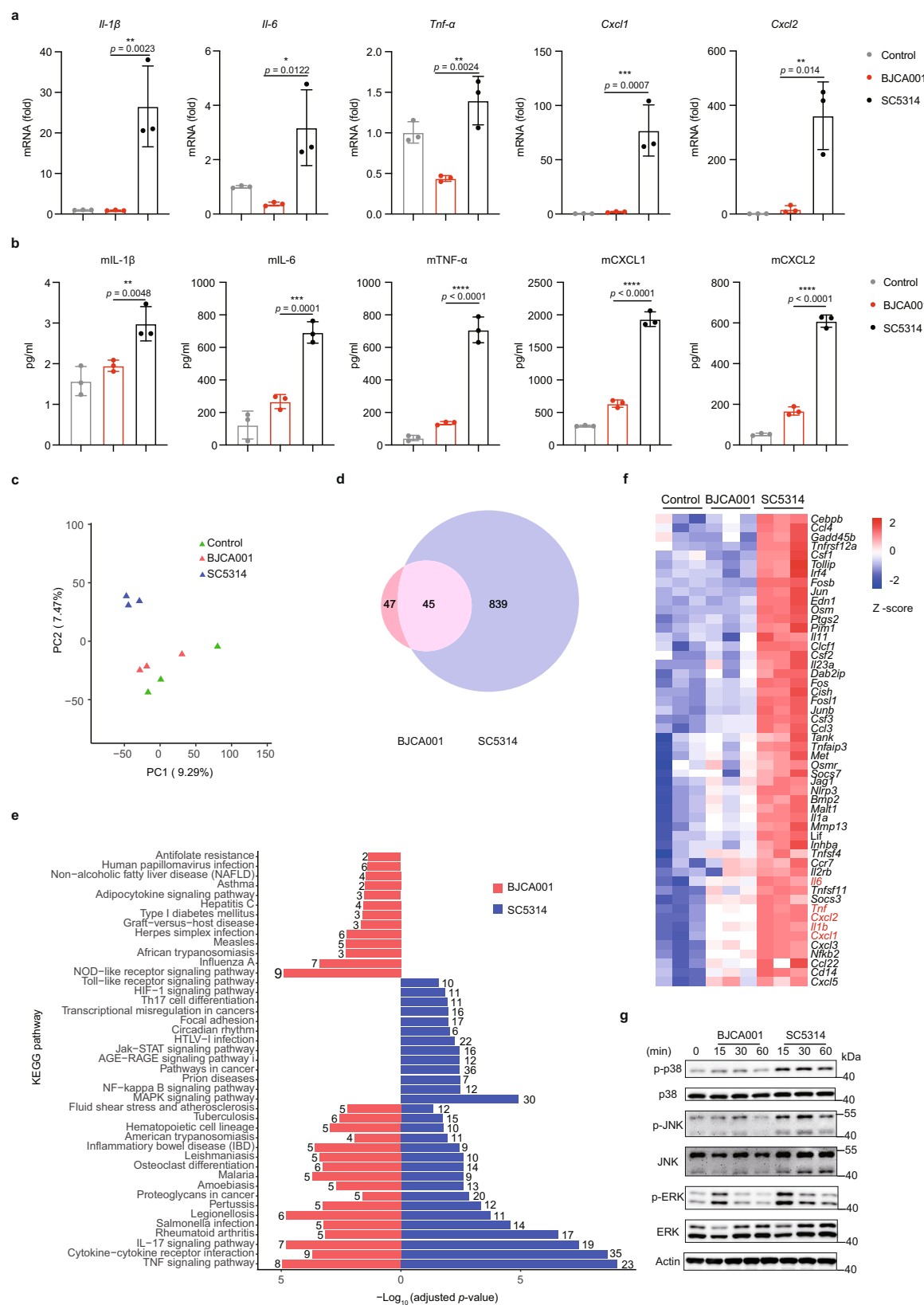

cytokines and chemokines in murine bone marrow-derived macrophages (BMDMs) treated with live *C. auris* BJCA001 or *C. albicans* SC5314 (MOI = 5) for 3 or 6 h. Longer incubation periods (*e.g.*, 24 h) were not considered in this study, since we observed robust cell death in BMDMs 24 h after co-incubation (Supplementary Fig. 2). As shown in Fig. 1a, b, we observed a

strong correlation between the results obtained by RT-qPCR and ELISA. Upon *C. albicans* stimulation, the mRNA levels of proinflammatory cytokines and chemokines, including IL-1β, IL-6, TNF-α, CXCL1, and CXCL2, significantly increased in parallel with the secretion of these factors, however, the induction was dramatically diminished by *C. auris* infection. When yeast-form

**Fig. 1 In vitro studies showing that *C. auris* BJCA001 induces a less potent innate immune response than *C. albicans* SC5314. a** Expression levels of IL-1β, IL-6, TNF-α, CXCL1, and CXCL2, as determined by real-time RT-qPCR, in BMDMs that were infected without (PBS; negative control) or with live *C. auris* or *C. albicans* yeast cells (MOI = 5) for 3 h (*n* = 3 biologically independent samples). Results were normalized to the expression of the control gene GAPDH and are presented relative to those of negative control, set as 1. **b** Production of IL-1β, IL-6, TNF-α, CXCL1, and CXCL2, as determined by ELISA, in the culture supernatants of BMDM that were infected without or with live *C. auris* or *C. albicans* yeast cells (MOI = 5) for 6 h (*n* = 3). **c** Principal component analysis (PCA) of the normalized RNA-seq data of BMDMs in response to challenge with PBS (control), live *C. auris* or *C. albicans* yeast cells (MOI = 5) for 6 h. **d** Venn diagram showing differentially expressed genes (DEGs; fold change ≥2, adjusted *p* ≤ 0.05). The number in each circle represents the total number of genes that are differentially expressed in each sample, and the overlapping part of circles indicates that the gene is down- or upregulated in both samples. **e** A schematic diagram of a bar chart for the top 30 enriched KEGG pathways ranked according to their values of −log$_{10}$ (adjusted *p* value). The number of DEGs in a given pathway is listed. **f** Heatmap of changes in gene expression of selected innate immune-related genes. **g** Immunoblot analysis for the indicated MAPKs using lysates from BMDMs that were infected with live *C. auris* or *C. albicans* yeast cells (MOI = 5) for the indicated amounts of time. Data are representative of two independent experiments. Data are expressed as mean ± SD and are representative of three independent experiments. *$p < 0.05$, **$p < 0.01$, ***$p < 0.001$, ****$p < 0.0001$, by one-way ANOVA with Sidak's test (**a**, **b**). Source data are provided as a Source Data file.

cells of *C. albicans* or *C. auris* were inactivated by 4% paraformaldehyde (PFA-killed) and treated with BMDMs, we also found that *C. albicans* was able to stimulate higher levels of cytokine/chemokine gene expression (Supplementary Fig. 3), minimizing a possible effect of hyphal induction during co-culture.

The results were further validated by a genome-wide RNA-seq analysis. BMDMs were challenged with PBS, live *C. auris,* or *C. albicans* yeast cells (MOI = 5) for 3 or 6 h and harvested for RNA-seq. A previous study has shown that during the interaction between BMDMs and *C. albicans*, the stimulus of ~1–3 h post infection represents the early phagocytosis events, and the ~6 h of treatment terms the initial fungal escape following caspase-1-dependent pyroptosis[24]. Principal component analysis (PCA) performed on accurately normalized data showed that mapped RNA-seq reads of both PBS and *C. auris* were clustered together, with minor variations between the biological replicates, whereas reads of *C. albicans* exhibited major variance, as a clear separation from PBS and *C. auris* was observed (Fig. 1c, Supplementary Fig. 4a), suggesting that the host response of *C. auris* was very similar to that of PBS but significantly separated from that of *C. albicans*. Indeed, when BMDMs were challenged with live *C. auris*, only 73 and 92 differentially expressed genes (DEGs; fold change ≥2, *p* adjusted ≤0.05) were observed at the 3 and 6 h time points, respectively, relative to PBS control. By contrast, treating BMDMs with *C. albicans* significantly increased the number of DEGs, with 201 and 884 genes at the 3 and 6 h time points, respectively (Fig. 1d, Supplementary Fig. 4b). In particular, we found that after 6 h of stimulation, a total of 884 DEGs were identified in *C. albicans*-infected BMDMs, including 45 being in common between *C. albicans* and *C. auris* (Fig. 1d). The entire list of DEGs is shown in Supplementary Data 1 and 2. KEGG pathway enrichment analysis of DEGs (Supplementary Data 3 and 4) showed that the host response specific for *C. albicans* stimulation includes genes related to chemokine, Th17, Toll-like receptor, JAK-STAT, NF-κB, and MAPK signaling pathways, whereas it is not the case for the response to *C. auris* (Fig. 1e, Supplementary Fig. 4c). For DEGs whose functions were linked to cytokine-cytokine receptor interaction, IL-17, and TNF-α signaling pathways, we observed a broader transcriptional induction, with 21 and 46 upregulated genes after *C. albicans* stimulation for 3 and 6 h, respectively. In comparison, only 8 and 10 genes in these three pathways were found to be upregulated after *C. auris* challenge. This tendency became even more obvious when we compared changes in the expression of selected innate immune-related genes during the interaction of BMDMs with PBS, *C. auris,* or *C. albicans* (Fig. 1f, Supplementary Fig. 4d). Of significance, expression of these selected genes was highly induced after challenge with *C. albicans* but not *C. auris*. Interestingly, the expression patterns of four pro-inflammatory

genes IL-1β, IL-6, TNF-α, CXCL1, and CXCL2 (marked in red) exactly recapitulated the results in Fig. 1a.

Numerous studies have demonstrated that the innate immune system acts to sense fungal pathogen mainly through Syk-coupled C-type lectin receptors and activate downstream signaling regulators by phosphorylation, including the classical mitogen-activated protein kinases (MAPKs)[25–27], we, therefore, evaluated the levels of active (phosphorylated) forms of ERK, JNK and p38, as well as the total protein levels, in BMDMs treated with live *C. auris* or *C. albicans*. As illustrated in Fig. 1g, treating BMDMs with *C. albicans* yeasts results in augmented phosphorylation of ERK, JNK, and p38 MAPK in BMDMs without significantly affecting the total level of each protein, however, phosphorylation of these signaling factors was unaltered in *C. auris*-treated macrophages, indicating MAPK suppression. The result was in line with our RNA-seq data showing that the expression of host genes belonging to MAPK signaling pathway failed to be induced by *C. auris* (Fig. 1e). Interestingly, we observed peak levels of phosphorylated ERK, JNK, and p38 MAPKs in BMDMs after co-culture with live *C. albicans* cells for 15 min (Fig. 1g), and at this time, the cells were still uniformly yeast-form (Supplementary Fig. 4e), downplaying the influence of hyphal formation on host MAPK activation. Collectively, our data suggest that *C. auris* is, in comparison with *C. albicans*, a less potent inducer of MAPK signaling pathway controlling the expression of proinflammatory cytokines and chemokines in macrophages.

Host-pathogen interaction is crucial for the pathogenesis of invasive fungal infections. The fungus has to first adhere to the host epithelial surface and then cross the epithelium in order to establish a successful infection[28], and phagocytic cells such as macrophages, are being increasingly appreciated as having important roles in recognizing and eliminating the microbial invaders[22]. To further test our proposition that *C. auris* may induce a less potent innate immune response than *C. albicans*, we implemented different approaches to verify if *C. auris* is weakly recognized by immune cells. First, we compared adherence of both *C. auris* and *C. albicans* to different human epithelia. Using an established in vitro adhesion assay, we found in Fig. 2a–e that in contrast to a remarkable capacity of *C. albicans* to adhere to multiple human cell lines, including the human skin keratinocyte cell line (HaCat), colorectal adenocarcinoma cell line (Caco-2), umbilical vein endothelial cell line (HUVEC), adenocarcinoma cancer cell line (A549) and HeLa cells, there is a striking reduction of adhesion activity in *C. auris*. Interestingly, no statistically significant difference was observed in *C. auris* adhesion between skin-derived HaCat and gut-derived Caco-2 (Supplementary Fig. 5a), and an in vitro *Candida* infection model using the commercial three-dimensional reconstructed epidermis-EpiSkin[29] also verified the weak adherence ability of *C. auris* (Supplementary Fig. 5b, c). However, in vivo model of

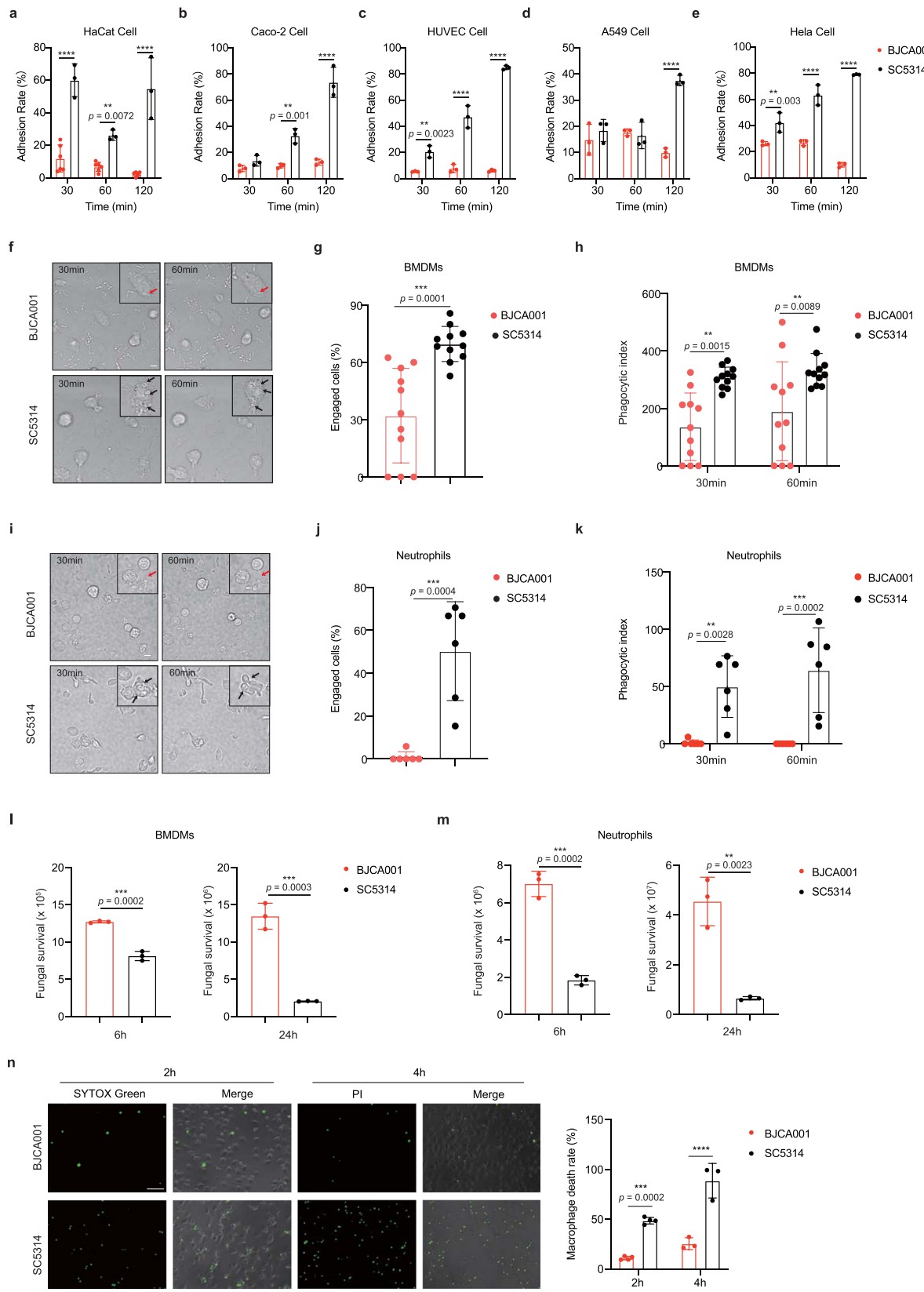

murine skin, topical exposure[30] revealed that *C. auris* established persistent colonization on the skin surface (Supplementary Fig. 5d, e). The exact reason for this difference (in vitro vs in vivo) is unclear at present, possibly related to structural variations between monoculture and actual skin, as well as differential growth responses of *C. auris* under different conditions. Moreover, we found that unlike *C. albicans*, *C. auris* failed to be endocytosed by Caco-2 cells (Supplementary Fig. 5f, g, Supplementary Movies 1, 2) and some other epithelial or endothelial cells (Supplementary Movies 3–10). Second, we investigated host-pathogen interactions by examining the abilities of different immune cells, including macrophages and neutrophils

**Fig. 2 *C. auris* BJCA001 is weakly recognized by the host cells, compared with *C. albicans* SC5314.** **a–e** Time course of adherence to the human skin keratinocyte cell line (HaCat) (**a**), human colorectal adenocarcinoma cell line (Caco-2) (**b**), human umbilical vein endothelial cell line (HUVEC) (**c**), human adenocarcinoma cancer cell line (A549) (**d**) and human HeLa cells (**e**) by *C. auris* BJCA001 and *C. albicans* SC5314. Adherence of *Candida* species to epithelial or endothelial cells was determined by counting the adhering CFU of the yeast and expressed as the percentage of the original inoculum (*n* = 3). **f** Representative snapshots were taken from live-cell videos after 30 and 60 mins co-incubation of murine BMDMs with live *C. albicans* SC5314 and *C. auris* BJCA001 cells (MOI = 1). The numbers in the upper left corner of each image represent the time of the phagocytic events, arrows indicate the fungal cells engulfed by BMDMs (scare bar = 10 μm). **g, h** Percentage uptake and phagocytic index for BMDMs ingesting *C. albicans* and *C. auris* (*n* = 11). Macrophages that have taken up at least one fungal cells were manually tracked to allow a quantitative analysis of percentage uptake during the 60 min co-incubation (**g**). The number of fungal cells ingested (phagocytic index) per 100 macrophages was manually counted during the 30 and 60 min co-incubation (**h**). **i–k** Live-cell imaging data from co-incubations with murine neutrophils and *C. albicans* or *C. auris* (MOI = 1). Data were analyzed using exactly the same methods described in **f–h** (*n* = 6). **l, m** The survival of *C. albicans* and *C. auris* after infection of murine BMDMs (**l**) or neutrophils (**m**) at a MOI of 1 for 6 and 24 h was evaluated by plating dilutions of sample on YPD agar (*n* = 3). **n** *C. auris* or *C. albicans* cells were co-cultured with murine BMDMs at MOI = 5 for 2 h and 4 h, and samples were stained by membrane-impermeable fluorescent dyes such as SYTOX green (2 h; *n* = 4) or PI (4 h; *n* = 3) and the nonviable cells (green) were examined by fluorescent microscope (Olympus IX73). Acquired images were automatically analyzed by ImageJ image analysis software (NIH, Bethesda, MD, USA). The dead BMDM cells were counted by the count function of ImageJ. Scare bar = 10 μm. Each data were analyzed by 10 different images. Data are representative of at least three independent experiments. Data are expressed as mean ± SD and are representative of three independent experiments. **$p < 0.01$; ***$p < 0.001$; ****$p < 0.0001$; by two-way ANOVA with Sidak's test (**a–e, h, k, n**) or two-side unpaired *t* test (**g, j, l, m**). Source data are provided as a Source Data file.

from humans and mice, to phagocytose and kill *C. auris* yeast cells in vitro. *C. albicans* or *C. auris* cells were co-cultured over time with mouse BMDMs and primary human monocyte-derived macrophages (MDMs), respectively, and initial recognition by macrophages was investigated by live video microscopy (Supplementary Movies 11–14). Results were presented as percentage uptake (the percentage of macrophages that engulfed at least one fungal cell) and phagocytic index (the number of fungal cells taken up per 100 macrophage). In both human and mouse macrophages, *C. auris* showed reduced rates of uptake compared to *C. albicans* (Fig. 2f, g, Supplementary Fig. 6a, b) and the phagocytic index of *C. auris* was also less than that of *C. albicans* after 30 and 60 min of co-culture (Fig. 2f, h, Supplementary Fig. 6a, c). Similar results were observed for PFA-killed yeast-form cells of both strains (Supplementary Fig. 6d–f). Furthermore, we investigated the interaction of murine and human neutrophils with *C. albicans* or *C. auris* and also found differences in the host-pathogen interaction (Supplementary Movies 15–18). Compared to *C. albicans*, *C. auris* appeared to be taken up and internalized by neutrophils at lower rates (Fig. 2i–k, Supplementary Fig. 6g–i). Importantly, the weaker interaction between *C. auris* and innate immune cells correlated with stronger resistance to intracellular killing by phagocytic cells (Fig. 2l, m, Supplementary Fig. 6j, k). These results indicate that *C. auris* may avoid being engaged and phagocytosed during the course of infection and host defense. Indeed, further analysis using fluorescent microscopy proved that compared to *C. albicans*, *C. auris* caused a small, but significant fall in the percentage of uptake (Supplementary Fig. 6l) and phagocytic index (Supplementary Fig. 6m), as well as less damage to the phagocytes like BMDMs (Fig. 2n).

Taken together, our cell culture-based in vitro experiments strongly suggest that unlike *C. albicans*, *C. auris* appears to be weakly recognized and eliminated by host innate immune system.

**In vivo studies investigating innate immune responses to *C. auris*.** We further evaluated innate immune responses using the mouse model described above (Supplementary Fig. 1). We intravenously infected groups of C57BL/6 mice with PBS, *C. auris* (1 × 10⁶ CFU/mice; L), *C. auris* (2 × 10⁷ CFU/mice; H), or *C. albicans* (5 × 10⁴ CFU/mice) and then assessed fungal burdens in various organs including kidney, liver, spleen, lung and brain, through CFU counting. It has to be noted that different inoculum doses of *C. albicans* (5 × 10⁴ CFU/mice) and *C. auris* (1 × 10⁶ CFU/mice) were used, largely because of the fact that mice

inoculated with *C. albicans* at $1 × 10^6$ CFU experience a clinical illness manifested by ruffled fur, reduced activity and weight loss, and rapidly reach the humane endpoints (within 5 days) (Supplementary Fig. 7a). In comparison, those receiving a lower inoculum ($5 × 10^4$ CFU/mice) could live long and healthy during the period of innate immune response (usually within the first 7 days). It has to be noted that all mice infected with *C. auris* cells survived and appeared to be clinically normal, although we did observe abnormal symptoms, including slight weight loss, head bobbing and body spinning, in mice receiving a high inoculum ($2 × 10^7$ CFU/mice) (Supplementary Fig. 7b, c). Consistent with the results in Supplementary Fig. 1, we observed in Fig. 3a and Supplementary Fig. 8a that following *C. auris* inoculation, the fungal burden was fairly constant and high on day 2 and 5 post-inoculations, regardless of inoculum size. However, a relative reduction in the fungal burden on day 2 and a comparably high level as that of *C. auris* on day 5 were observed after *C. albicans* infection, possibly due to the early activation of innate immune eradication. No significant difference in fungal burden was observed between the low or high inoculum of *C. auris*. These results suggest that *C. auris* may act differently from *C. albicans* and elicit less potent innate immune responses.

To document the nature of innate immune response found in innate immune cells during *C. auris* or *C. albicans* infection, we used the same animals as treated in Fig. 3a and performed flow cytometry to assess the recruitment of neutrophils to infected organs such as spleen and kidney. In line with previous reports[31,32], we also found that stimulation with *C. albicans* leads to a sizeable increase in the number of neutrophils on both day 2 and 5 post-inoculation, when compared to baseline levels. However, mice challenged with *C. auris* ($1 × 10^6$ CFU/mice; L) showed significantly reduced neutrophil recruitment, with similar sizes of the neutrophil population in an uninfected condition (Fig. 3b). More importantly, higher innocula ($2 × 10^7$ CFU/mice; H) only slightly increased the number of neutrophils (Fig. 3c), ruling out the possibility that inability of *C. auris* to induce neutrophil recruitment might be due to a low-density inoculation. Therefore, our data strongly suggested that *C. auris* infection resulted in less neutrophil egress than did *C. albicans*.

The relative abundance of an array of immune cell populations (e.g., leukocytes, CD11b⁺ cells, macrophages, monocytes, NK cells, B cells, T cells) was also quantified by flow cytometry. Consistently, we found that the distribution of innate immune cells (e.g., leukocytes, CD11b⁺ cells, macrophages, monocytes, and NK cells) mirrored the patterns described for neutrophils, is

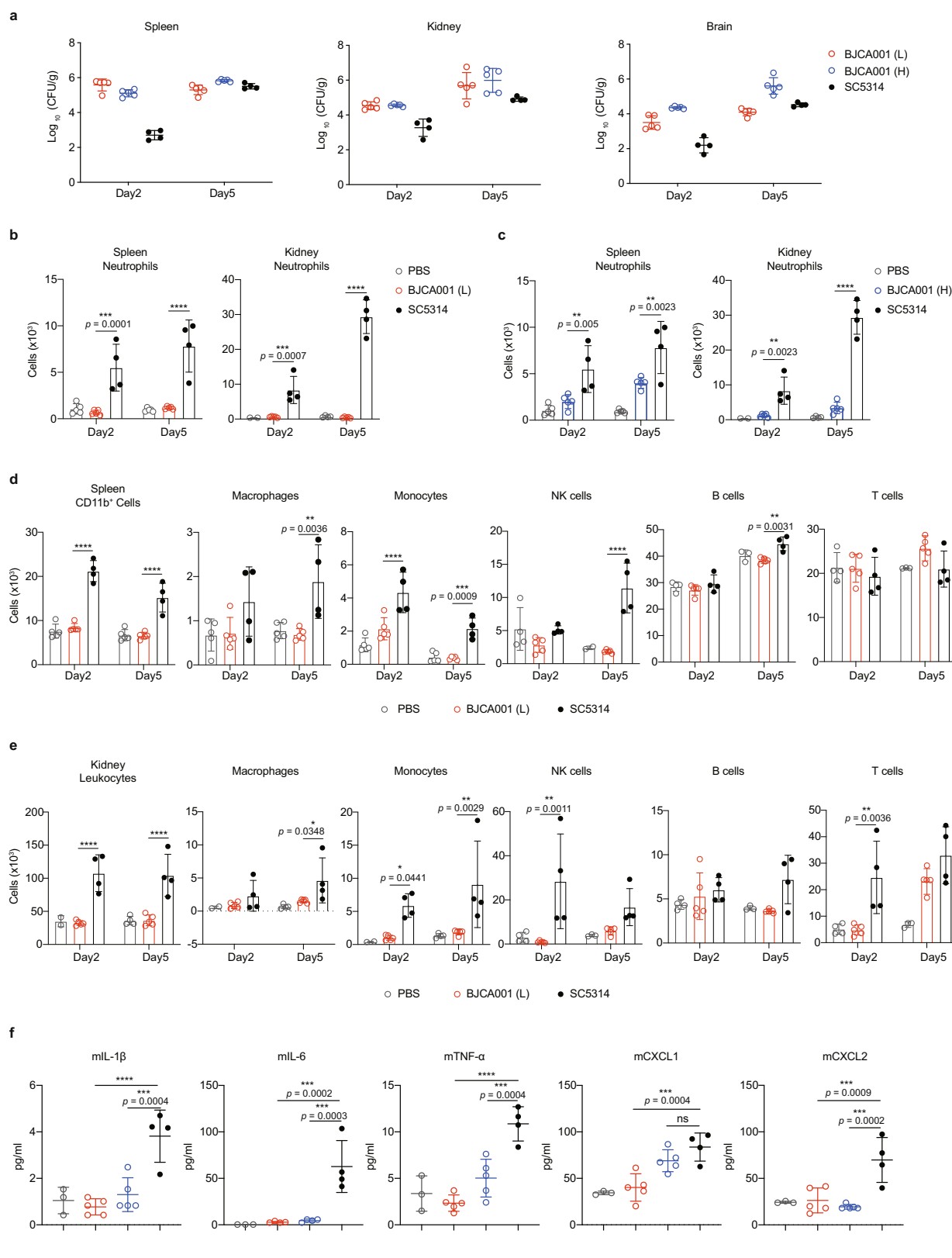

particularly present in significantly higher numbers in *C. albicans*-infected tissues versus *C. auris*-infected tissues (Fig. 3d, e and Supplementary Fig. 8b, c). As expected, no significant changes were observed for the population of B and T cells, which normally accumulate during an adaptive immune response. Thus, our studies show starkly different responses by innate immune

cells to *C. albicans* and *C. auris*, with the latter showing weaker recognition and innate immune evasion. This conclusion was further supported by examining serum levels of pro-inflammatory cytokines and chemokines, as we found that a marked induction of cytokines (IL-1β, IL-6, TNF-α) and chemokines (CXCL1 and CXCL2) were recorded in mice infected with *C. albicans*, however,

**Fig. 3 In vivo studies showing that *C. auris* BJCA001 induces a less potent innate immune response than *C. albicans* SC5314. a** Quantitative fungal burden of spleen, kidney, and brain of C57BL/6 mice ($n = 5$ mice/group/time point) intravenously infected with either $1 \times 10^6$ (L) or $2 \times 10^7$ (H) of *C. auris* yeast cells at day 2 and 5 post-inoculations. For comparison, a parallel experiment was conducted using *C. albicans* ($5 \times 10^4$ CFU; $n = 4$ mice/group/time point). **b, c** Shown are neutrophil populations in spleen (**b**) and kidney (**c**) cell suspensions of mice that were infected without (PBS, negative control) or with live *C. auris* ($n = 5$) or *C. albicans* ($n = 4$) for 2 and 5 days. The number of neutrophils was determined by flow cytometry. *C. albicans* was injected into mice at an inoculum of $5 \times 10^4$ CFU and *C. auris* was injected at an inoculum of $1 \times 10^6$ CFU(L) or $2 \times 10^7$ CFU (H). **d, e** Shown are the major immune cell populations in spleen (**d**) and kidney (**e**) cell suspensions of mice treated as in **b**. The number of each immune population was determined by flow cytometry. **f** Serum cytokine and chemokine levels, as determined by ELISA, in mice treated as in **a**. Data are expressed as mean ± SD and are representative of three independent experiments. ns, no significance; *$p < 0.05$; **$p < 0.01$; ***$p < 0.001$; ****$p < 0.0001$; by two-way ANOVA with Tukey's test (**b–e**) or one-way ANOVA with Dunnett's test (**f**). Source data are provided as a Source Data file.

their levels in the mice challenged with either a high or low dosage of *C. auris* were almost identical to those of uninfected control mice (Fig. 3f). It is noteworthy that the mice exhibit an overall unhealthy status on day 2 when the inoculum size of *C. albicans* was increased to the same level as for *C. auris* ($1 \times 10^6$ CFU/mice). In terms of tissue fungal burden, there was no significant difference between *C. albicans* and *C. auris* (Supplementary Fig. 8d). However, we noticed that serum levels of cytokines/chemokines were significantly higher in *C. albicans*-infected mice than in *C. auris*-inoculated mice (Supplementary Fig. 8e) (the data on day 5 were unavailable because most mice reach the humane endpoint and have to be sacrificed before that time), minimizing the possible effect of inoculum size on innate immune activation.

In conclusion, the data from our in vivo studies are in very good agreement with the results obtained from in vitro experiments, supporting that compared to *C. albicans*, *C. auris* exhibited weaker responses including changes in innate immune cell populations and expression levels of pro-inflammatory cytokines/chemokines, and thus evaded from innate immune clearance.

**Morphology-independent and functionally conserved innate immune evasion of *C. auris*.** Unlike *C. albicans*, *C. auris* has long been recognized as a fungus with no yeast-hyphae switch[3,8,33]. However, Yue et al. unexpectedly found that the isolate (BJCA001) used in this study was able to undergo morphological switches among three distinct cell types, including typical yeast (Y), filamentation-competent (FC) yeast, and filamentous cells (F)[17]. Importantly, the switch between typical yeast and FC/filamentous phenotype turns out to be heritable and triggered by passage through the mammalian body[17]. These observations prompted us to ask whether innate immune recognition may differ in the three cell types of *C. auris*, given the fact that innate immune responses could be discriminated between the yeast and hyphal forms of *C. albicans*[34]. To test this, we sought to carry out in vitro and in vivo assays and compare the innate immune responses among the different cell types. Our in vitro assays using different mammalian cells showed that all three cell types of *C. auris* displayed no differences in cell adherence (Supplementary Fig. 9a) and macrophage cytotoxicity (Supplementary Fig. 9b). Moreover, all three cell types of *C. auris* also failed to induce p38 MAPK activation (Supplementary Fig. 9c) and cytokine production (Fig. 4a, Supplementary Fig. 9d). Consistently, when mice were challenged with either Y or FC form of *C. auris* in two different inoculum sizes, we observed that neither low ($10^6$ CFU/mice; L) nor high ($2 \times 10^7$ CFU/mice; H) dose inoculum shows significant differences in fungal loads (Supplementary Fig. 9e, f), innate immune cell populations (Fig. 4b, c, Supplementary Fig. 9g, h) and cytokine production (Fig. 4d), highly suggesting that innate immune evasion of *C. auris* appears to be morphology-independent.

The isolate BJCA001 used in this study behaves somewhat differently from most typical fluconazole-resistant isolates identified worldwide. It was characterized as susceptible to fluconazole[15] and able to rapidly establish acquired resistance upon fluconazole treatment[16], raising a possibility that the observed immune evasion strategy might be strain specific. To rule out this possibility, we collected 16 fluconazole-resistant clinical isolates (covering all four Clades) and examined the kinetics of infection-induced innate immune responses in macrophages. Among them, 14 strains were isolated in different geographical areas, including 4 from China (Cau-C #1–2, Clade I; Cau-C#3–4, Clade III), 2 from Japan (Cau-J #1–2, Clade II), 6 from India (Cau-I #1–4, Clade I; Cau-I #5-6, Clade III), 1 from Korea (Cau-K, Clade II), 1 from South America (Cau-S, Clade IV), and 2 strains were independently isolated following repeated serial passage of the parental strain BJCA001 in fluconazole-containing medium (BJCA001-R #1–2) (Supplementary Table 2). Fluconazole resistance in each strain was assessed by a checkerboard assay according to the respective minimal inhibitory concentration (MIC) of the drug (Supplementary Fig. 10a). Similar to BJCA001, the rest of *C. auris* strains also failed to elicit meaningful proinflammatory response in BMDMs (MOI = 5; 3 or 6 h), as evidenced by unaltered p38 MAPK phosphorylation (Supplementary Fig. 10b) and decreased expression and secretion of cytokines/chemokines (Fig. 4e, f, Supplementary Fig. 10c, d). Similar results were obtained in experiments with lower doses of inoculum (MOI = 0.1) and longer time in stimulation (24 h) (Fig. 4g, Supplementary Fig. 10e). In contrast, the recognition of *C. albicans* by BMDMs results in the release of inflammatory and chemotactic cytokines, aiding further immune cell recruitment and infection resolution.

However, different results have been reported regarding interactions between *C. auris* and human peripheral blood mononuclear cells (PBMCs). In a recent study, Bruno et al. found that co-incubation of PBMCs with *C. auris* strains from five clades caused even higher expression of proinflammatory cytokines, claiming that *C. auris* might be a strong inducer of innate host defense[14]. To make sure we could reproduce the results, 11 *C. auris* strains in our collection were tested by examining cytokine production in human PBMCs after challenge at MOI 5 and 0.1 for 6 h and 24 h, respectively. The results are summarized in Supplementary Fig. 11a, b. Overall, both *C. albicans* and *C. auris* could effectively induce cytokine production in human PBMCs, when compared to the untreated control. However, all *C. auris* strains tested showed higher levels of TNF-α only in experiments using a high MOI (MOI = 5). Some, but not all of *C. auris* strains elicited higher production of cytokines than did *C. albicans*. In most cases, we observed the same pattern of increased innate immune responsiveness to *C. auris* or *C. albicans*. These results demonstrate differential innate immune responses to *C. auris* seen in different immune cell types, arguing that the exact role of different phagocytes in the context of *C.*

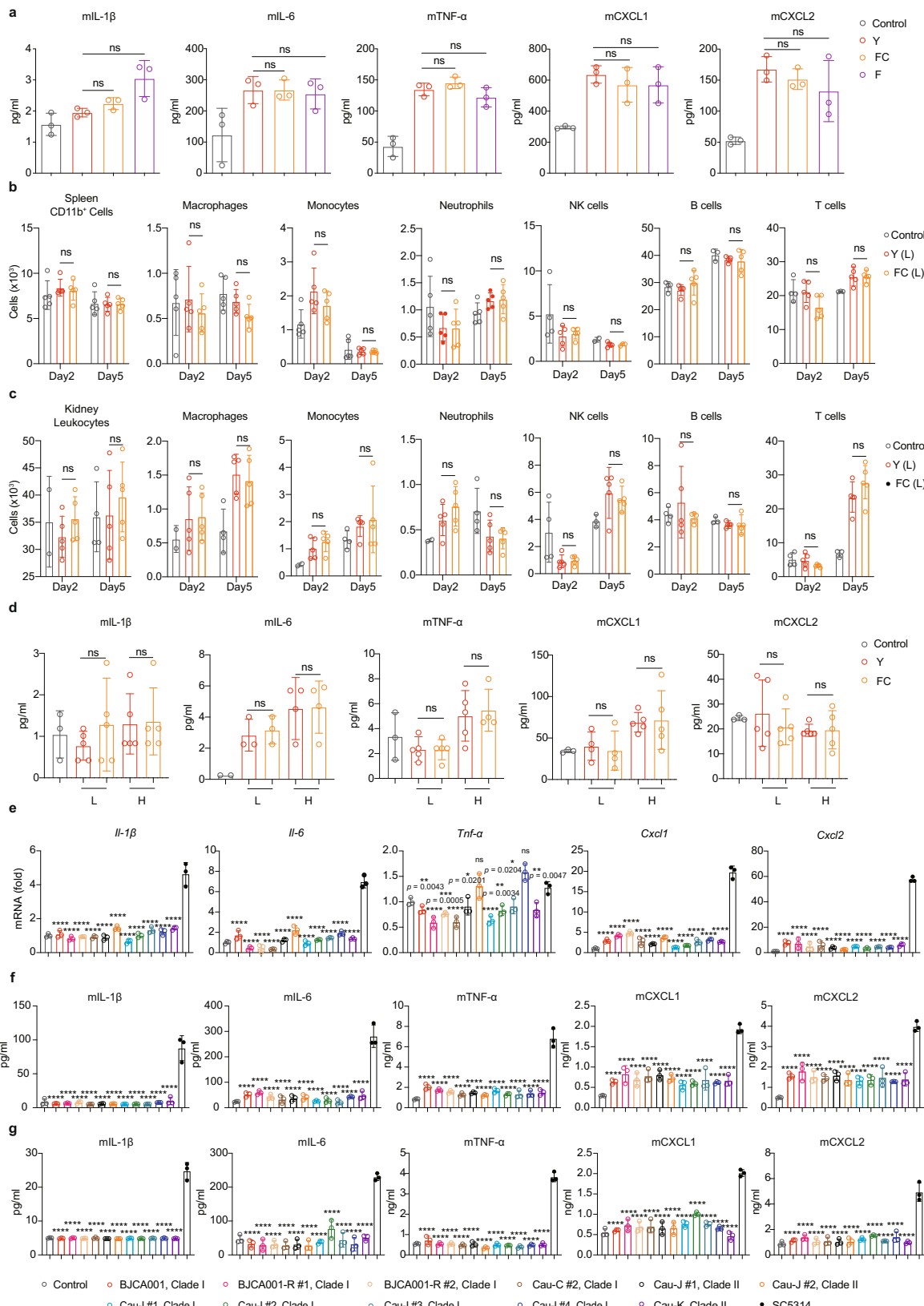

*auris* infection awaits further precise and comprehensive investigations for clarification.

**Comparison of the cell wall structure of *C. auris* BJCA001 and *C. albicans* SC5314.** The fungal polysaccharide cell wall is a complex two-layer structure composed mainly of mannan,

β-glucan, and chitin[35,36]. Meanwhile, the fibrillar outermost layer consists of mannosylated glycoproteins and the inner layer is sequentially assembled with β-glucan and chitin[35,36]. Upon infection, pathogenic fungi expose specific pathogen-associated molecular patterns (PAMPs) at their cell surface that could be recognized by host pattern recognition receptors (PRRs) and

**Fig. 4 Reduced innate immune response to *C. auris* is morphology-independent and functionally conserved. a** Production of TNF-α, IL-6, IL-1β, CXCL1, and CXCL2, as determined by ELISA, in the culture supernatants of BMDM that were stimulated without (PBS) or with the yeast (Y), filamentation-competent (FC) yeast or filamentous (F) form of *C. auris* BJCA001 (MOI = 5) for 6 h (n = 3). **b**, **c** Shown are the major immune cell populations in spleen (**b**) and kidney (**c**) cell suspensions of mice that were infected without (PBS, negative control) or with the yeast (Y; n = 5) or filamentation-competent (FC; n = 5) yeast form of *C. auris* BJCA001 for 2 and 5 days. *C. auris* was injected into mice at an inoculum of 1× 10[6] CFU (L). **d** Serum cytokine and chemokine levels were determined by ELISA, using mice treated as in **b**, **c**. **e–g** The innate immune response against various fluconazole-resistant clinical isolates of *C. auris*. **e** Expression of IL-1β, IL-6, TNF-α, CXCL1 and CXCL2 were analyzed by real-time RT-qPCR. BMDMs were stimulated without (PBS control) or with each of the eleven fluconazole-resistant clinical isolates of *C. auris* or *C. albicans* SC5314 at MOI = 5 for 3 h. Results were normalized to the expression of the control gene GAPDH and are presented relative to those of negative control, set as 1. **f** Production of the cytokines and chemokines were analyzed by ELISA after BMDMs were stimulated with indicated strains at MOI = 5 for 6 h (n = 3). **g** Similar to **f**, except for lower inoculum of *C. auris* (MOI = 0.1) and longer stimulation time (24 h). Data are expressed as mean ± SD and are representative of three independent experiments. ns, no significance; *$p < 0.05$; **$p < 0.01$; ***$p < 0.001$; ****$p < 0.0001$; by one-way ANOVA with Sidak's test (**a**, **d–g**) or two-way ANOVA with Tukey's test (**b**, **c**). Source data are provided as a Source Data file.

potently stimulate innate immune response[37,38]. On the other hand, human fungal pathogens were able to alter cell wall composition and architecture enabling them to evade and escape immune system[22]. We examined innate immune responses in BMDMs after challenge with live or heat-killed *C. auris* yeasts. Only heat-killed, but not live cells, were able to induce cytokine production (Fig. 5a) and p38 MAPK activation (Fig. 5b), supporting the importance of an intact fungal cell wall in the induction of innate immune responses. These results prompt us to take a closer look at the structural complexity of *C. auris* cell wall.

First, we took advantage of methods for high-pressure freezing and freeze substitution (FS) to examine the ultrastructure of *C. auris* BJCA001 or *C. albicans* SC5314 yeast cell wall by transmission electron microscopy (TEM)[39]. To our surprise, we found that the outer mannan fibril layer of *C. auris* cell wall is thicker than that of *C. albicans* and statistically, the mannoprotein fibril length in *C. auris* is longer than that in *C. albicans* (100 nm vs 50 nm in average) (Fig. 5c). Second, we investigated cell wall composition by quantifying the total amounts of three major cell wall polysaccharides (mannan, glucan, and chitin). Both gas chromatography-mass spectrometry (GC-MS) and high-performance liquid chromatography (HPLC) analyses for sugar composition gave similar results (Fig. 5d, e). The major cell wall components of both species are mannan and glucan (β-(1,3)-glucan and β-(1,6)-glucan) and a small quantity of chitin. However, structural differences were detected. For example, *C. auris* BJCA001 showed an increase in the relative proportion of mannan in the cell wall when compared to *C. albicans* SC5314 (40.34% vs 31.42% by GC-MS; 41.40% vs 32.72% by HPLC). Coincidently, the relative proportion of β-glucan in cell wall was much less in *C. auris* than that in *C. albicans* (55.51% vs 66.43% by GC-MS; 55.68% vs 64.33% by HPLC), presumably reflecting the presence of β-glucan masking in *C. auris*. The content of chitin in both strains is almost the same (4.15% vs 2.14% by GM-MS; 2.91% vs 2.95% by HPLC). Notably, the HPLC elution profiles of the two strains were identical (Supplementary Fig. 12) and no additional distinct peaks were present between the two HPLC chromatograms, suggesting that *C. auris* should have no alternative PAMPs which may differ considerably from *C. albicans*. Interestingly, unlike the results from TEM, HPLC analyses of the outer mannan levels only showed a mild increase (10%) in *C. auris*, possibly due to its smaller cell size than *C. albicans*. The changes in cell wall polysaccharide fractions in *C. albicans* and *C. auris* were further confirmed by both fluorescent staining and flow cytometry (Fig. 5f, Supplementary Fig. 13a). When compared to what we observed in *C. albicans*, the proportion of the outer mannan layer was significantly increased in *C. auris* and this pattern appears to be conserved across most *C. auris* strains. A similar exposure pattern was observed during

host-pathogen interaction (Supplementary Fig. 13b). Upon contact with BMDMs, *C. auris* cell wall still displayed an increase in relative proportion of mannan followed by a decrease in β-glucan levels when compared to that of *C. albicans*. To prevent any changes in cell wall structure during the experiments, we also compared β-glucan exposure between PFA- and heat-killed yeast cells, using both Fc-Dectin-1 and β-glucan antibodies. As shown in Supplementary Fig. 13c, same patterns of β-glucan exposure were shown in both staining methods. Moreover, we expectedly observed higher fluorescent intensity in PFA-killed *C. albicans* than in PFA-killed *C. auris*. Compared to PFA, heat treatment disrupted the cell wall structure and markedly increased β-glucan unmasking in both species, and interestingly, its level in *C. albicans* was still higher than *C. auris*. Finally, we used the proton nuclear magnetic resonance ($^1$H NMR) spectroscopy technique to determine and compare the structure of the outer mannan layer. Mannan fraction was extracted from both *C. auris* and *C. albicans*, using a method described previously[40]. The proton NMR spectra of mannans isolated from *C. auris* were significantly different from that of *C. albicans*, as that the acid-labile mannan resonances (5.572, 5.560, 5.179, 4.917, 4.904, 4.840 ppm), which were highly enriched in *C. albicans* mannan, were almost completely lost in *C. auris* mannan (Fig. 5g, h). Based on the data from $^1$H NMR, we generated the most likely mannan structures for *C. albicans* and *C. auris*, respectively (Fig. 5i). Confirmation of *C. auris* acid-labile mannan (phosphomannan) deficiency was observed by the reduced Alcian blue staining (Supplementary Fig. 13d). In summary, our findings suggest that the outer cell wall layer of *C. auris* showing a high density of mannan and low structural complexity, may contribute to a reduced innate immune recognition, compared with *C. albicans*. A relatively simple structure of the cell wall may generate less variability in antigen surface exposure and therefore more easily shield the fungal cells from being recognized by host's innate immune system.

The importance of phosphomannan cell wall content for *C. albicans* recognition by macrophages was reported by recent studies showing that the depletion of phosphomannan resulted in significantly reduced rates of uptake[41–43], raising a question as to whether the lack of phosphomannan may account for innate immune evasion by *C. auris*. We used an indirect approach to test this possibility by generating a phosphomannan-deficient mutant strain (*mnn4Δ/Δ*) in *C. albicans*. Consistent with previous findings[44], the *mnn4Δ/Δ* mutant strain had a complete loss of phosphomannan (Supplementary Fig. 13e) and displayed a significant reduction in macrophage uptake (Supplementary Fig. 13f). Surprisingly, this mutant was still able to induce at least the same or even higher cytokine production when compared to the wild-type (Supplementary Fig. 13g). The same results were obtained in previous studies using PBMCs and

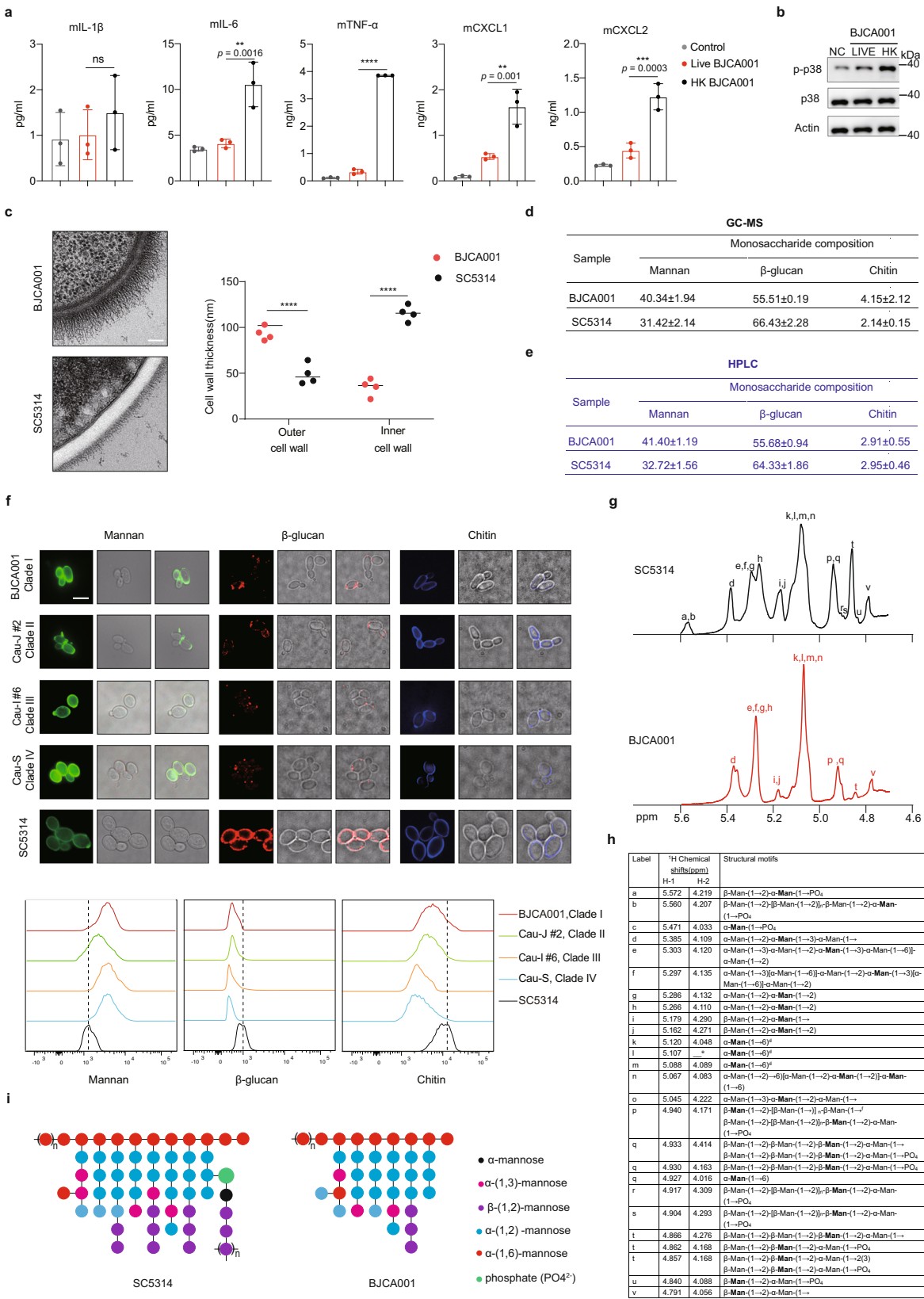

mononuclear cells (MNCs)[39,45]. Furthermore, we also found that *MNN4* deletion had no effect on *C. albicans* pathogenicity because the *mnn4Δ/Δ* mutant was equally virulent in mice compared to its wild-type counterpart (Supplementary Fig. 13h)[44]. Hence, the removal of phosphomanan in *C. albicans* through deletion of *MNN4*, which mimics its deficiency in *C. auris*, may

provide indirect evidence supporting the weak recognition and ingestion of *C. auris* by macrophages.

To further validate the essential role of the outer mannan layer of *C. auris* cell wall in protection against innate immune responses, we focused on various mannan-binding receptors in host, including the mannose receptor (MR), Mincle, DC-SIGN,

**Fig. 5 The distinct structure of the cell wall mannan may shield *C. auris* from innate immune recognition. a, b** The host innate immune response against live or heat-inactivated *C. auris*. BMDMs were stimulated with live or heat-killed (HK) *C. auris* yeast (65 °C for 3 h) at MOI = 5 for 24 h. **a** The production of IL-1β, IL-6, TNF-α, CXCL1, and CXCL2 were analyzed by ELISA (*n* = 3). **b** Cell lysates were subjected to immunoblotting analysis using the indicated p38 MAPK antibodies. Data are representative of two independent experiments. **c** TEM imaging shows the ultrastructural architecture of *C. auris* and *C. albicans* cell wall (left panel, Scale bar = 100 nm). Mannan fibril length (right panel) was measured in four randomly selected cells (*n* = 4). Each cell was measured 10 times in different locations. **d, e** The monosaccharide composition of the cell wall, as determined by GC-MS (**d**) and HPLC (**e**), in *C. auris* and *C. albicans*. Values are percentage of each molecule. **f** Representative fluorescence micrographs showing the three main polysaccharide layers of *C. auris* or *C. albicans* cell wall. Exponentially growing fungal cells were stained with ConA-FITC to visualize mannan, Fc-Dectin-1 to visualize β-glucan, and calcofluor white (CFW) to visualize chitin. Scale bar, 5 μm. The intensity of fluorescence representing the level of mannan, β-glucan, or chitin (upper) and was quantified by flow cytometry (bottom). Data are representative of three independent and reproducible experiments. **g, h** Chemical structure of the cell wall mannan, as determined by NMR analysis, in *C. auris* BJCA001 and *C. albicans* SC5314. **g** The proton NMR spectra of mannans isolated from *C. albicans* and *C. auris* cell wall isolations, showing the structural variations. **h** The proton $^1$H chemical shift and structural motif assignments are presented for the mannans of *C. albicans* and *C. auris*. **i** Proposed mannan structures of *C. albicans* and *C. auris*, as identified by NMR. Data are expressed as mean ± SD and are representative of three independent experiments. ns, no significance; **$p < 0.01$; ***$p < 0.001$, ****$p < 0.0001$; by one-way ANOVA with Sidak's test (**a, c**). Source data are provided as a Source Data file.

TLR4, and TLR2. *C. albicans* stimulation induced higher expression levels of these receptor-encoding genes in BMDMs than did *C. auris* (Supplementary Fig. 14a). Moreover, blockade of Mincle or DC-SIGN in BMDMs, by using specific antibodies, resulted in unexpected cytokine responses seen either increasing or decreasing expression of cytokines in response to *C. albicans* depending on the type of cytokine and blocking antibody tested (Supplementary Fig. 14b, c). Interestingly, induction of certain cytokines was also observed in *C. auris*-infected BMDMs that were preincubated with receptor-blocking antibodies, highlighting the multiplicity of these fungal cell wall-macrophage receptor interactions. Blocking a certain receptor may somehow disturb the dynamic balance between host and pathogen, leading to varied immune responses. More critical studies are needed to precisely address questions about the relative importance of these receptors, as well as their differential roles in response to different fungal species.

**Disruption of manno-glycosylated cell wall proteins in *C. auris* increases innate immune activation**. As noted, the outer layer of fungal cell wall mostly consists of highly glycosylated mannoproteins that are decorated with linear *O*-linked mannan, highly branched *N*-linked mannan, and glycosylphosphatidylinositol anchor[46–48]. In *C. albicans*, glycosylation of cell wall mannoproteins involves activities of a number of enzymes, for example, the families of fungal-specific mannosyltransferases[49] (Supplementary Fig. 15a). Searching the *Candida* genome database for the number of genes encoding mannosyltransferases identified 51 in *C. albicans* and 49 in *C. auris*, respectively (Supplementary Fig. 15b, Supplementary Data 5). Further RT-qPCR analyses indicated that the transcript levels of genes encoding the major protein *O*- and *N*- mannosyltransferases were significantly increased in *C. auris* when compared to their levels in *C. albicans* (Supplementary Fig. 15c), which may explain the observed phenotype that *C. auris* has an extensive outer mannan layer. To further elucidate the importance of mano-glycosylated cell wall proteins in shielding innate immune recognition, we asked whether disruption of manno-glycosylation by deleting any of the mannosyltransferase-encoding genes in *C. auris* could potentially restore innate immune activation. For this purpose, we generated *C. auris* mutants by deleting each of the three putative genes *PMR1*, *PMT1*, and *OCH1*.

*PMR1* encodes a secretory pathway P-type $Ca^{2+}/Mn^{2+}$-ATPase and has been verified to be required for protein glycosylation and cell wall maintenance in *C. albicans*[50]. Moreover, *C. albicans* mutant lacking *PMR1* exhibited an altered cell wall with defects in both *O*-mannan structure and *N*-linked outer chain glycosylation[50]. Searching the released genome sequence of

*C. auris* identified a gene orthologue encoding the protein product that shares 76.6% sequence similarity with Pmr1 of *C. albicans* (Supplementary Fig. 16a). In vitro analysis showed that *pmr1*Δ mutant of *C. auris* displayed significant vegetative growth defects, formed large aggregates and maintained yeast forms (Supplementary Fig. 16b–e). Furthermore, similar to its counterpart in *C. albicans*, the *PMR1*-deletion mutant of *C. auris* also exhibited a weakened cell wall, exemplified by increased sensitivity to cell wall- and cell membrane-damaging agents such as Calcofluor White (CFW), Congo Red and SDS (Supplementary Fig. 16f), and activation of the cell wall integrity pathway (Supplementary Fig. 16g), suggesting that this gene is the functional ortholog of *C. albicans PMR1*. TEM analysis indicated that *PMR1*-deletion impairs the outer mannan fibril layer and reduces the mannoprotein fibril length in *C. auris* (Fig. 6a). The remarkable reduction of mannan content in *pmr1*Δ mutant of *C. auris* was further verified by GC-MS and HPLC analysis (Fig. 6b, c), as well as by fluorescent staining and flow cytometry (Fig. 6d), indicating that knocking out the *PMR1* gene in *C. auris* leads to exposure of glucan and chitin. In agreement with a recent study that disruption of *C. auris* mannosylation enhances phagocytosis and susceptibility to killing by human neutrophils[51], we also found that BMDM uptake of *pmr1*Δ mutant was significantly increased compared to the parent wild-type strain (Fig. 6e) and its high uptake rate correlates with a remarkable decrease in the number of viable yeast cells after 24 h incubation (Fig. 6f). More importantly, stimulation with the *pmr1*Δ mutant increased the level of phosphorylated p38 MAPK (Fig. 6g) followed by increased expression and secretion of cytokines/chemokines (Fig. 6h, i). These results indicate that disruption of the mannan layer of *C. auris* leads to exposure to glucan and chitin, which stimulates innate immune responses.

*PMT1* belongs to one of the *PMT* family members encoding a protein *O*-mannosyltransferase catalyzing the initial step of protein *O*-glycosylation (Supplementary Fig. 15a)[52]. *C. auris* Pmt1 shares a 55.2% sequence similarity with its *C. albicans* orthologue Pmt1 (Supplementary Fig. 17a). In vitro assays revealed that *pmt1*Δ mutant of *C. auris* had similar growth phenotypes as *pmr1*Δ mutant (Supplementary Fig. 17b–d). Similar to *pmt1*Δ/Δ mutant of *C. albicans*[52], *pmt1*Δ mutant of *C. auris* did exhibit growth and morphology defects in cell wall- and cell membrane-damaging agents (Supplementary Fig. 17e), as well as activation of the cell wall integrity pathway (Supplementary Fig. 17f). In line with the observations in *C. albicans*[52], *PMT1*-deletion in *C. auris* resulted in a moderate reduction of mannan content followed by increased exposure of β-glucan and chitin (Fig. 6j), suggesting changes in cell wall compositions. Similar to the results from *pmr1*Δ mutant, we found in *pmt1*Δ

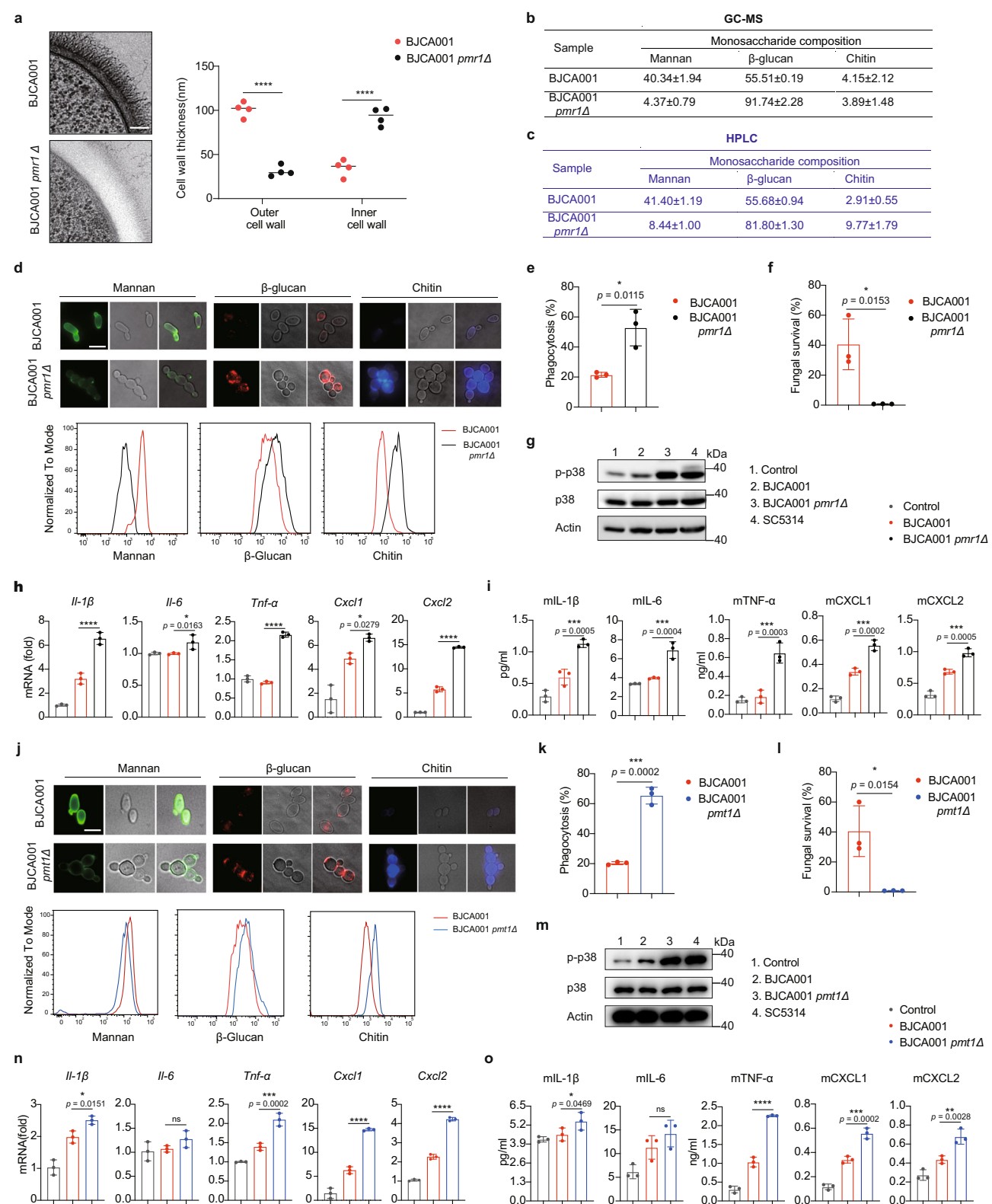

mutant that macrophage uptake was significantly increased (Fig. 6k) and its viability inside BMDMs was decreased after 24 h incubation (Fig. 6l), and this reflects a strong activation of p38 MAPK in BMDMs (Fig. 6m) and moderate increases in the expression and secretion of cytokines/chemokines (Fig. 6n, o). Our results hence suggest that protein *O*-glycosylation in the outer layer of the cell wall is important for protection of *C. auris* against innate immunity.

In yeast fungi like *C. albicans*, *N*-mannan outer chain synthesis is initiated by the α-(1,6)-mannosyltransferase encoded by *OCH1* whose protein product shares 76.6% sequence similarity with Och1 of *C. albicans* (Supplementary Fig. 18a). The *och1*Δ/Δ mutant in *C. albicans* has been found to display a severe defect in *N*-mannan fibrils with elevated β-glucan and chitin levels in cell wall[39]. Repeated attempts to delete the *C. auris* homolog of *OCH1* gene failed, indicating that this gene may be essential for the

**Fig. 6 C. auris mutants with impaired cell wall mannosylation trigger more potent innate immune activation. a** TEM imaging shows the ultrastructural architecture of wild type and pmr1Δ in C. auris cell wall. Mannan fibril length was measured in four randomly selected cells (n = 4). Each cell was measured 10 times in different locations. Scale bar = 100 nm. **b, c** The monosaccharide composition of the cell wall, as determined by GC-MS (**b**) and HPLC (**c**), in C. auris WT or pmr1Δ mutant. **d** Representative fluorescence micrographs showing the three main polysaccharide layers of wild type and pmr1Δ mutant cell wall. Exponentially growing fungal cells were stained with ConA-FITC to visualize mannan, Fc-Dectin-1 to visualize β-glucan, and calcofluor white (CFW) to visualize chitin. Scale bar, 5 μm. The intensity of fluorescence represents the level of mannan, β-glucan, or chitin (upper) and was quantified by flow cytometry (bottom). Data are representative of three independent and reproducible experiments. **e, f** BMDMs were infected with C. auris WT or pmr1Δ mutant at an MOI of 1. The phagocytized fungal cells at 1 h post infection (**e**) or the survived fungal cells at 24 h post infection (**f**) were harvested, diluted, and plated on YPD agar (n = 3). **g** BMDMs were infected with indicated strains and cell lysates were subjected to immunoblotting analysis using the indicated p38 MAPK antibodies. **h** Expression levels of IL-1β, IL-6, TNF-α, CXCL1, and CXCL2, as determined by real-time RT-qPCR, in BMDMs that were infected without (PBS control) or with live C. auris WT or pmr1Δ mutant (MOI = 5) for 3 h (n = 3). Results were normalized to the expression of the control gene GAPDH and are presented relative to those of negative control, set as 1. **i** Production of IL-1β, IL-6, TNF-α, CXCL1, and CXCL2, as determined by ELISA, in the culture supernatants of BMDMs that were infected without (PBS control) or with live C. auris WT or pmr1Δ mutant (MOI = 5) for 6 h (n = 3). **j** Representative fluorescence micrographs showing the three main polysaccharide layers of wild type and pmt1Δ mutant cell wall. **k, l** As in **e, f**, BMDMs were infected with C. auris WT or pmt1Δ mutant at an MOI of 1. The phagocytized fungal cells at 1 h post-infection (**k**) or the survived fungal cells at 24 h post infection (**l**) were determined by CFU counting (n = 3). **m–o** The same as **g–l**, except the indicated strains (n = 3). Data are expressed as mean ± SD and are representative of three independent experiments. ns, no significance; *p < 0.05; **p < 0.01; ***p < 0.001; ****p < 0.0001; by one-way ANOVA with Sidak's test (**a**, **h**, **i**, **n**, **o**) or two-side unpaired t test (**e**, **f**, **k**, **l**). Source data are provided as a Source Data file.

viability of this species. We, therefore, generated a Tetoff-OCH1 mutant in which the endogenous OCH1 promoter in C. auris BJCA001 was replaced with the tetracycline-regulatable promoter (Supplementary Fig. 18b, c). When mutant cells were cultured with doxycycline, growth was unaffected on standard YPD medium (Supplementary Fig. 18d) but the mannan content at the cell surface was remarkably decreased, based on fluorescent staining and flow cytometry (Supplementary Fig. 18e), reconfirming the conserved function of Och1 in the elaboration of the outer N-mannan chains. More importantly, growth in 5 μg/ml doxycycline, which is sufficient to significantly reduce the expression of OCH1 (Supplementary Fig. 18c), significantly stimulated the production of cytokines (TNFα) by BMDMs (Supplementary Fig. 18f), suggesting that protein N-glycosylation in the outer layer of cell wall also plays a key role in suppressing the innate immune response. Notedly, these results are in agreement with the findings from Yadav et al.[53] that removal of N-mannan in C. auris caused β-glucan exposure (unmasking), leading to increased β-glucan receptor (Dectin-1)-dependent elicitation of key proinflammatory cytokines by macrophages like BMDMs.

Collectively, our results support that the mannan-protein layer of the C. auris cell wall contributes to protection against host innate immune responses.

## Discussion

Here, we studied innate immune responses elicited by the emerging fungal pathogen C. auris. Our results indicate that, compared to C. albicans SC5314, C. auris BJCA001 is less immunoinflammatory, possibly due to the exposure of a structurally different outer layer of the cell wall with distinct properties and characteristics related to mannosylated glycoproteins. Our studies support that the outer mannan layer acts to mask the inner layer of β-glucan from exposure and detection by innate immune cells, and therefore plays a key role in protecting C. auris against host innate immune clearance (Fig. 7). Interestingly, recent studies suggest that other Candida species like C. albicans or C. glabrata, may behave similarly or differently as C. auris to induce innate immune responses when the outer mannan layer was disrupted[41,54], arguing that the role of cell wall mannan in innate immune evasion strategies might be similar or divergent in different Candida species, depending on the type of immune cells interacting with the fungus.

Our results differ from the recent observations reported by Bruno et al.[14] in some respects. Bruno et al. evaluated host

immune response to a list of C. auris clinical isolates from five clades, and showed that C. auris appears to induce a stronger innate immune response compared to C. albicans. We sought to identify possible bases for the differences between these two independent studies, which are summarized below.

(1) Cell types. For our work, we compiled murine bone marrow macrophages (BMDMs) datasets to discover potentially different effects of C. albicans and C. auris on host innate immune activation, while Bruno et al. mainly used human PBMCs for tackling similar questions. A recent study by Yadav et al.[53] provided us with a seemingly rational explanation. The differences between these two studies might be due to recognition differences, as the authors argued that during C. albicans-macrophage interaction, N-mannan acts to mask exposure of β-glucan, allowing fungal cells to evade innate immunity by restricting the access of dectin-1 to β-glucan. In contrast, this cell wall component behaves as a major PAMP to induce cytokine response from human PBMCs. Indeed, our results confirmed that during its interactions with macrophages like BMDMs, the mannan layer of C. auris cell wall is associated with an immunosuppressive activity masking cell wall β-glucans from recognition by Dectin-1. Interestingly, when human PBMCs were stimulated with Candida strains (C. albicans SC5314 and all 11 C. auris strains), both Candida species could effectively induce cytokine production compared to the untreated control. We did find much higher cytokine production with C. auris-stimulated cells compared to cells challenged with C. albicans, but this induction pattern took place only in a few tests. In most cases, we observed the same pattern of increased innate immune responsiveness to C. auris or C. albicans. Thus, stimulation with C. auris may yield different cytokine responses, depending on the type of immune cell being tested. Given that human PBMCs are a mixture of immune cells and made up of lymphocytes (B cells, T cells and NK cells), monocytes and dendritic cells, it will be imperative to clarify the actual cell type that specifically and differentially interacts with C. auris and C. albicans.

(2) Fungal strains. The standard C. albicans laboratory strain SC5314 was used throughout our study and three different C. albicans strains, including SC5314, CWZ10061110 and UC820, were selected by Bruno et al. In addition, the clinical isolates of C. auris are different in the two studies.

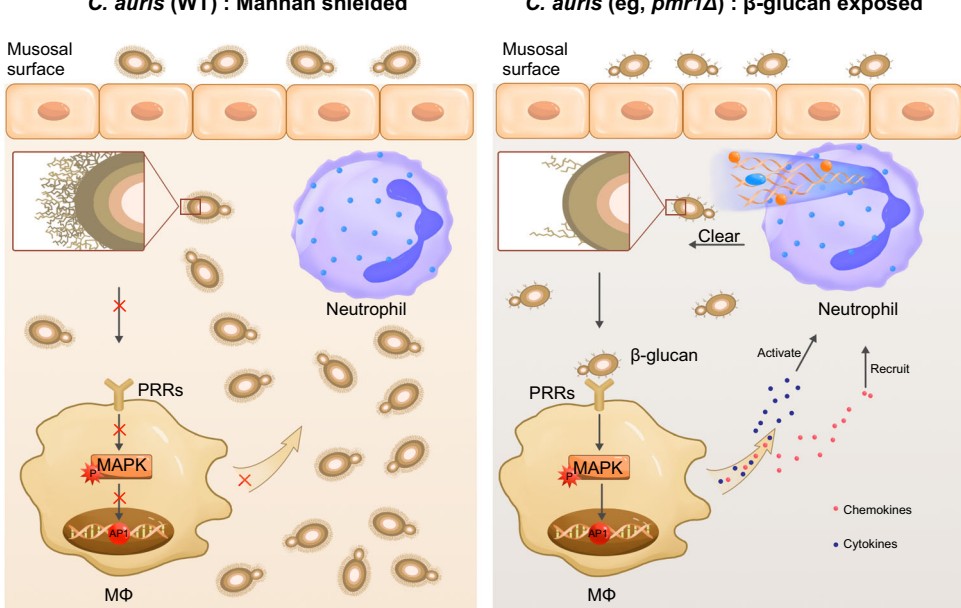

**Fig. 7 Proposed model for the role of *C. auris* outer mannan layer in dampening innate immune responses.** *C. auris* has a structurally distinct mannan layer at the cell wall surface that shields the inner β-glucan, reducing recognition by the receptors of innate immune cells such as macrophages. Reduced macrophage activation leads to successful colonization and proliferation of *C. auris* by avoiding being cleared by immune cells such as neutrophils. In contrast, the disrupted mannan layer in *C. auris* leads to exposure to β-glucan, which can be recognized by the surface receptors of macrophages. The host-pathogen interaction results in the activation of MAPK signaling pathway and release of pro-inflammatory cytokines and chemokines, which finally leads to activation and recruitment of neutrophils and triggers the clearance of invading *C. auris*. Note: Inset represents an enlarged area of the three layers of *C. auris* cell wall, which is organized in an inner skeletal layer comprising chitin, β-glucan, and an outermost layer dominated by highly glycosylated mannoproteins.

Therefore, the impact of differences in strain background, which may vary among PAMPs, could not be excluded. Indeed, we found from the work by Bruno et al. that one isolate (*C. auris* 10051893, clade 1) behaves differently from the others and shows lower phagocytic index during co-culture with BMDMs. Moreover, stimulation with the *C. auris* isolate 10111018 (clade V) significantly decreased the production of cytokines (IL-1β and IL-1RA) from BMDMs. Both results are consistent with our model.

(3) MOIs and inoculum size. Different MOIs were applied for cell culture-based assays in the two studies. However, we found that the failure of *C. auris* to elicit innate immune responses in macrophages and neutrophils of humans and mice is highly conserved in all tested isolates and independent of MOIs. Another factor considered to be important could be inoculum size. Different doses of *C. albicans* and *C. auris* ($5 \times 10^4$ CFU/mice and $1 \times 10^6$ CFU/mice, respectively) were used for our in vivo analyses. However, the same inocula ($1 \times 10^6$ CFU/mice) did not change serum cytokine profiles, minimizing a possible impact of the inoculum level.

(4) Experimental time points. In our work, we disentangled the detailed changes of innate immune response in BMDMs during early stage of infection by *C. albicans* or *C. auris* (3 or 6 h following fungal challenge). Interestingly, when we extend the incubation time to 24 h (the time point used by Bruno et al.), patterns of cytokine induction by *C. albicans* or *C. auris* remain largely unchanged, suggesting that the influence of incubation time could be minor.

(5) Growth conditions and experimental protocols. As described by Bruno et al., both *C. albicans* and *C. auris* strains were prepared by growing cells for 24 h in Sabouraud medium at 30 °C. In our experiments, saturated overnight culture of indicated yeast strain was diluted and incubated to logarithmic growth phase at 30 °C. All strains were routinely grown in YPD medium. It is possibly that growth condition will affect the structure of PAMPs. Moreover, differences between the two results could be due to modified experimental protocols used in the two studies.

In summary, only a few studies thus far have compared the innate immune responses following the challenge with *C. albicans* and *C. auris*. More evidence is needed to draw firm conclusions due to differences in cell types, strains, methodologies employed in these studies.

The outermost layer of *Candida* species is coated with highly glycosylated cell wall proteins that are extensively post-translationally modified through the addition of highly branched *N*-linked mannan, linear *O*-linked mannan, and phospholipomannan[55,56]. Recent studies showed that the chemical structure of the cell wall mannans may differ among *C. auris* strains. For example, we found that compared to that of *C. albicans* SC5314, the outer mannan layer of *C. auris* BJCA001 exhibits less complexity due to the lack of phosphomannan and acid-labile mannan. This distinct structure was further observed in another three clinical isolates (*C. auris* B11203, 16–4, and 17–12 from South Asian clade II)[51,57]. Interestingly, the strains tested by Bruno et al. did not show the loss of phosphomannan at the outer layer[14], which may explain differential immune responsiveness among strains with different backgrounds. There are other, as-yet-undiscovered, factors and mechanisms involved in the recognition of *C. auris* as invading pathogens by cells of the innate immune system.

A series of in vivo assays support our claim that compared to *C. albicans*, *C. auris* is less immunoinflammatory. First, *C. auris*

cells were able to persist in the host and avoid being recognized and cleared by the host innate immune system. Second, major immune cell populations (e.g., neutrophils, leukocytes, CD45[+] cells, macrophages, monocytes and NK cells) were present in significantly lower levels in *C. auris*-infected tissues versus *C. albicans*-infected tissues. Third, serum levels of pro-inflammatory cytokines and chemokines were markedly decreased in mice infected with *C. auris*. In addition, we provide ex vivo results that all four innate immune cell subsets examined, including human and mouse macrophages and neutrophils, displayed weak recognition (uptake, phagocytosis, and killing) with *C. auris*. Consistently, Bruno et al. also found that for mannans from different *C.auris* isolates, their overall binding affinities to mannan recognition receptors (e.g., Dectin-2 and MRs) were more than an order of magnitude lower than those that were observed for *C. albicans* mannans. These results suggest a weaker recognition of *C. auris* by innate immune cells.

In conclusion, our study indicates that *C. auris* is less immunoinflammatory than *C. albicans* SC5314, and persistently resides in the body of infected animals.

## Methods

**Ethics statement**. All animal experiments were performed in compliance with the Regulations for the Care and Use of Laboratory Animals issued by the Ministry of Science and Technology of the People's Republic of China, which enforces the ethical use of animals. The protocol was approved by Institutional Animal Care and Use Committee (IACUC) at the Institut Pasteur of Shanghai, Chinese Academy of Sciences (Permit Number: A2020016).

**Animals**. Female C57BL/6 mice (6–8 weeks old, weighing 18–20 g) were purchased from Beijing Vital River Laboratory Animal Technology Company (Beijing, China). The mice were routinely maintained in a pathogen-free animal facility at a temperature of 21 °C, relative humidity of 50–70%, and under a constant 12-h light/dark cycle. Mice were given free access to food and water throughout the study. All procedures were conducted in compliance with a protocol approved by the IACUC at Institut Pasteur of Shanghai, Chinese Academy of Sciences, China.

**Reagents and antibodies**. All reagents and antibodies used in this study are listed in Supplementary Table 1.

**Fungal preparations**. All strains used in the experiments are listed in Supplementary Table 2. *C. auris* strain BJCA001, the first clinical isolate of *C. auris* in China[15], and *C. albicans* wild-type reference strain SC5314[58], were utilized for all in vitro and in vivo studies. In addition, 16 fluconazole-resistant clinical isolates derived from different geographical regions were used to verify the conservation of the mechanism of innate immune evasion. These strains include 4 from China (Cau-C #1–2, Clade I; Cau-C#3–4, Clade III), 2 from Japan (Cau-J #1–2, Clade II), 6 from India (Cau-I #1–4, Clade I; Cau-I #5-6, Clade III), 1 from Korea (Cau-K, Clade II), 1 from South America (Cau-S, Clade IV), and 2 strains were independently isolated following repeated serial passage of the parental strain BJCA001 in fluconazole-containing medium (BJCA001-R #1–2). Strains were obtained from glycerol stocks stored at −80 °C, plated on YPD (2% Bacto peptone, 1% yeast extract, 2% dextrose), and routinely grown at 30 °C. Prior to treatment in cell culture or challenge in an animal model, the suspension was made by transferring the fungus to YPD liquid medium and incubating at 30 °C with shaking for indicated time periods. Cells were grown exponentially to an $OD_{600}$ of 0.6–0.8, washed three times with PBS, and then used as live yeast. For PFA- and heat-killed yeast, yeast cells were either treated with 4% paraformaldehyde at 4 °C overnight or heated at 65 °C for 3 h. After treatment, yeast cells were washed three times in PBS, enumerated by hemocytometer counts, and tested for viability. *Candida* cell suspensions were diluted at required cell number in PBS for mouse inoculations, or in appropriate cell culture media for co-culture with innate immune cells.

**Cell isolation and culture**. BMDMs were prepared as described previously[59]. Briefly, bone marrow cells were harvested from the femurs and tibias of C57BL/6 mice (6–8 weeks old) and processed by hypotonic lysis to remove erythrocytes. Cells were cultured in RPMI-1640 medium containing 10% fetal bovine serum (FBS), 30% conditioned medium from L929 cells expressing macrophage colony-stimulating factor, and 1% penicillin–streptomycin. Nonadherent cells were removed and adherent cells were 80–90% F4/80[+]CD11b[+], as determined by flow cytometric analysis, and were passaged every 3 days. After six days of incubation, cultures were seeded in six-well plates at a density of $1 \times 10^6$ cells/well, 24-well plates at a density of $1 \times 10^5$ cells/well, or 48-well plates at a density of $2 \times 10^5$ cells/well, and used for infection assays.

Bone marrow-derived neutrophils were prepared as described previously[60]. Briefly, neutrophils were isolated from bone marrow cells by density gradient centrifugation using Histopaque 1077 and Histopaque 1119 (Sigma), and the mononuclear cell layer at the interface was taken and determined by flow cytometric analysis with anti-CD45, anti-Ly6G and anti-CD11b antibody.

Human blood was obtained from volunteering donors with oral consent through a protocol that was approved by the Institutional Review Board and Human Ethics Committee of Shanghai General Hospital, Shanghai Jiao Tong University School of Medicine. Human PBMCs were isolated from blood samples obtained from healthy volunteers by density gradient centrifugation using Ficoll-Paque PLUS (GE Healthcare), and the mononuclear cell layer at the interface was taken. PBMCs was then washed twice with phosphate-buffered saline (PBS; pH 7.4) and resuspended in RPMI-1640 medium (supplemented with 2 mM L-glutamine, 1 mM pyruvate, 10% FBS) prior to use.

Human monocyte-derived macrophages (MDMs) were differentiated from isolated human PBMCs cells as reported[61]. Briefly, $5 \times 10^6$ cells in RPMI-1640 medium were placed in flat-bottom 24-well plates and incubated 2 h at 37 °C and 5% (v/v) $CO_2$. Wells were washed gently with PBS at 37 °C to remove nonadherent cells and cell debris. Then, 1 ml of X-VIVO 15 serum-free medium (Lonza) supplemented with 1% (v/v) PS and 10 ng/ml recombinant human granulocyte–macrophage colony-stimulating factor (Sigma) were added to each well and incubated for 7 days at 37 °C and 5% (v/v) $CO_2$. Fresh medium was exchanged every 3 days.

Human neutrophils were isolated from blood samples obtained from healthy volunteers using MACSxpress negative antibody selection kit and purified with MACSxpress erythrocyte depletion kit (Miltenyi Biotec, Inc., Auburn, CA). Isolated neutrophils were resuspended in RPMI-1640 (lacking phenol red) supplemented with glutamine (0.3 mg/ml) and 2% heat-inactivated FBS. Incubations involving neutrophils were performed at 37 °C with 5% $CO_2$.

For different cell lines such as Hacat, Caco-2, HUVEC, A549, and Hela cells obtained from the American Type Culture Collection (ATCC), cultures were cultivated in DMEM (Dulbecco's modified Eagle's medium) medium containing 10% FBS and 1% penicillin–streptomycin and seeded in 6-well plate at a density of $1 \times 10^6$ cells/well for 24 h prior to use. All cells were regularly maintained in a humidified atmosphere containing 5% $CO_2$ at 37 °C.

**Adhesion assay**. A monolayer cell culture model. Different mammalian cells were grown to confluency in 24-well polystyrene plates and subsequently, culture medium was removed and fresh DMEM without fetal calf serum (FBS) was added to each well. *C. auris* or *C. albicans* grew to the exponential phase in YPD at 30 °C was diluted in DMEM and added to each well with a density of $1 \times 10^5$ cells. After incubation at 37 °C under 5% $CO_2$ for 1 h, wells were washed twice with PBS to remove excess unbound cells. Then cells were scraped, lysed in a lysis buffer (50 mM Tris, pH 7.5, 150 mM NaCl, 1% Triton X-100, 1 mM EDTA), resuspended, serially diluted, and plated onto YPD agar. Fungal colony-forming units (CFUs) were counted after incubation at 30 °C for 48 h.

A three-dimensional human epidermis co-culture model. Reconstructed human epidermis (RHE)- EpiSkin, which is formed from normal human keratinocytes cultured on a collagen matrix at the air-liquid interface and histologically similar to the in vivo human epidermis, was purchased from L'Oreal Research and Innovation Center, Shanghai, China. Upon arrival and prior to experimental procedures, RHE was prepared following the manufacturer's instructions. Live yeast cells of *C. albicans* or *C. auris* ($2 \times 10^6$ CFU/ml) were suspended in sterile PBS and directly added to the RHE tissue. After co-culture with yeast cells at 37 °C under 5% $CO_2$ for 24 h, RHE tissue was cut by a 19-gauge needle and washed with sterile PBS to remove nonadherent cells, and then prepared for either CFU determination or histological PAS staining.

**Endocytosis assay**. Endocytosis of *C. auris* or *C. albicans* yeast cells by endothelial cells was determined by the differential fluorescence assay described previously[62]. In brief, yeast cells ($10^5$) were co-incubated with endothelial cells pre-grown on fibronectin-coated glass coverslips for 45 or 180 min at 37 °C in 5% $CO_2$. After fixation with 4% paraformaldehyde, the adherent but non-endocytosed yeast cells were stained with an FITC anti-Candida antibody (Abcam). The endothelial cells were permeabilized with 0.05% Triton X-100 and yeast cells (including endocytosed and non-endocytosed) were stained with anti-*Candida* antibody (Abcam) conjugated with AlexaFluor 568 (Invitrogen). The coverslips were mounted on microscopic slides and visualized by a confocal microscope. Each experiment was repeated in triplicate at least three times.

**Live video microscopy**. Briefly, $5 \times 10^5$ live yeasts were added to $5 \times 10^5$ macrophages or neutrophils in μ-Slide 4-well chambers (ibidi GmbH, Germany) immediately prior to imaging. Live video microscopy was conducted using an Olympus SpinSR10 Ixplore under the same conditions described for the phagocytosis assay. Images were taken every minute over a 6-h period at a ×63 or ×100 magnification and analyzed using imageJ analysis software. Results are expressed as engaged cells (the percentage of cells engulfing or adherent to fungal cell) and phagocytic index (the total number of fungal cells taken up per 100 cells). Data

were obtained in triplicate from at least 3 separate experiments by analyzing at least 200 macrophages per well.

**Candida-BMDM interaction assay.** BMDMs were seeded in 24-well plates, with coverslips in each well, at a density of $1 \times 10^5$ cells/well and grown overnight at 37 °C under 5% $CO_2$. The next day, plates were washed with PBS and fresh culture media were added. The cells are ready to use.

For macrophage cytotoxicity assay, exponentially growing cells of *C. auris* or *C. albicans* were co-cultured with BMDMs (MOI = 5:1) at 37 °C with 5% $CO_2$ for 2 and 4 h, respectively. After incubation, cells were fixed with 4% formaldehyde, washed, stained with either SYTOX Green (2 h of co-incubation) or propidium iodine (4 h of co-incubation), and observed with a fluorescence microscope (Olympus IX73). Both SYTOX Green and Propidium iodine are membrane-impermeable fluorescent dyes, which are only accessible to nonviable yeast cells. A minimum of 100 cells were counted in each slide, from a total of 6 different slides in 3 independent experiments.

For phagocytosis assay, exponentially growing cells of *C. auris* or *C. albicans* were co-cultured with BMDMs (MOI = 1:1) for 1 h at 37 °C with 5% $CO_2$. After incubation, cells were fixed with 4% formaldehyde, washed, and blocked with the primary anti-*C. albicans* antibody overnight at 4 °C. The fixed cells were washed with PBS and incubated with the fluorescent secondary antibody (FITC-conjugated IgG) for 2 h at room temperature. Wells were washed with PBS, mounted with ProLong™ Gold Antifade Reagent with DAPI (Invitrogen), and examined under fluorescent microscopy (Olympus IX73). The engaged cells and phagocytic index (PI) were determined as above.

For fungal survival assay, exponentially growing cells of *C. auris* or *C. albicans* were co-cultured with BMDMs (MOI = 1:1) for 1 h at 37 °C with 5% $CO_2$. After 60 min of co-incubation, wells were washed twice with PBS to remove excess unbound cells. Then BMDMs were scraped, lysed in a lysis buffer (50 mM Tris, pH 7.5, 150 mM NaCl, 1% Triton X-100, 1 mM EDTA), resuspended, serially diluted, and plated onto YPD agar. For longer time phagocytosis, BMDMs were scraped at 6 h or 24 h post infection, and the phagocytized fungal cells were plated onto YPD agar. Fungal CFUs were counted after incubation at 30 °C for 48 h.

For receptor antibody blocking assay, prior to stimulation with *C. albicans* or *C. auris*, BMDMs were preincubated for 1 h with 1 µg/ml anti-DC-SIGN and 0.6 µg/ml anti-mincle antibodies. After 1 h, cells were stimulated with *C. albicans* and *C. auris* (MOI = 5). A detailed list of specific antibodies is provided in Supplementary Table 1. For dectin-1 receptor antibody block assay, the concentration used was 0.5 µg/ml. Then BMDMs were treated with *C. auris* wild-type strain or mutant strain.

**Mouse model of systemic Candidiasis.** In vivo assessment of innate immune response upon *Candida* infection was carried out using an established murine systemic infected model described previously[63]. Briefly, groups of female C57BL/6 mice (6–8 weeks old, weighing 18–20 g) were inoculated intravenously with different dosages of fungal cells as indicated. For assessment of clinical scores, the health of animals was monitored by two independent researchers daily, based on the following standard procedures: (1) Body weight was recorded every day; (2) After infection, the health status of animals, including weight loss, ruffled coat caused by reduced grooming, decreased movement, abnormal posture like hunched back, and trembling, was checked at least twice a day. A clinical score was determined to evaluate the disease severity of each animal, using parameters that were described previously[64] and mainly include body weight, fur, lethargy, and motility (Supplementary Table 3). Mice were humanely sacrificed when infection had progressed to a human endpoint at which the mice started to suffer. The humane endpoint in our mouse model of systemic candidiasis was defined as signs of severe illness: the mouse has reached a permissible percentage loss of body weight (~20% to 25%) or looks moribund. For fungal burden determination, infected mice were euthanized at indicated time points after infection, and organs, including kidney, spleen, liver, lung, and brain, were aseptically removed, weighed, homogenized, serially diluted, and plated onto YPD agar plates. Fungal CFUs were determined after incubation at 30 °C for 48 h and fungal burdens were expressed as CFUs/g of tissue. For histological analysis, mice were euthanized and organs were removed and fixed with 4% paraformaldehyde. The routinely processed paraffin-embedded tissue sections were stained with PAS or H&E. For gene expression and immunological analyses, the control uninfected mice and mice infected with different fungal strains were euthanized at indicated time points after treatment, and organs (spleen and kidneys) were immediately removed and treated for varying purposes of experiments.

**Murine model of Candida skin colonization.** In vivo assessment of skin colonization of *Candida* spp. was carried out as described previously[30]. Briefly, hairs in mouse dorsal skin were shaved two days prior to topical application. Yeast cells of *C. auris* or *C. albicans* ($1 \times 10^9$ CFUs) were topically applied to the shaved dorsal skin and pinna areas of mice, using a Puritan cotton swab every other day for four times. At defined time points, skin swabs and skin tissue were removed, weighed, and processed for CFU determination, and fungal burdens on the mouse skin surface or within skin tissue were measured.

**Plasmids and strain construction.** Construction of *C. auris* mutants was performed using protocols developed for use in *C. albicans*, with minor modifications (Supplementary Figs. 16b and 18b)[65]. In brief, BJCA001 genomic DNA was used as the template for all PCR amplification of *C. auris* genes. Both upstream and downstream flanking regions of the selected genes were ~1000 bp in length and amplified using appropriate primers. The dominant marker *NAT1* was PCR amplified from plasmid pSFS2A[66]. For generation of *C. auris* mutant lacking *PMR1* and conditional TetOff-*OCH1* mutant, a standard fusion PCR strategy was used following a protocol described previously[67]. For the creation of *PMT1* deletion mutant, both the 5′ and 3′ flanking fragments of gene ORF region were amplified and cloned into the ApaI/XhoI and NotI/SacII sites of plasmid pSFS2A, respectively, generating the knockout plasmid pSFS2A-*PMT1*. Transformation of *C. auris* BJCA001 was conducted using a GenPulser Xcell™ electroporation system (BioRad) according to the manufacturer's instructions. The primers used for PCR amplification are listed in Supplementary Table 4 and plasmids used for gene deletion are listed in Supplementary Table 2.

**Detection of proinflammatory cytokines and chemokines by enzyme-linked immunosorbent assay (ELISA).** For detection of proinflammatory cytokines and chemokines such as IL-1β, IL-6, TNF-α, CXCL1, and CXCL2 in macrophage culture supernatants, BMDMs from C57BL/6 mice and PBMCs from human blood ($2 \times 10^5$ cells/well in 48-well plates) were infected with PBS, *C. albicans* or *C. auris* cells (MOI = 5, or MOI = 0.1 for some experiments) for the time indicated, and cytokine/chemokine production in supernatant was measured by ELISA with ELISA kits according to the manufacturers' recommendations (eBioscience and R&D Systems).

For detection of serum cytokines and chemokines as above, sera were collected from mice infected with *C. albicans* or *C. auris* at different time points and subjected to ELISA analysis.

For detection of proinflammatory cytokines and chemokines in PBMC, PBMCs were isolated and stimulated following the same method[14]. After treatment, supernatants were collected and stored at −80 °C until analysis.

**Quantitation of cytokine and chemokine transcripts by real-time RT-PCR.** The expression levels of proinflammatory cytokines and chemokines such as IL-1β, IL-6, TNF-α, CXCL1, and CXCL2 in BMDMs were determined by Real-time RT-PCR assays. Briefly, BMDMs ($1 \times 10^6$ in six-well plates) were infected with *C. albicans* or *C. auris* cells (MOI = 5) for 1 and 3 h, respectively, and flash-frozen in liquid nitrogen. Total RNA was extracted using TRIZOL (Invitrogen) according to the manufacturer's directions and stored at −80 °C. Samples were reverse transcribed with PrimerScirpt RT Kit (Takara) and subjected to PCR amplification using an ABI 7900HT Fast Real-time PCR System. Relative levels of gene transcripts were quantitatively normalized against the level of mGAPDH. All primers listed in Supplementary Table 4.

**Sample Collection and RNA-seq analysis.** BMDMs ($1 \times 10^6$) were treated with PBS, live *C. auris* or live *C. albicans* for 3 and 6 h. Cells were harvested and immediately frozen in liquid nitrogen and stored at −80 °C. Three biological replicates were performed per strain. Total RNA was obtained from the same preparation as above. RNA quality was assessed and sequenced using Illumina NovaSeq 6000 platform at Beijing Novogene Bioinformatics Technology Co., Ltd. Quality trimming and adapter trimming were performed by Skewer (V0.2.2). Filtered reads were mapped with STAR version 2.7.4. to the mouse genome (GRCm38/mm10; https://useast.ensembl.org/index.html).

The expression profile of each gene was obtained by StringTie (V2.1.2). Each gene expression was normalized by DESeq2 (V1.20), followed by differential expression analysis. Significantly expressed genes were filtered with twofold cutoff and adjusted $p$ value ≤ 0.05. KEGG pathway enrichment analysis identifies significantly enriched metabolic or signaling pathways in the whole genome background using Fisher exact test, taking adjusted $p ≤ 0.05$ as a threshold to identify enrichment pathway. For heatmap visualization of gene clusters, expression values of differential genes were transformed in $\log_2$ scale.

**Western analysis.** BMDMs ($1 \times 10^6$) from C57BL/6 mice were infected with *C. auris* or *C. albicans* cells (MOI = 5) at the indicated times and lysed in RIPA for 30 min on ice. Cell debris was cleared by centrifugation and the protein concentration of cell lysate was measured using Takara BCA kit (T9300A, Takara). After boiled at 100 °C for 10 min, protein samples were separated by SDS-PAGE and transferred onto PVDF membrane. As indicated, blots were incubated with appropriate antibodies.

**Flow cytometry analysis of immune cells and fungal cell wall components.** For leukocyte enrichment from mouse tissues, a method has been adapted from Lionakis et al.[68]. Briefly, mice were euthanized on day 2 and 5 after infection. Spleens were removed from any connective tissues and mechanically digested in cold FACS buffer to produce a single-cell suspension. After passing through a 100-µm filter, the remaining red cells were lysed with RBC lysing buffer. Red cell-free cell suspensions were centrifuged at $400 \times g$ for 10 min at 4 °C and splenocytes were suspended in 8 ml of FACS buffer after passing through a 40-µm filter. In similar,

kidneys were decapsulated and finely chopped with surgical scissors before enzymatic digestion at 37 °C in 3 ml of 1% DMEM containing 0.5 mg/ml collagenase and 0.2 mg/ml DNase I (Roche) for 45 min with intermittent shaking. Mechanical dissociation with an 18-gauge needle resulted in a single cell suspension and the digested tissue was passed through a 70-µm filter, washed, and the remaining red cells were lysed with ACK lysis buffer. Red cell-free cell suspensions were then treated from the same preparations as above. Leukocyte enrichment from kidneys suspended in 40% Percoll (GE Healthcare) was performed by the overlay of the cell suspensions on 70% Percoll, and centrifuged at $600 \times g$ for 20 min at RT. The leukocytes enriched at the interphase were isolated, washed three times in 5% DMEM, and suspended in FACS buffer. A total of $1 \times 10^6$ live cells were blocked with CD16/CD32 and stained with antibodies listed in Supplementary Table 1. Dead cells were excluded with Fixable Viability Dye eFluor™ 780 (eBioscience). A detailed description of gating strategy is provided in Supplementary Fig. 19.

For quantitative measurement of cell wall components, we used a method described previously[69], with minor modifications. Yeast cells ($1 \times 10^7$) of $C.$ $albicans$ or $C.$ $auris$ were prepared as indicated, washed three times with PBS and fixed in 4% paraformaldehyde at room temperature for 1 h or 4 °C overnight. For mannan labeling, fixed cells were washed with PBS and stained with Concanavalin A (ConA; 50 µg/ml) for 1 h at 37 °C in the dark. For β-glucan labeling, PFA-killed cells were washed with PBS and stained with Fc-hDectin-1a (Invitrogen, fc-hdec1a; 0.2 µg/ml) at 4 °C for 24 h. The fluorescent AlexaFluor 488 conjugated anti-human IgG antibody was used as a secondary antibody and added to the primary stained yeast cells for 1 h incubation at 37 °C in the dark. For chitin labeling, fixed cells were stained with calcofluor white (CFW; 30 µg/ml) for 10 min at 37 °C in the dark. After staining, cells were centrifugated and washed in PBS three times. Data were collected using BD LSR Fortessa flow cytometer (BD Bioscience). Data were analyzed using FlowJo (Treestar, Ashland, OR, USA) software. All data presented represents a triplicate of biological replicates.

**TEM combined with freeze substitution.** Exponentially growing cells of $C.$ $auris$ or $C.$ $albicans$ were harvested and frozen in liquid nitrogen at high pressure using an EM ICE high pressure freezing machine (Leica). Freeze substitution of the frozen cells was carried out in an automatic temperature-controlled freeze substitution system (AFS; Leica Microsystems) in dried acetone containing 2% osmium tetroxide and 0.1% uranyl acetate at −90 °C for 72 h. The samples were gradually warmed to 4 °C. After being washed with acetone three times (15 min/each), samples were infiltrated and embedded in Eponate 12 resin. After polymerized at 37 °C for 12 h and 65 °C for 48 h, the resin blocks were cut with an ultra-microtome (Leica UC7). Serial sections of 70-nm thickness were collected on 150 mesh formvar coated copper grids and counterstained with 3% uranyl acetate for 10 min, followed by lead citrate for 5 min. Then the grids were examined under a TEM (Thermo Fisher/FEI, Talos L120C) with a voltage acceleration of 120 KV.

**Confocal laser scanning microscopy.** Following the same procedures used for the examination of fungal cell wall components by flow cytometry, the stained cells were washed with PBS and scanned at ×63 magnification with a confocal laser scanning microscope (Olympus FV-1200). Pictures were taken and analyzed by ImageJ program.

**Isolation of cell wall Mannan and $^1$H NMR spectroscopy.** The cell wall mannans of indicated $C.$ $auris$ or $C.$ $albicans$ strain were isolated using a mild alkali extraction method described previously[70]. In brief, yeast cells were boiled for 20 min in a mild alkaline solution. The extracted samples were harvested by centrifugation, neutralized, and treated with 50 mg/sample pronase at 37 °C for 18 h. After carbohydrate precipitation by methanol, samples were allowed to sit and settle the materials. The supernatant was decanted and the pellets were dried, resuspended in ddH$_2$O, frozen, lyophilized, and stored at −20 °C until analysis. The structure of cell wall mannan isolated from $C.$ $auris$ BJCA001 or $C.$ $albicans$ SC5314 was analyzed by the $^1$H NMR spectroscopy technique, following a procedure described previously[70].

**GC-MS and HPLC analysis of $Candida$ cell wall.** Exponentially growing cells of $C.$ $auris$ or $C.$ $albicans$ were harvested and washed with sterile water and lysed in a bead beater (15 cycles of 6.0 m/sec, 40 sec with 5 min on ice between cycles). Cell pellets were washed five times with 1 M NaCl and pellets boiled in protein extraction buffer (50 mM Tris-HCl pH 6.8, 2% SDS, 1 mM EDTA, and 0.3 M 2-mercaptoethanol) for 10 min to remove contaminating cytoplasmic proteins. Cell wall pellets were washed with water, snap-frozen in liquid nitrogen, and freeze-dried. The cell wall (2–3 mg) was hydrolyzed with trifluroacetic acid for 3 h at 100 µC. Hydrolyzed samples were washed and resuspended in water to a concentration of 10 mg/ml and 1:10 dilutions were analyzed by HPLC and the relative proportions of sugars in the cell wall were calculated.

The sugar composition of $C.$ $auris$ or $C.$ $albicans$ cell wall was determined by GC-MS, using the procedure described above except analysis by GC-MS.

**Alcian Blue binding assays.** The phosphomannan content was determined by measuring the binding of the cationic dye Alcian Blue to cell surface. Briefly, cells in the exponential growth phase were collected and washed twice with deionized

water, then adjusted to an OD$_{600}$ of 0.2. Then cells were suspended in 1 mL of 30 µg/ml Alcian Blue (in 0.02 M HCl) at room temperature for 15 min. The cell suspension was centrifuged at $13,800 \times g$ for 10 min and the supernatant was collected and used to quantify the content of Alcian Blue absorbance at 620 nm. The concentration of free (non-bound) dye was then determined and used to calculate the amount of Alcian Blue bound to cells.

**Statistical analysis.** Statistical parameters, including the exact value of $n$ and statistical significance, are reported in figures and figure legends. Statistical tests were performed with GraphPad PRISM software 8.2.1 (GraphPad Software, La Jolla, CA, USA). Specific statistical tests are indicated in the figure legends.

**Reporting summary.** Further information on research design is available in the Nature Research Reporting Summary linked to this article.

## Data availability

The authors declare that the data supporting the findings of this study are available within the article and its Supplementary Information files. RNA-Seq data can be found under the GEO accession number GSE203508. Source data are provided in this paper.

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

## Acknowledgements

The authors thank Dr. Linqi Wang from the Institute of Microbiology, Chinese Academy of Sciences, and Dr. Ding Chen from Northeastern University, China for providing strains. And we thank all the lab members in both Institut Pasteur of Shanghai, Chinese Academy of Sciences and State Key Laboratory of Pathogen and Biosecurity, Beijing Institute of Microbiology and Epidemiology, for their help in the discussion and preparation of the manuscript. We also thank the Core Facility for Cell Biology, CAS center for excellence in Molecular Cell Science, and the Electron Microscopy Center of Shanghai Institute of Precision Medicine, Shanghai Ninth People's Hospital, Shanghai Jiaotong University School of Medicine, for their technical support and assistance in the electron microscopy. CC is supported by grants from the MOST Key R&D Program (2020YFA0907200); National Natural Science Foundation of China (32170195, 31870141); The Shanghai Municipal Science and Technology Major Project (2019SHZDZX02); The Key Research Program of the Chinese Academy of Sciences (KGFZD-135-19-11,153831KYSB20170043); the Innovation Capacity Building Project of Jiangsu Province (BM2020019). D.Z. is supported by a grant from the National Key R&D Program of China (2018YFC1200100). X.H. is supported by the National Natural Science Foundation of China (32070146, 31600119); the Youth Innovation Promotion Association, CAS; Natural Science Foundation of Shanghai (20ZR1463800).

## Author contributions

D.Z. and C.C. conceived and designed the study; D.Z., C.C., Y.W., Y.Z., and X.C. performed data analysis and wrote the manuscript; Y.W., Y.Z., H.L., Z.Y., Y.Z., X.H., W.Y., C.X., T.J., Q.T., Z.Z., Y.J., Y.L., L.H., and J.Z. conducted all the experiments and

performed the statistical analysis of the data; B.Z. performed the [1]H NMR analysis; Y.X. performed the GC-MS analysis; Y.Z. and W.F. performed HPLC analysis; H.C. performed TEM analysis; H.L. and R.Z. provided blood samples and performed PBMCs-related analysis; X.C. performed RNA-seq analysis; D.Z., C.C., Y.W., Y.Z., X.C., J.Z., G.H., and N.L. discussed the experiments and results.

## Competing interests
The authors declare no competing interests.

## Additional information

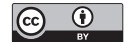

