## [Peer Review File · Nature Communications]

Reviewers' Comments:

Reviewer #1:

Remarks to the Author:

This study compared the host immune response against *C. auris* with that to *C. albicans* using both in vitro and in vivo models. The authors suggest that *C. auris* is poorly recognized and fails to trigger the immune response in comparison to *C. albicans*. The study also investigated the mannan structure of the outer layer of *C. auris* cell wall and showed that the outer mannan layer is important for protecting the exposure of inner β -glucan layer, similar to *C. albicans*. This study is similar to a recent Nature Microbiology paper by Bruno et al, but differs in some conclusions. Bruno et al. also investigated the host immune response against *C. auris* isolates, in comparison to *C. albicans* isolates, and elucidated the innate defense mechanism against *C. auris*. Bruno et al. showed that *C. auris* induces a specific transcriptome in human mononuclear cells, a strong cytokine response, but a low macrophage lysis capacity. The strong cytokine response is due to faster cell doubling of *C. auris* in macrophage than *C. albicans*. They also showed that *C. auris*-induced immune activation is mediated mostly through mannoproteins, and the mannans have features unique to *C. auris*. Both studies analyzed the mannan structures in *C. auris* isolates. *C. auris* is less virulent than *C. albicans* in in vivo models of disseminated candidiasis as shown by both studies and other publications. Different from Bruno et al. this study concluded that *C. auris* cells are poorly recognized by host immune cells and stimulated much lower levels of immune response in comparison to *C. albicans*. Another major difference in data is that high fungal loads of *C. auris* were observed in multiple organs in mice in this study. The different conclusions from this study was not due to the particular *C. auris* strain used, as other *C. auris* isolates behaved similarly in their in vitro cytokine assays. This study validated the protection of β -glucan exposure by the outer mannan layer in *C. auris* by deleting PMT1, which is important for protein glycosylation in *C. albicans*. Why similar studies reach different conclusions need to be addressed.

Major problems with this study:

1. Throughout the manuscript, authors used "innate immune evasion by *C. auris*", but did provide direct experimental data to demonstrate that *C. auris* is poorly recognized by immune cells. Their data in Supplemental Fig. 2d showed phagocytosis % of BJCA001 and SC5314 by BMDMs in 1 hr are about 20% and 30%. This is similar to Fig. 2 in Bruno et al. where dynamics of *C. auris* phagocytosis by macrophage is extensively investigated. Bruno et al. conclude that the difference between *C. auris* and *C. albicans* is not at initial uptake.
2. The difference in phagocytosis between *C. auris* BJCA001 and *C. albicans* SC5314 (20% vs. 30%) cannot explain the dramatic differences in cytokines induced by *C. auris* and *C. albicans* observed in this study. Also, their cytokine results are very different from that published by Bruno et al. In addition to differences in MOI and types of myeloid cells used, another major difference in exposure time. This study used 3 hrs. Bruno et al. used 24 hrs. Growth of *C. auris* inside myeloid cells was suggested to contribute to the strong cytokine production induced by *C. auris*.
3. The in vivo cytokine and CFU data are also very different between two studies. This is surprising consider that both studies use the same mouse model and C57BL/B6 mice. Bruno et al. injected 10⁶ *C. auris* or *C. albicans* per mouse. This study used different inoculums (10⁷ and 10⁶ for *C. auris*, and 5x10⁴ for *C. albicans*) in all in vivo experiments in this study.
4. β -glucan is exposed in the pmt1 mutant of both *C. auris* and *C. albicans*. It does not suggest that *C. auris* uses a novel mechanism to evade host immune detection. In fact, the level of β -glucan exposure is similar in *C. auris* BJCA001 and *C. albicans* SC5314 in this study (Fig. 5e). This data does not support authors' conclusions.

Reviewer #2:

Remarks to the Author:

This paper describes an impressive and interesting dataset relating to the immunogenicity and virulence of the emerging fungal pathogen, *Candida auris*. The authors report that *C. auris* is less

virulent than *C. albicans* in a murine model of systemic candidiasis, that these pathogens differ in their ability to stimulate innate immune responses, and that structural differences in the outer mannan of their cell walls correlate with their different immunostimulatory capacities. The authors are unfortunate for two reasons: (a) a parallel study was published recently in *Nature Microbiology* by Bruno et al., and (b) this new study appears, at least superficially, to contradict some findings reported by Bruno et al.. Consequently, the novelty of some findings in this new paper has been compromised, and interesting questions have arisen about the underlying reasons for the differences between these two studies.

Both studies report that *C. auris* is less pathogenic than *C. albicans* in murine models of systemic candidiasis but, unlike the Bruno study, this new study reports that fungal burdens are maintained or increase over time. Different *C. auris* isolates are used in each case. Both studies investigate the structure of *C. auris* mannan, the Bruno study describing a "unique" type of mannose side-chain. Bruno et al reported that *C. auris* is more immunostimulatory than *C. albicans*, whereas this study suggests that *C. auris* is less immunostimulatory. Is this because each study used different *C. auris* isolates, different immune cell types (PBMCs versus BMDMs), different MOIs, different experimental time points, and/or different fungal growth/fixing protocols? All of these factors are known to influence immunological outputs. For example, mannan is not an immunostimulatory PAMP for macrophages but is for monocytes, and fungal PAMP exposure is known to change with growth phase.

In the end, the differences between these studies are likely to prove highly informative, but this will require a significant amount of additional work. An alternative strategy might be to shift the focus of this interesting story by placing a more positive spin on the differences between the two studies, and by placing more emphasis on the interesting cell wall phenotypes of the *C. auris* *pmr1* and *pmt1* mutants that are currently buried in the supplementary data. It might also be advisable to downplay the idea that *C. auris* evolved a "unique immune evasion strategy" because this species only emerged recently as a fungal pathogen of humans (where did it evolve this strategy?) and because mannan is known to shield β -glucan in other pathogenic *Candida* species.

Reviewer #3:

Remarks to the Author:

The major findings from this work are that *C. auris* does not elicit a strong innate immune response in mice due to enhanced/altered mannan composition of the fungal cell wall, which potentially masks beta-glucan. The manuscript combines both in vivo and in vitro experimental data to investigate possible reasons for the lack of immune response againsts *C. auris*, which require further confirmation. Comments and suggestions for improvement are listed below:

Major comments

- When analysing immune cells by FACS why did you only include neutrophils, one of the powers of FACS is to be able to identify multiple cell populations from a single sample.
- From your histology what morphology is *C. auris* is, and does this vary in the different niches?
- Isn't *C. auris* more of a skin coloniser than a gut coloniser could this not explain what it is less able to adhere to gut cells, did you look at skin cells/models of infection?
- Why use BMDMs, why not quantify phagocytosis/killing in human monocyte derived macrophages, as these would be more relevant, as would the inclusion of human neutrophils.
- In some places a student's T-test is used to analyse data with greater than two samples sizes, please repeat statistical analysis with more appropriate methodology that accounts for multiple comparisons, and potentially non-normally distributed data.
- HPLC of the cell wall dry weight would enable you to quantify the relative % of mannan glucan and chitin in the cell wall, and could also identify alternative PAMPs not present in *C. albicans*.
- *C. auris* lacks phosphomannan and the acid liable mannan, which are known to be recognised by specific PRRs on innate immune cells, have you looked into the role of these receptors? If you use the *C. albicans* mutants that are defective in acid liable mannan do you see an attenuation of virulence in your models?
- The exposed glucan levels are comparable between *C. auris* and *C. albicans* suggesting concealment of beta-glucan from Dectin-1 recognition is not the driving factor here, and the text

should be modified to reflect this. The model presented in Fig 7 should also be updated. The legend states that upon contact with an immune competent host *C. auris* masks its glucan, but there is no data present to support this. The cell wall staining was all performed on lab grown cells, no other conditions were tested to demonstrate that the cell wall actually changes upon contact with the host. The glucan staining also does not look as it should, glucan is not normally exposed in SC5314 as shown. The beta-glucan antibody used is known to have problems, Fc-Dectin-1 is a much more reliable way to quantify glucan exposure.

- Did you do basic analysis of the cell wall form all the isolates you tested, and where there differences between them?

-

Minor comments

- Why apply a 2-fold cut off to the RNA seq data, why not just focus on adjusted P values?

- Line 282: you say that *C. auris* does not amount an innate immune response, but here you say *C. auris* is more tolerant to against the innate immune response, seems contradictory

- Line 432: are you sure your measurements are right here. Previous publications have indicated that the *C. albicans* mannan fibrils are between 50-100 nm, while you say they are only 1.5 nm? The included images are too small to be able to visualise the fibrils in detail. Your % of mannan seems comparable to other studies, so the difference reported in fibril length is unlikely to result from different growth conditions. It's more likely that these differences are due to the fixative method used. The gold standard for TEM to visualise the mannan fibrils is high pressure freeze substitution, as chemical fixation does not preserve the mannan layer. Therefore, not sure how reliable these measurements are.

- Does deletion of PMR1 and PMT1 in *C. auris* affect growth and morphology?

- Discussion seems a little long and unfocused

- English in places could be better

Response to the reviewers

Reviewer #1:

1. Major issues

Q1. *Why similar studies reach different conclusions need to be addressed.*

Answer: We thank the reviewer for this critical question. Actually the other two reviewers had the same concern. It is true that our data contradicts the findings from a recent study (Bruno *et al.*, Transcriptional and functional insights into the host immune response against the emerging fungal pathogen *Candida auris*. *Nat Microbiol.* 2020; 5(12): 1516-1531). In that study, the authors also evaluated the host immune response against a list of *C. auris* clinical isolates from five clades, and the main conclusion is that *C. auris* appears to induce a stronger innate immune response compared to *C. albicans*. However, the evidence from our work revealed that compared to *C. albicans*, *C. auris* acts as an innate immune silencer, rather than an inducer. We thoroughly analyzed the possible source of discrepancies in these two independent studies and sought to find valid explanations for the disagreement, which are summarized below.

- 1) **Cell types.** For our work, we compiled murine bone marrow macrophages (BMDMs) datasets to discover potentially different effects of *C. albicans* and *C. auris* on host innate immune activation, while Bruno *et al.* mainly used human PBMCs for tackling similar questions. A recent study by Yadav *et al.* provided us a seemingly rational explanation. This discrepancy might be due to the recognition difference, as the authors claimed that during *Candida*-macrophage interaction, *N*-mannan at the surface of *Candida* cell wall acts to mask exposure of β -glucan, allowing fungal cells to evade innate immunity by restricting the access of dectin-1 to β -glucan. In contrast, *N*-mannan behaves as a major PAMP to induce

cytokine response from human PBMCs. Indeed, our results confirmed that during its interactions with macrophages or neutrophils from human and mouse, the mannan layer of *C. auris* cell wall is tightly associated with an immunosuppressive activity masking cell wall β -glucan from recognition by Dectin-1. We found that *C. auris* cell wall displayed increase in relative proportion of mannan followed by a decrease in β -glucan levels when compared to that of *C. albicans* (Different techniques were used in our assays, including TEM, GC-MS, HPLC, ^1H NMR, Fluorescent staining and Flow cytometry) (**Fig. 5; Supplementary Fig. 11**). Strikingly, disruption of the outer mannan layer by knocking out *PMR1* or *PMT1* in *C. auris* leads to the unmasking of β -(1-3)-glucan and induction of innate immune responses (**Fig. 6; Supplementary Fig. 14 and 15**). Using the Tetoff-*OCH1* mutant of *C. auris*, we further demonstrate that the protein *N*-glycosylation in the outer layer of cell wall plays a key in suppressing the innate immune response and the removal of *N*-mannan in *C. auris* caused β (1,3)-glucan exposure (unmasking), leading to increased β -glucan receptor (dectin-1)-dependent elicitation of key proinflammatory cytokines by macrophages like BMDMs (**Supplementary Fig. 16**). Our results thus strongly suggest a highly conserved innate immune evasion strategy that *C. auris* employs its distinct mannan-protein layer of cell wall to withstand host innate immune responses, allowing for the successful host invasion. (**Lines 395-441 on pages 21-23; Lines 490-556 on pages 25-28**)

To make sure whether we could reproduce the results, we performed ELISA assays using all 11 *C. auris* strains in our collection and examined cytokine production in human PBMCs after challenge at MOI 5 and 0.1 for 6 h and 24h, respectively (Note: Bruno et al. did the same assay using a low MOI of 0.1 and a 24-h coinubation). Our results did show some differences (**Supplementary Fig. 9a, b**). Unlike what we observed in macrophages, both *C. albicans* and *C. auris* could effectively induce

cytokine production in human PBMCs, when compared to the untreated control. We did find much higher cytokine production with *C. auris*-stimulated cells compared to cells challenged with *C. albicans*, but this induction pattern took place in only a few tests. For example, all *C. auris* strains tested showed higher levels of TNF α only in the experiment using a high MOI (MOI=5). Some, but not all of the *C. auris* strains elicited higher production of cytokines than did *C. albicans*. In most cases, we observed the same pattern of increased innate immune responsiveness to *C. auris* or *C. albicans*. Our results demonstrate differential innate immune responses to *C. auris* seen in different immune cell types, arguing that the exact role of different phagocytes in the context of *C. auris* infection awaits further precise and comprehensive investigations for clarification. Given that human PBMCs are a mixture of immune cells and made up of lymphocytes (B cells, T cells and NK cells), monocytes and dendritic cells, it will be imperative to find out the actual cell type that specifically and differentially interacts with *C. auris* and *C. albicans*. **(Lines 359-378 on pages 19-20)**

- 2) **Fungal strains.** The standard *C. albicans* laboratory strain SC5314 was used throughout our study while three different *C. albicans* strains, including SC5314, CWZ10061110 and UC820, were selected by Bruno *et al.* Of course, the clinical isolates of *C. auris* are totally different in the two studies. As for *C. auris* strains, Bruno *et al.* used isolates covering all five clades and we used 11 clinical isolates belonging to clade I and II. Therefore, the impact of differences of strain background, which may vary among PAMPs, could not be excluded. Indeed, we found from the work by Bruno *et al.* that one isolate (*C. auris* 10051893, clade I) behaves differently from the others and shows lower phagocytic index during the coculture with BMDMs. Moreover, stimulation with the *C. auris* isolate 10111018 (clade V) significantly decreased the production of cytokines

(IL-1b and IL-1RA) from BMDMs. In this case, their results somehow favor our model.

- 3) **MOIs.** Different MOIs were applied for cell culture-based assays in the two studies. For example, we used a MOI of 1 for assessment of percentage uptake and phagocytic index, and a MOI of 5 for all cytokine production assays. However, Bruno et al. used a MOI of 3 for phagocytosis assays and a MOI of 0.1 for cytokine production assay. But, we don't think that the disparity between the two studies mainly stems from the influence of different MOIs, as we found that the failure of *C. auris* to elicit innate immune responses in the macrophages and neutrophils of human and mouse is highly conserved in all tested isolates and independent of MOIs **(Fig. 4g). (Lines 355-358 on page 19)**

Another factor considered to be important could be the inoculum size. It has to be noted that different inoculum doses of *C. albicans* (5×10^4 CFU/mice) and *C. auris* (1×10^6 CFU/mice) were used in our *in vivo* animal studies, largely because of the fact that the mice inoculated with *C. albicans* at 1×10^6 CFU experience a clinical illness manifested by ruffled fur, reduced activity and weight loss, and died rapidly within 5 days **(Supplementary Fig. 5)**. In comparison, those receiving a lower inoculum (5×10^4 CFU/mice) could live long and healthy during the period of innate immune response (usually within the first 7 days). We did perform the experiment by inoculating the mice with the same doses of *C. albicans* or *C. auris* (1×10^6 CFU/mice). As expected, the mice exhibit an overall unhealthy status on day 2 when the inoculum size of *C. albicans* was increased to the same level as for *C. auris*. In terms of tissue fungal burden, there was no significant difference between *C. albicans* and *C. auris* **(Supplementary Fig. 6d)**. However, we still noticed that serum levels of cytokines/chemokines were significantly higher in *C. albicans*-infected mice

than in *C. auris*-inoculated mice (**Supplementary Fig. 6e**) (the data on day 5 were unavailable because most mice were found dead), minimizing the possible effect of inoculum size on innate immune activation. (**Lines 255-261 on page 14; Lines 299-307 on page 16**)

- 4) **Experimental time points.** In our work, we disentangled the detailed changes of innate immune response in BMDMs during early stage of infection by *C. albicans* or *C. auris* (3 or 6 h following fungal challenge). All *C. auris* strains tested failed to elicit meaningful proinflammatory response in BMDMs (MOI=5; 3 or 6 h), as evidenced by unaltered p38 MAPK phosphorylation (**Supplementary Fig. 8b**) and decreased expression and secretion of cytokines/chemokines (**Fig. 4e, f**). Interestingly, when we extend the incubation time to 24 h (the time point used by Bruno *et al.*), patterns of cytokine induction by *C. albicans* or *C. auris* remain largely unchanged (**Fig. 4g**), suggesting that the influence of incubation time could be minor. In contrast, the recognition of *C. albicans* by BMDMs results in the release of inflammatory and chemotactic cytokines, aiding further immune cell recruitment and infection resolution. (**Lines 351-358 on page 19**)

- 5) **Growth conditions and experimental protocols.** As described by Bruno *et al.*, both *C. albicans* and *C. auris* strains were prepared by growing cells for 24 h in Sabouraud medium for 30 °C. In our experiments, the saturated overnight culture of indicated yeast strain was diluted and incubated to logarithmic growth phase at 30 °C. All strains were routinely grown in YPD medium. It is possibly that growth condition will affect the structure of PAMPs. Moreover, the discrepancies in results could be due to the modified experimental protocols used in the two studies. (**Lines 694-699 on page 35**)

In summary, only a few studies thus far have compared the innate immune responses following the challenge with *C. albicans* and *C. auris*. More evidence is needed to draw firm conclusions due to differences in the cell types, strains, methodologies employed in these studies.

References:

Bruno, M. *et al.* Transcriptional and functional insights into the host immune response against the emerging fungal pathogen *Candida auris*. *Nat Microbiol* **5**, 1516-1531 (2020)

Yadav, B. *et al.* Differences in fungal immune recognition by monocytes and macrophages: N-mannan can be a shield or activator of immune recognition. *Cell Surf* **6**, 100042 (2020).

Q2. *Throughout the manuscript, authors used “innate immune evasion by C. auris”, but did provide direct experimental data to demonstrate that C. auris is poorly recognized by immune cells. Their data in Supplemental Fig. 2d showed phagocytosis % of BJCA001 and SC5314 by BMDMs in 1 hr are about 20% and 30%. This is similar to Fig. 2 in Bruno et al. where dynamics of C. auris phagocytosis by macrophage is extensively investigated. Bruno et al. conclude that the difference between C. auris and C. albicans is not at initial uptake.*

Answer: We thank the reviewer for raising this crucial point. Previous studies have shown that phagocytic cells such as macrophages, are being increasingly appreciated as having important roles in recognizing and eliminating the microbial invaders. To further reinforce our proposition that *C. auris* may effectively escape from innate immune recognition and killing, we implemented different approaches to verify if *C. auris* is weakly recognized by immune cells. By doing these, we provide sufficient *ex vivo* results that all four innate immune cell subsets examined, including human and mouse

macrophages and neutrophils, displayed weak recognition (uptake, phagocytosis and killing) with *C. auris*.

1) Adherence to human epithelia. We compared adherence of both *C. auris* and *C. albicans* to different human epithelia including the human skin keratinocyte cell line (HaCat), colorectal adenocarcinoma cell line (Caco-2), umbilical vein endothelial cell line (HUVEC), adenocarcinoma cancer cell line (A549) and HeLa cells. Using an established *in vitro* adhesion assay, we found in **Fig. 2a-e** that in contrast to a remarkable capacity of *C. albicans* to adhere to multiple human cell lines, there is a striking reduction of adhesion activity in *C. auris*. **(Lines 210-217 on page 12)**

2) Percentage uptake and phagocytic index. We investigated host-pathogen interactions by examining the abilities of different immune cells, including macrophages and neutrophils from humans and mice, to phagocytose and kill *C. auris* yeast cells *in vitro*.

First, *C. albicans* or *C. auris* cells were co-cultured over time with mouse BMDMs and primary human monocyte-derived macrophages (MDMs), respectively, and initial recognition by macrophages was investigated by live video microscopy **(Supplementary Video 1, 2, 3, 4)**. Results were expressed as percentage uptake (the percentage of macrophages that engulfed at least one fungal cell) and phagocytic index (the number of fungal cells taken up per 100 macrophages). In both human and mouse macrophages, *C. auris* showed reduced rates of uptake compared to *C. albicans* **(Fig. 2f, g, Supplementary Fig. 4b, c)** and the phagocytic index of *C. auris* was also less than that of *C. albicans* after 30 and 60 min of coculture **(Fig. 2f, h, Supplementary Fig. 4b, d)**. **(Lines 221-233 on pages 12-13)**

Second, we investigated the interaction of murine and human neutrophils with *C. albicans* or *C. auris* and also found differences in the host-pathogen interaction (**Supplementary Video 5, 6, 7, 8**). Compared to *C. albicans*, *C. auris* appeared to be ignored, taken up and internalized by neutrophils at lower rates (**Fig. 2i-k, Supplementary Fig. 4e-g**). (Lines 233-237 on page 13)

Third, we also used fluorescent microscopy technique to further analyze the interactions between macrophages and *C. auris*. The results proved that compared to *C. albicans*, *C. auris* caused a small, but significant fall in percentage of uptake (**Supplementary Fig. 4j, k**) and phagocytic index (**Supplementary Fig. 4l**), as well as less damage to the phagocytes like BMDMs (**Fig. 2n**). (Lines 241-245 on page 13)

- 3) **Fungal killing.** We found that the weaker interaction between *C. auris* and innate immune cells correlated with stronger resistance to intracellular killing by phagocytic cells (**Fig. 2l, m, Supplementary Fig. 4h, i**). We obtained very similar results by testing the survival of *C. auris* inside various innate immune cells, including murine BMDMs, human MDMs, mouse bone marrow neutrophils and human neutrophils. (Lines 238-240 on page 13)
- 4) Consistently, Bruno *et al.* also found that for mannans from the different *C. auris* isolates, their overall binding affinities to mannan recognition receptors (e.g., Dectin-2 and mannose receptors) were more than an order of magnitude lower than those that were observed for *C. albicans* mannans.
- 5) Johnson *et al.* found that neutrophils exert a strong antifungal response to *C. albicans* whereas these immune cells failed to engage and phagocytose the yeast cells and form extracellular traps when interacting with *C. auris*.

Taken together, our cell culture-based *in vitro* experiments provide a list of direct evidence supporting that unlike *C. albicans*, *C. auris* appears to be weakly taken up and eliminated by host innate immune system, at least in macrophages and neutrophils. *C. auris* may effectively avoid being engaged and phagocytosed during the course of infection and host defense.

References:

Erwig, L.P. & Gow, N.A. Interactions of fungal pathogens with phagocytes. *Nat Rev Microbiol* **14**, 163-76 (2016)

Johnson, C.J., Davis, J.M., Huttenlocher, A., Kernien, J.F. & Nett, J.E. Emerging Fungal Pathogen *Candida auris* Evades Neutrophil Attack. *mBio* **9**(2018).

Q3. *The difference in phagocytosis between A. auris BJCA001 and C. albicans SC5314 (20% vs. 30%) cannot explain the dramatic differences in cytokines induced by A. auris and C. albicans observed in this study. Also, their cytokine results are very different from that published by Bruno et al. In addition to differences in MOI and types of myeloid cells used, another major difference in exposure time. This study used 3 hrs. Bruno et al. used 24 hrs. Growth of A. auris inside myeloid cells was suggested to contribute to the strong cytokine production induced by A. auris.*

Answer: We appreciate the reviewer's suggestions. Based on our answers to Q2, we have made it clear that unlike *C. albicans*, *C. auris* appears to be poorly taken up and eliminated by host innate immune system. This conclusion turns out to be true because we obtained reproducible results using four different types of immune cells. The reduced recognition by macrophages and neutrophils correlated with much weaker cytokine responses. It is noteworthy that we normally used a MOI of 5 to measure the expression (3 h after co-incubation) and secretion (6 h after co-incubation) of proinflammatory

cytokines and chemokines during the host-pathogen interaction, whereas Bruno *et al.* carried out the same experiments using different setting (MOI=0.1; 24 h). Longer incubation periods (e.g., 24 h) were not considered in our study, since we observed robust cell death in BMDMs 24 h after co-incubation (**Supplementary Fig. 2**). After a long incubation time (MOI=5, 24h), we found that most BMDMs were dead and there is absolutely no way to have enough RNA for RNA-seq analysis. **(Lines 145-147 on page 9)**

We did consider the possibility that the difference in exposure time may account for the discrepancy of the two studies. To test this, we examined the production of the cytokines and chemokines by ELISA after BMDMs were stimulated with *C. albicans* or *C. auris* at a MOI of 0.1 for 24 h. Similar to the results with higher doses of the inoculum (MOI=5) and shorter time in stimulation (6 h), we still found that all *C. auris* strains tested failed to elicit meaningful proinflammatory responses (**Fig. 4f, g**), suggesting that the influence of incubation time could be minor. **(Lines 351-358 on page 19)**

This observation was further validated in another independent experiment. Here, we investigated the cytokine production in BMDMs after co-incubation with indicated strains at a MOI of 5 for 24 h. In contrast to *C. albicans*, *C. auris* WT failed to elicit secretion of cytokines in BMDMs with lower doses of the inoculum (MOI=5) and longer incubation time (24 h), however, *C. auris* mutants with impaired cell wall mannosylation were able to effectively trigger innate immune activation.

Q4. *The in vivo cytokine and CFU data are also very different between two studies. This is surprising consider that both studies use the same mouse model and C57BL/B6 mice. Bruno et al. injected 10^6 C. auris or C. albicans per mouse. This study used different inoculums (10^7 and 10^6 for C. auris, and 5×10^4 for C. albicans) in all in vivo experiments in this study.*

Answer: We very much appreciate the careful reading of our manuscript and valuable suggestions of the reviewer. It is true that different inoculum doses of *C. albicans* (5×10^4 CFU/mice) and *C. auris* (1×10^6 CFU/mice) were used in our study, largely because of the fact that the mice inoculated with *C. albicans* at 1×10^6 CFU experience a clinical illness manifested by ruffled fur, reduced activity and weight loss, and died rapidly within 5 days (**Supplementary Fig. 5**). In comparison, those receiving a lower inoculum (5×10^4 CFU/mice) could live long and healthy during the period of innate immune response (usually within the first 7 days). (**Lines 255-259 on page 14**)

We also compared the serum levels of cytokines/chemokines in mice infected with the same doses of *C. albicans* or *C. auris* (1×10^6 CFU/mice). It is noteworthy that the mice exhibit an overall unhealthy status on day 2 when the inoculum size of *C. albicans* SC5314 was increased to the same level as for *C. auris* (1×10^6 CFU/mice). In terms of tissue fungal burden, there was no significant difference between *C. albicans* and *C. auris* (**Supplementary Fig. 6d**). However, we noticed that serum levels of cytokines/chemokines were significantly higher in *C. albicans*-infected mice than in *C. auris*-inoculated mice (**Supplementary Fig. 6e**) (the data on day 5 were unavailable because most mice were found dead), minimizing the possible effect of inoculum size on innate immune activation. (**Lines 301-307 on page 16**)

Q5. *b-glucan is exposed in the pmt1 mutant of both C. auris and C. albicans. It does not suggest that C. auris uses a novel mechanism to evade host immune detection. In fact, the level of b-glucan exposure is similar in A. auris BJCA001*

and *C. albicans* SC5314 in this study (Fig. 5e). This data does not support authors' conclusions.

Answer: We apologize for the lack of clarity. We totally agree with the reviewer's comment that the innate immune evasion strategy employed by *C. auris* is not unique and novel, and actually has been identified in the other human fungal pathogens like *C. albicans*. For example, Graus *et al.* found that *N*-mannan structural features regulated by *Candida* mannosyltransferases control glucan exposure. Loss of mannan increased the frequency and size of glucan exposures and changed multivalent receptor engagement. Moreover, previous studies by Netea *et al* have indicated that *C. albicans* masks underlying β -(1,3)-glucan with a dense layer of mannan and/or mannoprotein and protects the pathogen against host immune system. We have rephrased the description about the cell wall-based innate immune evasion strategy in our revised manuscript by removing the words like "unique" and "novel". We proposed an effective innate immune evasion strategy employed by the emerging fungal pathogen *C. auris*. Compared to *C. albicans*, this fungus behaves as a silent invader by exposing a structurally different outer layer of cell wall with distinct properties and characteristics related to the mannosylated glycoproteins. Our studies strongly indicate that the outer mannan layer acts to mask the inner layer of β -glucan from exposure and detection by innate immune cells, and therefore plays a key role in protecting *C. auris* against the host innate immune clearance.

In our original submission, we stained the exponentially growing fungal cells (*C. albicans* SC5314 and *C. auris* BJCA001) with β -glucan antibody to visualize β -(1,3)-glucan. Based on the comments from the reviewer #3, β -glucan antibody is known to have problem and Fc-Dectin-1 is a much more reliable way to quantify glucan exposure. We therefore re-examined the exposure of cell wall glucan in all *C. auris* strains, using Fc-Dectin-1 for β -1,3-glucan

staining. As shown in **Fig. 5f, Fig. 6d and j, supplementary Fig. 11a**, we could observe high quality images showing β -1,3-glucan staining. The changes of cell wall polysaccharide fractions in *C. albicans* and *C. auris* were confirmed by both fluorescent staining and flow cytometry (**Fig. 5f, Supplementary Fig. 11a**). When compared to what we observed in *C. albicans*, the proportion of the outer mannan layer was significantly increased in *C. auris* and this pattern appears to be conserved across most *C. auris* strains. This conclusion was further verified by other approaches, including 1) the methods for high-pressure freezing (HPF) and freeze substitution (FS) to examine the ultrastructure of *C. auris* BJCA001 or *C. albicans* SC5314 yeast cell wall by transmission electron microscopy (TEM); 2) gas chromatography-mass spectrometry (GC-MS) and high-performance liquid chromatography (HPLC) analyses for sugar composition; and 3) fluorescent staining and flow cytometry. **(Lines 395-441 on pages 21-23)**

References:

Graus, M.S. *et al.* Mannan Molecular Substructures Control Nanoscale Glucan Exposure in Candida. *Cell Rep* **24**, 2432-2442 e5 (2018).

Netea, M.G., Joosten, L.A., van der Meer, J.W., Kullberg, B.J. & van de Veerdonk, F.L. Immune defence against Candida fungal infections. *Nat Rev Immunol* **15**, 630-42 (2015)

Netea, M.G., Brown, G.D., Kullberg, B.J. & Gow, N.A. An integrated model of the recognition of *Candida albicans* by the innate immune system.

Netea, M.G. *et al.* Immune sensing of *Candida albicans* requires cooperative recognition of mannans and glucans by lectin and Toll-like receptors. *J Clin Invest* **116**, 1642-50 (2006).

Reviewer #2:

1. Major Comments

Q1. *The authors are unfortunate for two reasons: (a) a parallel study was published recently in Nature Microbiology by Bruno et al., and (b) this new study appears, at least superficially, to contradict some findings reported by Bruno et al. Consequently, the novelty of some findings in this new paper has been compromised, and interesting questions have arisen about the underlying reasons for the differences between these two studies.*

Answer: We are grateful to the reviewer for this constructive suggestion. Indeed, Bruno *et al* published a very similar paper in *Nature Microbiology*, with conclusions that contradict some of our findings. They also evaluated the host immune response to a list of *C. auris* clinical isolates from five clades, claimed that *C. auris* appears to induce a stronger innate immune response compared to *C. albicans*. On the contrary, our *in vitro* and *in vivo* studies indicate that compared to *C. albicans*, *C. auris* acts as an innate immune silencer, rather than an inducer. In this revised manuscript, we carried out a number of new experiments and provided strong evidence supporting our claim that compared to *C. albicans*, *C. auris* behaves as a silent invader and evasion of innate immunity is mainly mediated by a thick and structurally distinct mannan layer of cell wall, which masks β -glucan exposure and avoids innate immune clearance.

1) *In vitro* evidence

- a. We observed in murine BMDMs that upon *C. albicans* stimulation, the expression and secretion levels of proinflammatory cytokines and chemokines, including IL-1 β , IL-6, TNF- α , CXCL1 and CXCL2, were significantly increased, however, the induction was dramatically diminished

by *C. auris* infection. Both RNA-seq and Western analyses further confirmed the role of *C. auris* in innate immune suppression; **(Lines 141-188 on pages 8-11)**

- b. In contrast to a remarkable capacity of *C. albicans* to adhere to multiple human cell lines, including the human skin keratinocyte cell line (HaCat), colorectal adenocarcinoma cell line (Caco-2), umbilical vein endothelial cell line (HUVEC), adenocarcinoma cancer cell line (A549) and HeLa cells, there is a striking reduction of adhesion activity in *C. auris*. **(Lines 210-217 on page 12)**
- c. *C. auris* may avoid being engaged and phagocytosed during the course of infection and host defense. We proposed this conclusion by implementing different approaches (e.g., percentage uptake and phagocytic index) to verify that *C. auris* is poorly recognized by various immune cells, including macrophages and neutrophils from humans and mice; **(Lines 221-237 on page 12)**
- d. All *C. auris* strains tested (11 clinical isolates belonging to clade I and II) failed to elicit meaningful proinflammatory response in BMDMs, as evidenced by unaltered p38 MAPK phosphorylation and decreased expression and secretion of cytokines/chemokines; **(Lines 351-358 on page 19)**
- e. Even in human PBMCs, both *C. albicans* and *C. auris* could effectively induce cytokine production in human PBMCs, when compared to the untreated control. Some, but not all, of the *C. auris* strains elicited higher production of cytokines than did *C. albicans*. In most cases, we observed the same pattern of increased innate immune responsiveness to *C. auris* or *C. albicans*; **(Lines 359-378 on pages 19-20)**

- f. Upon contact with BMDMs, *C. auris* cell wall still displayed increase in relative proportion of mannan followed by a decrease in β -glucan levels when compared to that of *C. albicans*; **(Lines 423-425 on page 22)**
- g. Blockade of Mincle or DC-SIGN in BMDMs, by using specific antibodies, resulted in unexpected cytokine responses seen either increasing or decreasing expression of cytokines in response to *C. albicans*. In comparison, induction of certain cytokines was also observed in *C. auris*-infected BMDMs that were preincubated with receptor-blocking antibodies, highlighting the multiplicity of these fungal cell wall-macrophage receptor interactions. Blocking a certain receptor may somehow disturb the dynamic balance between host and pathogen, leading to varied immune responses; **(Lines 467-477 on pages 24-25)**
- h. BMDM uptake of the *C. auris* *pmr1* Δ or *pmt1* Δ mutant was significantly increased compared to the parent wild-type strain and its high uptake rate correlates with a remarkably decrease in the number of viable yeast cells after 24 h incubation. And this reflects a strong activation of p38 MAPK in BMDMs and significant induction in the expression and secretion of proinflammatory cytokines/chemokines; **(Lines 511-518 on pages 26-27; Lines 530-534 on page 27)**
- i. The production of cytokines (TNF α) by BMDMs was significantly induced by the challenge with Tetoff-*OCH1* mutant of *C. auris*, suggesting that protein *N*-glycosylation in the outer layer of cell wall also plays a key in suppressing the innate immune response. **(Lines 548-552 on page 28)**

2) *in vivo* evidence

- a. After infection, *C. auris* cells were able to persist in the host and avoid to be recognized and cleared by the host innate immune system; **(Lines 104-130 on pages 7-8; Lines 261-270 on pages 14-15)**
- b. Stimulation with *C. albicans* leads to a sizeable increase in the number of neutrophils in mice. However, the mice challenged with *C. auris* showed significantly reduced neutrophil recruitment. These results suggest that *C. auris* infection resulted in less neutrophil egress than did *C. albicans*; **(Lines 271-283 on page 15)**
- c. The relative abundance of an array of immune cell populations (e.g., leukocytes, CD11b⁺ cells, macrophages, monocytes, NK cells, B cells, T cells) was also quantified by flow cytometry. Importantly, distribution of innate immune cells (e.g., leukocytes, CD11b⁺ cells, macrophages, monocytes and NK cells) mirrored the patterns described for neutrophils, being particularly present in significantly higher numbers in *C. albicans*-infected tissues versus *C. auris*-infected tissues; **(Lines 284-294 on pages 15-16)**
- d. A marked induction of cytokines (IL-1 β , IL-6, TNF- α) and chemokines (CXCL1 and CXCL2) was recorded in the mice infected with *C. albicans*, however, their levels in the mice challenged with either a high or low dosage of *C. auris* were almost identical to those of uninfected control mice. **(Lines 294-303 on page 16)**

Taken together, our *in vitro* and *in vivo* results should be sufficient to support the conclusion that the outer mannan layer acts to mask the inner layer of β -glucan from exposure and detection by innate immune cells, and therefore plays a key role in protecting *C. auris* against the host innate immune clearance.

Q2. *Is this because each study used different C. auris isolates, different immune cell types (PBMCs versus BMDMs), different MOIs, different experimental time points, and/or different fungal growth/fixing protocols? All of these factors are known to influence immunological outputs. For example, mannan is not an immunostimulatory PAMP for macrophages but is for monocytes, and fungal PAMP exposure is known to change with growth phase.*

Answer: We very much appreciate the careful reading of our manuscript and valuable suggestions of the reviewer. Actually the reviewer #1 asked the same question. We thoroughly analyzed the possible source of discrepancies in these two independent studies and performed a number of new experiments to find valid explanations for the disagreement, which are summarized below.

1) Cell types. For our work, we compiled murine bone marrow macrophages (BMDMs) datasets to discover potentially different effects of *C. albicans* and *C. auris* on host innate immune activation, while Bruno *et al.* mainly used human PBMCs for tackling similar questions. A recent study by Yadav *et al.* provided us a seemingly rational explanation. This discrepancy might be due to the recognition difference, as the authors claimed that during *Candida*-macrophage interaction, *N*-mannan at the surface of *Candida* cell wall acts to mask exposure of β -glucan, allowing fungal cells to evade innate immunity by restricting the access of dectin-1 to β -glucan. In contrast, *N*-mannan behaves as a major PAMP to induce cytokine response from human PBMCs. Indeed, our results confirmed that during its interactions with macrophages or neutrophils from human and mouse, the mannan layer of *C. auris* cell wall is tightly associated with an immunosuppressive activity masking cell wall β -glucan from recognition by Dectin-1. We found that *C. auris* cell wall displayed increase in relative proportion of mannan followed by a decrease in β -glucan levels when compared to that of *C.*

albicans (Different techniques were used in our assays, including TEM, GC-MS, HPLC, ¹H NMR, Fluorescent staining and Flow cytometry) (**Fig. 5; Supplementary Fig. 11**). Strikingly, disruption of the outer mannan layer by knocking out *PMR1* or *PMT1* in *C. auris* leads to the unmasking of β -(1-3)-glucan and induction of innate immune responses (**Fig. 6; Supplementary Fig. 14 and 15**). Using the Tetoff-*OCH1* mutant of *C. auris*, we further demonstrate that the protein *N*-glycosylation in the outer layer of cell wall plays a key in suppressing the innate immune response and the removal of *N*-mannan in *C. auris* caused β (1,3)-glucan exposure (unmasking), leading to increased β -glucan receptor (dectin-1)-dependent elicitation of key proinflammatory cytokines by macrophages like BMDMs (**Supplementary Fig. 16**). Our results thus strongly suggest a highly conserved innate immune evasion strategy that *C. auris* employs its distinct mannan-protein layer of cell wall to withstand host innate immune responses, allowing for the successful host invasion. (**Lines 395-441 on pages 21-23; Lines 498-536 on pages 26-27; Lines 537-556 on page 28**)

To make sure whether we could reproduce the results, we performed ELISA assays using all 11 *C. auris* strains in our collection and examined cytokine production in human PBMCs after challenge at MOI 5 and 0.1 for 6 h and 24h, respectively (Note: Bruno et al. did the same assay using a low MOI of 0.1 and a 24-h coincubation). Our results did show some differences (**Supplementary Fig. 9a, b**). Unlike what we observed in macrophages, both *C. albicans* and *C. auris* could effectively induce cytokine production in human PBMCs, when compared to the untreated control. We did find much higher cytokine production with *C. auris*-stimulated cells compared to cells challenged with *C. albicans*, but this induction pattern took place in only a few tests. For example, all *C. auris* strains tested showed higher levels of TNF α only in the experiment

using a high MOI (MOI=5). Some, but not all of the *C. auris* strains elicited higher production of cytokines than did *C. albicans*. In most cases, we observed the same pattern of increased innate immune responsiveness to *C. auris* or *C. albicans*. Our results demonstrate differential innate immune responses to *C. auris* seen in different immune cell types, arguing that the exact role of different phagocytes in the context of *C. auris* infection awaits further precise and comprehensive investigations for clarification. Given that human PBMCs are a mixture of immune cells and made up of lymphocytes (B cells, T cells and NK cells), monocytes and dendritic cells, it will be imperative to find out the actual cell type that specifically and differentially interacts with *C. auris* and *C. albicans*. **(Lines 359-378 on pages 19-20)**

- 2) **Fungal strains.** The standard *C. albicans* laboratory strain SC5314 was used throughout our study while three different *C. albicans* strains, including SC5314, CWZ10061110 and UC820, were selected by Bruno *et al.* Of course, the clinical isolates of *C. auris* are totally different in the two studies. As for *C. auris* strains, Bruno *et al.* used isolates covering all five clades and we used 11 clinical isolates belonging to clade I and II. Therefore, the impact of differences of strain background, which may vary among PAMPs, could not be excluded. Indeed, we found from the work by Bruno *et al.* that one isolate (*C. auris* 10051893, clade I) behaves differently from the others and shows lower phagocytic index during the coculture with BMDMs. Moreover, stimulation with the *C. auris* isolate 10111018 (clade V) significantly decreased the production of cytokines (IL-1b and IL-1RA) from BMDMs. In this case, their results somehow favor our model.
- 3) **MOIs.** Different MOIs were applied for cell culture-based assays in the two studies. For example, we used a MOI of 1 for assessment of percentage

uptake and phagocytic index, and a MOI of 5 for all cytokine production assays. However, Bruno et al. used a MOI of 3 for phagocytosis assays and a MOI of 0.1 for cytokine production assay. But, we don't think that the disparity between the two studies mainly stems from the influence of different MOIs, as we found that the failure of *C. auris* to elicit innate immune responses in the macrophages and neutrophils of human and mouse is highly conserved in all tested isolates and independent of MOIs **(Fig. 4g). (Lines 355-358 on page 19)**

Another factor considered to be important could be the inoculum size. It has to be noted that different inoculum doses of *C. albicans* (5×10^4 CFU/mice) and *C. auris* (1×10^6 CFU/mice) were used in our *in vivo* animal studies, largely because of the fact that the mice inoculated with *C. albicans* at 1×10^6 CFU experience a clinical illness manifested by ruffled fur, reduced activity and weight loss, and died rapidly within 5 days **(Supplementary Fig. 5)**. In comparison, those receiving a lower inoculum (5×10^4 CFU/mice) could live long and healthy during the period of innate immune response (usually within the first 7 days). We did perform the experiment by inoculating the mice with the same doses of *C. albicans* or *C. auris* (1×10^6 CFU/mice). As expected, the mice exhibit an overall unhealthy status on day 2 when the inoculum size of *C. albicans* was increased to the same level as for *C. auris*. In terms of tissue fungal burden, there was no significant difference between *C. albicans* and *C. auris* **(Supplementary Fig. 6d)**. However, we still noticed that serum levels of cytokines/chemokines were significantly higher in *C. albicans*-infected mice than in *C. auris*-inoculated mice **(Supplementary Fig. 6e)** (the data on day 5 were unavailable because most mice were found dead), minimizing the possible effect of inoculum size on innate immune activation. **(Lines 255-261 on page 14; Lines 299-307 on page 16)**

4) Experimental time points. In our work, we disentangled the detailed changes of innate immune response in BMDMs during early stage of infection by *C. albicans* or *C. auris* (3 or 6 h following fungal challenge). All *C. auris* strains tested failed to elicit meaningful proinflammatory response in BMDMs (MOI=5; 3 or 6 h), as evidenced by unaltered p38 MAPK phosphorylation (**Supplementary Fig. 8b**) and decreased expression and secretion of cytokines/chemokines (**Fig. 4e, f**). Interestingly, when we extend the incubation time to 24 h (the time point used by Bruno *et al.*), patterns of cytokine induction by *C. albicans* or *C. auris* remain largely unchanged (**Fig. 4g**), suggesting that the influence of incubation time could be minor. In contrast, the recognition of *C. albicans* by BMDMs results in the release of inflammatory and chemotactic cytokines, aiding further immune cell recruitment and infection resolution. **(Lines 351-358 on page 19)**

5) Growth conditions and experimental protocols. As described by Bruno *et al.*, both *C. albicans* and *C. auris* strains were prepared by growing cells for 24 h in Sabouraud medium for 30 °C. In our experiments, the saturated overnight culture of indicated yeast strain was diluted and incubated to logarithmic growth phase at 30 °C. All strains were routinely grown in YPD medium. It is possibly that growth condition will affect the structure of PAMPs. Moreover, the discrepancies in results could be due to the modified experimental protocols used in the two studies. **(Lines 694-699 on page 35)**

In summary, only a few studies thus far have compared the innate immune responses following the challenge with *C. albicans* and *C. auris*. More evidence is needed to draw firm conclusions due to differences in the cell types, strains, methodologies employed in these studies.

References:

Bruno, M. *et al.* Transcriptional and functional insights into the host immune response against the emerging fungal pathogen *Candida auris*. *Nat Microbiol* **5**, 1516-1531 (2020)

Yadav, B. *et al.* Differences in fungal immune recognition by monocytes and macrophages: N-mannan can be a shield or activator of immune recognition. *Cell Surf* **6**, 100042 (2020).

Q3. *In the end, the differences between these studies are likely to prove highly informative, but this will require a significant amount of additional work. An alternative strategy might be to shift the focus of this interesting story by placing a more positive spin on the differences between the two studies, and by placing more emphasis on the interesting cell wall phenotypes of the C. auris pmr1 and pmt1 mutants that are currently buried in the supplementary data. It might also be advisable to downplay the idea that C. auris evolved a “unique immune evasion strategy” because this species only emerged recently as a fungal pathogen of humans (where did it evolve this strategy?) and because mannan is known to shield β -glucan in other pathogenic Candida species.*

Answer: We thank the reviewer for these pertinent suggestions. According to the reviewer’s advices, we re-organized the **Fig. 6** by adding the results showing cell wall changes in both *pmr1* Δ and *pmt1* Δ mutant strains. For both mutants, we examined the macrophage uptake and fungal survival inside the host, and the results were also added to **Fig. 6**. In addition, inhibitory expression of *OCH1* in the Tetoff-*OCH1* mutant of *C. auris* significantly stimulated the production of cytokines (TNF α) by BMDMs (**Supplementary 16**), hinting that the removal of N-mannan in *C. auris* caused β -glucan exposure (unmasking), leading to increased β -glucan receptor (dectin-1)-dependent elicitation of key proinflammatory cytokines by macrophages like BMDMs. **(Lines 548-552 on page 28)**

We totally agree with the reviewer's comment that the innate immune evasion strategy employed by *C. auris* is not unique and novel, and actually has been identified in the other human fungal pathogens like *C. albicans*. We have rephrased the description about the cell wall-based innate immune evasion strategy in our revised manuscript by removing the words like "unique" and "novel".

Reviewer #3: (Remarks to the Author)

1. Major Comments

Q1. *When analysing immune cells by FACS why did you only include neutrophils, one of the powers of FACs is to be able to identify multiple cell populations from a single sample.*

Answer: We thank the reviewer for the valuable comments. We performed the experiments according to the reviewer's suggestion. The relative abundance of an array of immune cell populations (e.g., leukocytes, CD11b⁺ cells, macrophages, monocytes, NK cells, B cells, T cells) was also quantified by flow cytometry. Consistently, we found that distribution of innate immune cells (e.g., leukocytes, CD11b⁺ cells, macrophages, monocytes and NK cells) mirrored the patterns described for neutrophils, being particularly present in significantly higher numbers in *C. albicans*-infected tissues versus *C. auris*-infected tissues (**Fig. 3d, e and Supplementary Fig. 6b, c**). As expected, no significant changes were observed for the population of B and T cells, which normally accumulate during an adaptive immune response. Our studies show starkly different responses by innate immune cells to *C. albicans* and *C. auris*, with the latter showing poor recognition and innate immune evasion. **(Lines 284-290 on page 16)**

Q2. *From your histology what morphology is C. auris is, and does this vary in the different niches?*

Answer: We thank the reviewer for this great question. Sections of different organs, including the kidney, spleen and brain, were stained using the Periodic acid-Schiff (PAS) method. Our histopathologic examination confirmed abundant tissue colonization of aggregate yeast cells with no pseudohyphae and filaments in different niches of *C. auris*-infected mice (**Supplementary Fig. 1d-f**). **(Lines 123-125 on page 8)**

Q3. Isn't *C. auris* more of a skin coloniser than a gut coloniser could this not explain what it is less able to adhere to gut cells, did you look at skin cells/models of infection?

Answer: A very interesting question. A recent paper by Huang et al. found that *C. auris* is able to establish long-term residence within the skin tissue compartment, and interestingly, *C. auris* colonizes skin surface and tissues but not the gut. In the revised manuscript, we compared adherence of both *C. auris* and *C. albicans* to different human epithelia. Using an established *in vitro* adhesion assay, we found that in contrast to a remarkable capacity of *C. albicans* to adhere to multiple human cell lines, including the human skin keratinocyte cell line (HaCat), colorectal adenocarcinoma cell line (Caco-2), umbilical vein endothelial cell line (HUVEC), adenocarcinoma cancer cell line (A549) and HeLa cells, there is a striking reduction of adhesion activity in *C. auris* (**Fig. 2a-e**). Interestingly, we observed no statistically significant difference in *C. auris* adhesion between skin-derived HaCat and gut-driven Caco-2 (**Supplementary Fig. 4a**). The weak *C. auris* adherence to human skin cells like HaCat cell line suggests that its persistent colonization on skin surface might rely on other factors (e.g., invasion and damage), rather than adhesion. Of course, this possibility awaits more comprehensive investigations. **(Lines 221-248 on pages 12-14)**

Reference:

Huang, X. *et al.* Murine model of colonization with fungal pathogen *Candida auris* to explore skin tropism, host risk factors and therapeutic strategies. *Cell Host Microbe* **29**, 210-221 e6 (2021)

Q4. Why use BMDMs, why not quantify phagocytosis/killing in human monocyte derived macrophages, as these would be more relevant, as would the inclusion of human neutrophils.

Answer: According to the reviewer's suggestions, we investigated host-pathogen interactions by examining the abilities of different immune cells, including macrophages and neutrophils from humans and mice, to phagocytose and kill *C. auris* yeast cells *in vitro*.

First, *C. albicans* or *C. auris* cells were co-cultured over time with mouse BMDMs and primary human monocyte-derived macrophages (MDMs), respectively, and initial recognition by macrophages was investigated by live video microscopy (**Supplementary Videos 1, 2, 3, 4**). Results were expressed as percentage uptake (the percentage of macrophages that engulfed at least one fungal cell) and phagocytic index (the number of fungal cells taken up per 100 macrophages). In both human and mouse macrophages, *C. auris* showed significantly reduced rates of uptake compared to *C. albicans* (**Fig. 2f, g, Supplementary Fig. 4b, c**) and the phagocytic index of *C. auris* was also significantly less than that of *C. albicans* after 30 and 60 min of coculture (**Fig. 2f, h, Supplementary Fig. 4b, d**). **(Lines 221-233 on pages 12-13)**

Second, we investigated the interaction of murine and human neutrophils with *C. albicans* or *C. auris* and also found differences in the host-pathogen interaction (**Supplementary Videos 5, 6, 7, 8**). Compared to *C. albicans*, *C. auris* appeared to be ignored, taken up and internalized by neutrophils at lower rates (**Fig. 2i-k, Supplementary Fig. 4e-g**). Importantly, the weaker interaction between *C. auris* and innate immune cells correlated with stronger resistance to intracellular killing by phagocytic cells (**Fig. 2l, m, Supplementary Fig. 4h, i**). **(Lines 233-237 on page 13)**

Third, the analysis using the fluorescent microscopy technique further proved that compared to *C. albicans*, *C. auris* caused a small but significant fall in percentage of uptake (**Supplementary Fig. 4j, k**) and phagocytic index

(Supplementary Fig. 4I), as well as less damage to the phagocytes like BMDMs (Fig. 2n). (Lines 241-245 on page 13)

Taken together, our results indicate that *C. auris* may avoid being engaged and phagocytosed during the course of infection and host defense.

Q5. *In some cases a student's T-test is used to analyse data with greater than two samples sizes, please repeat statistical analysis with more appropriate methodology that accounts for multiple comparisons, and potentially non-normally distributed data.*

Answer: We really appreciate the reviewer's valuable comments. We re-analyzed our data using more appropriate statistical methods, described by Ost *et al* in a recent *Nature* paper. Key points of these methods were described below.

We use one-way or two-way ANOVA method to analyze the data having more than two sample sizes. However, an ANOVA test has limitations because it cannot tell which pairs of means are different. We therefore perform an additional analysis, namely multiple comparison test (MCT), to clarify the differences between specific pairs of our experimental samples. There are many methods for MCT analysis, including Tukey method, Bonferroni method, Dunnett method, Sidak method.

If the samples have one factor and the size greater than two, we use one-way ANOVA with some MCT methods; When compare mean of each sample with the mean of every other sample, Tukey method is recommended by the Graphpad Prism software; When compare the mean of sample with the mean of a control sample, Dunnett method is recommended by the Graphpad Prism software; When compare the means of preselected pairs of samples, Sidak method is recommended the Graphpad Prism.

References:

Ost KS, O'Meara TR, Stephens WZ, Chiaro T, Zhou H, Penman J, Bell R, Catanzaro JR, Song D, Singh S, Call DH, Hwang-Wong E, Hanson KE, Valentine JF, Christensen KA, O'Connell RM, Cormack B, Ibrahim AS, Palm NW, Noble SM, Round JL. Adaptive immunity induces mutualism between commensal eukaryotes. *Nature*. 2021 Aug;596(7870):114-118.

Lee S, Lee DK. What is the proper way to apply the multiple comparison test? *Korean J Anesthesiol*. 2018 Oct;71(5):353-360.

Q6. *HLPC of the cell wall dry weight would enable you to quantify the relative % of mannan glucan and chitin in the cell wall, and could also identify alternative PAMPs not present in C. albicans.*

Answer: We thank the reviewer for such a valuable suggestion. We quantified the exposure levels of the three major cell wall polysaccharides (mannan, glucan and chitin) at the surface of *C. albicans* and *C. auris*, using both gas chromatography-mass spectrometry (GC-MS) and high-performance liquid chromatography (HPLC) techniques. The two analyses gave similar results (**Fig. 5d and 5e**). The major cell wall components of both species are mannan and glucan ($\beta(1,3)$ -glucan and $\beta(1,6)$ -glucan) and a small quantity of chitin. However, structural differences were detected. For example, *C. auris* BJCA001 showed increase in relative proportion of mannan in the cell wall when compared to *C. albicans* SC5314 (40.34% vs 31.42% by GC-MS; 41.40% vs 32.72% by HPLC). The relative proportion of β -glucan in the cell wall was much less in *C. auris* than that in *C. albicans* (55.51% vs 66.43% by GC-MS; 55.68% vs 64.33% by HPLC), presumably reflecting the presence of β -glucan masking in *C. auris*. The content of chitin in both strains are almost the same (4.15% vs 2.14% by GM-MS; 2.91% vs 2.95% by HPLC).

Notably, we also found in HPLC analysis that the HPLC elution profiles of *C. albicans* and *C. auris* strains were identical (**Supplementary Fig. 10**) and no additional distinct peaks were present between the two HPLC chromatograms, suggesting that *C. auris* should have no alternative PAMPs which may differ considerably from *C. albicans*. (**Lines 400-417 on pages 21-22**)

Q6. *C. auris* lacks phosphomannan and the acid labile mannan, which are known to be recognised by specific PRRs on innate immune cells, have you looked into the role of these receptors? If you use the *C. albicans* mutants that are defective in acid labile mannan do you see an attenuation of virulence in your models?

Answer: We really appreciate the reviewer's valuable suggestions. Phosphomannan is found in mannan as the mannosyl-phosphate moiety and studies in *C. albicans* have demonstrated that this negatively charged cell structure is required for phagocytosis by macrophages. Our analysis identified that compared to that of *C. albicans* SC5314, the outer mannan layer of *C. auris* BJCA001 exhibits less complexity with the lack of phosphomannan and acid-labile mannan (**Fig. 5g-I**, by ¹H NMR spectroscopy; **Supplementary Fig. 11c**, by Alcian blue staining). We searched published literatures that may provide any detailed information about the host receptors (PRRs) specifically recognizing the phosphomannan and acid-labile mannan of fungal cell wall. However, the searching did not yield any valuable information and we couldn't identify which receptor is the one that specifically binds to the phosphomannan and acid-labile mannan of fungal cell wall. As such, we are unable to use the direct strategies (e.g., blocking the receptor activities by specific antibodies or inhibitors) to test the idea whether the lack of phosphomannan may account for innate immune evasion by *C. auris*, and alternative ways should be considered. (**Lines 425-436 on pages 22-23**)

First, we decide to focus on several known mannan-binding receptors, including the mannose receptor (MR), Mincle, DC-SIGN, TLR4 and TLR2 and validate the essential role of the outer mannan layer of *C. auris* cell wall in protection and innate immune evasion. *C. albicans* stimulation induced higher expression levels of these receptor-encoding genes in BMDMs than did *C. auris* (**Supplementary Fig. 12a**). Moreover, blockade of Mincle or DC-SIGN in BMDMs, by using specific antibodies, resulted in unexpected and complicated cytokine responses seen either increasing or decreasing expression of cytokines in response to *C. albicans* depending on the type of cytokine and blocking antibody tested (**Supplementary Fig. 12b**). Interestingly, induction of certain cytokines was also observed in *C. auris*-infected BMDMs that were preincubated with receptor-blocking antibodies, highlighting the multiplicity of these fungal cell wall-macrophage receptor interactions. Blocking a certain receptor may somehow disturb the dynamic balance between host and pathogen, leading to varied immune responses. More critical studies are needed to precisely address questions about the relative importance of these receptors, as well as their differential roles in response to different fungal species. (**Lines 462-477 on pages 24-25**)

Second, we used an indirect approach for testing whether the lack of phosphomannan may account for innate immune evasion by *C. auris* by generating a phosphomannan-deficient mutant strain (*mnn4Δ/Δ*) in *C. albicans*. The *mnn4Δ/Δ* mutant strain had a complete loss of phosphomannan (**Supplementary Fig. 11d**) and showed a significant reduction in BMDM uptake (**Supplementary Fig. 11e**). Surprisingly, this mutant was still able to induce at least the same or even higher cytokine production compared to the wild-type (**Supplementary Fig. 11f**). Previous studies using PBMCs and mononuclear cells (MNCs) obtained similar results. Groups of C57BL/6 mice were injected intravenously with wild type and *mnn4Δ/Δ* mutant in a mouse model of disseminated *C. albicans* infection and the results showed that the

mnn4Δ/Δ mutant was equally virulent in mice compared to its wild-type counterpart. An attenuation of virulence was not observed in this mutant (**Supplementary Fig. 11g**). Hence, the removal of phosphomannan in *C. albicans* through deletion of *MNN4*, which mimics its deficiency in *C. auris*, may provide indirect evidence supporting the poor recognition and ingestion of *C. auris* by macrophages. However, the neglected role of this component in cytokine induction and pathogenicity told us that innate immune evasion mechanisms in *C. auris* should be more complicated than simply removing the phosphomannan. (**Lines 442-461 on pages 23-24**)

References:

McKenzie, C.G. et al. Contribution of *Candida albicans* cell wall components to recognition by and escape from murine macrophages. *Infect Immun* 78, 1650-8 (2010).

Lewis, L.E. et al. Stage specific assessment of *Candida albicans* phagocytosis by macrophages identifies cell wall composition and morphogenesis as key determinants. *PLoS Pathog* 8, e1002578 (2012).

Gonzalez-Hernandez, R.J. et al. Phosphomannosylation and the Functional Analysis of the Extended *Candida albicans* *MNN4*-Like Gene Family. *Front Microbiol* 8, 2156 (2017).

Netea, M.G. et al. Immune sensing of *Candida albicans* requires cooperative recognition of mannans and glucans by lectin and Toll-like receptors. *J Clin Invest* 116, 1642-50 (2006).

Ifrim, D.C. et al. *Candida albicans* primes TLR cytokine responses through a Dectin-1/Raf-1-mediated pathway. *J Immunol* 190, 4129-35 (2013).

Q7. *The exposed glucan levels are comparable between C. auris and C. albicans suggesting concealment of beta-glucan from Dectin-1 recognition is not the driving factor here, and the text should be modified to reflect this. The model presented in Fig 7 should also be updated. The legend states that upon contact with an immune competent host C. auris masks its glucan, but there is no data present to support this. The cell wall staining was all performed on lab grown cells, no other conditions were tested to demonstrate that the cell wall actually changes upon contact with the host. The glucan staining also does not look as it should, glucan is not normally exposed in SC5314 as shown. The beta-glucan antibody used is known to have problems, Fc-Dectin-1 is a much more reliable way to quantify glucan exposure.*

Answer: We would like thank the reviewer for these precious suggestions. First, we accepted the reviewer's advice by replacing the β -glucan antibody with Fc-Dectin-1, and indeed we obtained high quality images showing β -1,3-glucan staining (**Fig. 5f, Fig.6d, j, Supplementary Fig. 11a**). The fluorescent staining and flow cytometry analyses strongly indicated that when compared to what we observed in *C. albicans*, the proportion of the outer mannan layer was significantly increased in *C. auris* (**Fig. 5f, Supplementary Fig. 11a**). The results were further confirmed by the other two approaches, including 1) the methods for high-pressure freezing (HPF) and freeze substitution (FS) to examine the ultrastructure of *C. auris* BJCA001 or *C. albicans* SC5314 yeast cell wall by transmission electron microscopy (TEM); and 2) gas chromatography-mass spectrometry (GC-MS) and high-performance liquid chromatography (HPLC) analyses for sugar composition (**Fig. 5**). Based on our model, the outer mannan layer acts to mask the inner layer of β -glucan from exposure and detection by innate immune cells, and therefore plays a key role in protecting *C. auris* against the host innate immune clearance. The direct evidence supporting this model came from the observations that disruption of the manno-glycosylated cell wall proteins by deleting the putative

mannosyltransferase-encoding genes of *C. auris*, such as *PMR1*, *PMT1* and *OCH1*, leads to β -glucan unmasking and is sufficient to restore the innate immune activation by macrophages (**Fig. 6, Supplementary Figs 14-16**). (**Lines 395-414 on pages 21-22; Lines 480-556 on pages 25-28**)

Second, we examined the exposure levels of the three major cell wall polysaccharides (mannan, glucan and chitin) by flow cytometry. A similar exposure pattern was observed during host-pathogen interaction (**Supplementary Fig. 11b**). Upon contact with BMDMs, *C. auris* cell wall still displayed increase in relative proportion of mannan followed by a decrease in β -glucan levels when compared to that of *C. albicans*. (**Lines 422-425 on page 22**)

Q8. *Did you do basic analysis of the cell wall form all the isolates you tested, and where there differences between them?*

Answer: According to the reviewer's suggestions, we used fluorescent staining and flow cytometry techniques to profile the exposure levels of the three major cell wall polysaccharides (mannan, glucan and chitin) in eleven clinical *C. auris* strains available in our study (**Fig. 5f, Supplementary Fig. 11a**). When compared to what we observed in *C. albicans*, the proportion of the outer mannan layer was significantly increased in most *C. auris* isolates suggesting that this pattern appears to be conserved across most *C. auris* strains. (**Lines 417-422 on page 22**)

2. Minor comments

Q1. *Why apply a 2-fold cut off to the RNA seq data, why not just focus on adjusted P values?*

Answer: In RNA-seq data analysis, one important step is to identify differentially expressed genes (DEGs). In our work, we combined both the adjusted *P*-value and fold changes to selected DEGs. The resulting data table assigns *p*-adjusted ≤ 0.05 and fold change ≥ 2 for each gene. The reason of applying a 2-fold cutoff is that a less stringent cutoff allows for more noise or false positives in the downstream analysis. Actually, the same analysis has been used by other groups for picking DEGs in *C. albicans*.

References:

Witchley JN, Basso P, Brimacombe CA, Abon NV, Noble SM. Recording of DNA-binding events reveals the importance of a repurposed *Candida albicans* regulatory network for gut commensalism. *Cell Host Microbe*. 2021 Jun 9;29(6):1002-1013.e9.

Bruno M, Kersten S, Bain JM, Jaeger M, Rosati D, Kruppa MD, Lowman DW, Rice PJ, Graves B, Ma Z, Jiao YN, Chowdhary A, Renieris G, van de Veerdonk FL, Kullberg BJ, Giamarellos-Bourboulis EJ, Hoischen A, Gow NAR, Brown AJP, Meis JF, Williams DL, Netea MG. Transcriptional and functional insights into the host immune response against the emerging fungal pathogen *Candida auris*. *Nat Microbiol*. 2020 Dec;5(12):1516-1531.

Q2. *Line 282: you say that C. auris does not amount an innate immune response, but here you say C. auris is more tolerant to against the innate immune response, seems contradictory*

Answer: We apologize for the lack of clarity. The whole paragraph, which was shown below, has been rephrased to avoid misunderstanding.

“The notion that *C. auris* may evolve an efficient innate immune evasion strategy was further evaluated using the mouse model described above (Supplementary Fig.1). We intravenously infected groups of C57BL/6 mice

with PBS, *C. auris* (1×10^6 CFU/mice; L), *C. auris* (2×10^7 CFU/mice; H), or *C. albicans* (5×10^4 CFU/mice) and then assessed fungal burdens in various organs including kidney, liver, spleen, lung and brain, through CFU counting. It has to be noted that different inoculum doses of *C. albicans* (5×10^4 CFU/mice) and *C. auris* (1×10^6 CFU/mice) were used, largely because of the fact that the mice inoculated with *C. albicans* at 1×10^6 CFU experience a clinical illness manifested by ruffled fur, reduced activity and weight loss, and died rapidly within 5 days (Supplementary Fig. 5). In comparison, those receiving a lower inoculum (5×10^4 CFU/mice) could live long and healthy during the period of innate immune response (usually within the first 7 days). Consistent with the results in Supplementary Fig. 1, we observed in Fig. 3a and Supplementary Fig. 6a that following *C. auris* inoculation, the fungal burden was fairly constant and high on day 2 and 5 post-inoculations, regardless of the inoculum size. However, a relative reduction in fungal burden on day 2 and a comparably high level as that of *C. auris* on day 5 were observed after *C. albicans* infection, possibly due to the early activation of innate immune eradication.” **(Lines 250-267 on pages 14-15)**

Q3. *Line 432: are you sure your measurements are right here. Previous publications have indicated that the C. albicans mannan fibrils are between 50-100 nm, while you say they are only 1.5 nm? The included images are too small to be able to visualise the fibrils in detail. Your % of mannan seems comparable to other studies, so the difference reported in fibril length is unlikely to result from different growth conditions. It's more likely that these differences are due to the fixative method used. The gold standard for TEM to visualise the mannan fibrils is high pressure freeze substitution, as chemical fixation does not preserve the mannan layer. Therefore, not sure how reliable these measurements are.*

Answer: We accepted the reviewer's advices and took advantage of the methods for high-pressure freezing (HPF) and freeze substitution (FS) to examine the ultrastructure of *C. auris* BJCA001 or *C. albicans* SC5314 yeast cell wall by transmission electron microscopy (TEM). The same approach was also used for the *pmr1* Δ mutant of *C. auris*. Using these methods, we are able to clearly visualize the mannan fibrils (**Fig. 5c and 6a**). (**Lines 395-400 on page 21; Lines 504-505 on page 26**)

Q4. Does deletion of *PMR1* and *PMT1* in *C. auris* affect growth and morphology?

Answer: We performed experiments to examine the impact of *PMR1* or *PMT1* deletion on the growth and morphology of *C. auris*, according to the reviewer's suggestions.

1) *PMR1*. *In vitro* analysis showed that the *pmr1* Δ mutant of *C. auris* displayed significant vegetative growth defects, formed large aggregates and maintained the yeast form (**Supplementary Fig. 14a-e**). Furthermore, similar to its counterpart in *C. albicans*, the *PMR1*-deletion mutant of *C. auris* also exhibited a weakened cell wall, exemplified by increased sensitivity to cell wall- and cell membrane-damaging agents such as Calcofluor White (CFW), Congo Red and SDS (**Supplementary Fig. 14f**), and activation of the cell wall integrity pathway (**Supplementary Fig. 14g**). (**Lines 496-500 on page 26**)

2) *PMT1*. *In vitro* assays revealed that the *pmt1* Δ mutant of *C. auris* also showed significant vegetative growth defects, formed large aggregates and maintained the yeast form (**Supplementary Fig. 15b-d**). Similar to the *pmt1*^{-/-} mutant of *C. albicans*, the *pmt1* Δ mutant of *C. auris* did exhibit growth and morphology defects in cell wall- and cell membrane-damaging agents (**Supplementary Fig. 15e**), as well as activation of the cell wall integrity pathway (**Supplementary Fig. 15f**). (**Lines 530-534 on page 27**)

Q5. *Discussion seems a little long and unfocused*

Answer: We thank for the reviewer's suggestion and rewrite the discussion in revised manuscript.

Q6. *English in places could be better*

Answer: We thank the reviewer for the valuable comments. We have carefully revised the manuscript by proofreading the English language and avoiding potential language errors.

Reviewers' Comments:

Reviewer #1:

Remarks to the Author:

The authors performed an impressive number of experiments to explain some controversial results of this study from recent publications. The revised manuscript provides additional data that compared innate immune responses to live *C. auris* and live *C. albicans* in vitro and in vivo. Authors made two main conclusions that are same as before: (1) *C. auris* evades immune cells better than *C. albicans*. "compared to *C. albicans*, *C. auris* behaves as a silent invader". (2) "evasion of innate immunity is mainly mediated by a thick and structurally distinct mannan layer of cell wall, which masks b-glucan exposure and avoids innate immune clearance." The authors further suggest that the distinct mannan structure of *C. auris* is responsible for its efficient innate immune evasion in comparison to *C. albicans*, and "Current literatures showed somewhat controversial effects of the ability of *C. auris* infection to induce the innate immune response". I carefully evaluated data provided in this revised manuscript and recent papers that compared immune responses between *C. auris* and *C. albicans*, and found the conclusions made in this manuscript not well supported by their experimental results as detailed below.

Among all in vitro and in vivo data provided in the manuscript (Fig. 2 and Fig. 3), neutrophils showed the most difference between *C. auris* and *C. albicans*. The results are consistent with two publications: *Candida auris* evades neutrophil attack (mBio 2018 Johnson et al. PMID: PMC6106086). *C. auris* cell wall mannosylation contributes to neutrophils evasion through pathways divergent from *Candida albicans* and *Candida glabrata* (mSphere 2021, PMID: PMC8265655). The second paper showed that *C. auris* has the unique mannan structure and genetic disruption of mannosylation pathways (deleting PMR1 or VAN1) diminishes the outer cell wall mannan, unmask immunostimulatory components (beta-glucan), and promotes neutrophil engagement, phagocytosis, and killing, as in this manuscript. But the mSphere paper still reframed from concluding that a difference in mannan structure/or thickness can cause a difference in beta-glucan masking. The strong antifungal response to *C. albicans*, but not *C. auris*, could be due to the formation of *C. albicans* hyphae. Neutrophil responses to *C. albicans* hyphae is much stronger than to *C. albicans* yeast cells as determined by the activation of Syk kinase. Syk phosphorylation levels are similar between fixed *C. albicans* yeast and *C. auris*, but much higher in response to fixed hyphae, based on this publication (Spleen Tyrosine Kinase Is a Critical Regulator of Neutrophil Responses to *Candida* Species. PMID: PMC7218286). Since this study exclusively used live yeast of *C. albicans* SC5314, which rapidly develops into hyphae upon interaction with macrophages or neutrophils in their in vitro assays, most of differences in the immune responses between *C. albicans* and *C. auris* presented in Fig. 1 and Fig. 2 may come from *C. albicans* hyphae. In fact, Bruno et al. (2020 Nature Microbiology) used both live and thimerosal-killed *C. auris* and *C. albicans* cells when assaying phagocytic index by BMDM (Fig. 2). No significant differences in *C. auris* and *C. albicans* phagocytosis (percentage uptake) were observed for fixed *Candida*. Fixed cells should be used to visualize beta-glucan exposure on *Candida* yeast cells, with either anti-beta-glucan antibody or Fc-Dectin-1. This is to prevent changes in cell wall structure during the experiments. Heat inactivated cells should be used as controls for fully exposed beta-glucan. In the revision, localizations of beta-glucan or chitin on *C. albicans* yeast cells in Fig. 5 do not look correct. Also the levels are higher than expected based on published images for wild-type *C. albicans* yeast cells (same image is use for both Fig. 5 and supplemental figure). Levels of beta-glucan in *C. albicans* are almost similar to some of the *C. auris* strains, which then do not correlate with their immune responses in Fig. 4. Anti-beta-glucan antibodies and Fc-Dectin-1 should give similar results. The opposite results provided in the two versions of manuscripts may indicate other factors (growth stage, growth media) affected the results.

Reviewer #2:

Remarks to the Author:

As indicated previously, this paper describes an impressive and interesting dataset relating to the immunogenicity and virulence of the emerging fungal pathogen, *Candida auris*. The resubmission contains a significant body of additional data that address most of the concerns of the Reviewers. However, significant editing is required to remove potentially antagonistic phraseology and to sharpen the text and increase its accuracy. For example:

Line 8: Delete "with limited and somewhat controversial results". These words are not necessary and potentially antagonistic.

Line 79: Change to "... showed differing effects ..."

Lines 10, 564, 652: Remove "silent invader" and use more scientific language (e.g. "is less immunoinflammatory")

Lines 9-12: Rephrase because this infers that mannan does not mask beta-glucan in *C. albicans*. It does!

Line 13: This study does NOT examine how "*C. auris* evolves rapidly".

Line 20: Which fungal pathogen (singular) is being referred to here?

Line 61: Again, remove potentially antagonistic terminology. These aren't necessarily "discrepancies". They probably reflect interesting genetic differences between strains.

Line 90: Phenotypic switch does NOT "mimic" yeast-hypha morphogenesis. These are different phenomena.

Lines 98-99: This paper does NOT present data indicate that *C. auris* "develops an effective cell wall based-immune evasion strategy when it is confronted with host innate immune defense". This could be a constitutive phenotype.

Lines 266-268: This sentence does not appear to be compatible with the authors hypothesis that the immune attack is less potent against *C. auris*.

Line 307-309: *C. auris* did NOT fail to induce changes. Rather the responses were much weaker.

Line 356: Change "contrary" to "different".

Lines 365-369: Delete "We did find only a few tests. For example ..." & replace with "However, ..."

General: Remove anthropomorphic terms such as "ignored (line 234).

Line 398: HPLC analysis of the cell wall does NOT reveal "exposure" of cell wall components. Rather it measures TOTAL mannan, glucan, chitin.

Line 554: Change "switch off" to "significantly reduce"

Line 570: Change to "... our data differ from the recent in some respects"

Line 574-576: Change to "We sought to identify the possible bases for the differences between these two independent studies, which are summarised below..."

Line 628: Change "discrepancies in" to "differences between"

Line 634: There is no evidence to suggest that *C. auris* acts as an "immune silencer". Rather it is less immunoinflammatory.

Line 647: Again, "ignorance" is the wrong term. See previous point.

Reviewer #3:

Remarks to the Author:

The study has been much improved by the addition of the extra data, and text changes requested by the reviewers and I acknowledge the amount of effort the authors have put into improving the manuscript.

The overall findings from the manuscript is that, in this study, *C. auris* has an extensive outer mannan layer (very impressive TEM images) which conceals the underlying beta-glucan shielding the fungus from the actions on the innate immune system. The conclusions are backed up with extensive data, but overall the novelty is low as the role of mannan in shielding fungi from the immune system is already well established. This study is in contradiction to previous published reports, where *C. auris* has been shown to induce innate immune responses. In the discussion, the authors argue many reasons for these discrepancies (growth conditions, cell types, strains etc) but do not actually experimentally address the issue.

There is also some concern over the ethics of the in vivo work as mice actually died in these studies rather than being humanely terminated when infection had progressed to a point at which the mice started to suffer.

Major

Did you actually request some of the strains from the Bruno study to see if they gave you the

same results in your assay set up to confirm whether the differences are due to the strain or growth conditions?

The authors conclude that *C. auris* is not good at adhering to human skin based on the ability to bind cell lines in monoculture. Single cell culture is very different to actual skin. Therefore, you may not want to be so bold with your statements here, unless you measure adhesion to skin biopsies/3D models that actually mimic skin

How sick are the mice when infected with *C. auris*, if there is not a strong immune response do the mice succumb to the infection?

Was *C. auris* endocytosed by the epithelial cells?

I am surprised by you images the SC5314 cell wall when stained for mannan, and glucan (Fig S11). Usually there is much brighter staining of the *C. albicans* cell wall with ConA, and a lot less Fc-Dectin-1 staining. This is also true for the TEM images of SC5314 where the outer mannan layer is not uniform like in other published images. Therefore, some of the explanation for your results may purely be down to the growth conditions. Did you try growing the *Candida* in YPD or YNB to match other published studies?

Why did you put Och1 under the Dox repressor rather than making a gene deletion?

Have you looked at the numbers/expression of the mannosyltransferases to see if you can explain why *C. auris* has such an extensive outer mannan layer?

The TEM shows a much greater level of outer mannans, but the biochemical analysis shows mild increases in mannan levels (10% by HPLC), how do you explain this, a comment should be made in the text. In Fig S11 the FACS does not show ConA staining for SC5314. ConA stains SC5314 well and so I am not sure why the authors are not detecting it here. There is something odd going on.

Minor

Line 7: begun should be beginning

Line 62: replace clear with understood

Line 83: replace women with female

Line 103: the sentence is quite complicated to read

Line 113: measured daily by CFU determination

Line 303: remove this sentence from the main manuscript as this will question the ethics of your study. Mice should not die in these studies but should be terminated in a humane manner

Line 324: not sure apparently is the right word here

Line 360: remove whether

Fig1A what do you mean by fold on the Y axes of the graphs?

Response to the reviewers

Reviewer #1 (Remarks to the Author):

1. Major issues

Q1. *Among all in vitro and in vivo data provided in the manuscript (Fig. 2 and Fig. 3), neutrophils showed the most difference between C. auris and C. albicans. The results are consistent with two publications: Candida auris evades neutrophil attack (mBio 2018 Johnson et al. PMID: PMC6106086). C. auris cell wall mannosylation contributes to neutrophils evasion through pathways divergent from Candida albicans and Candida glabrata (mSphere 2021, PMID: PMC8265655). The second paper showed that C. auris has the unique mannan structure and genetic disruption of mannosylation pathways (deleting PMR1 or VAN1) diminishes the outer cell wall mannan, unmasks immunostimulatory components (beta-glucan), and promotes neutrophil engagement, phagocytosis, and killing, as in this manuscript. But the mSphere paper still reframed from concluding that a difference in mannan structure/or thickness can cause a difference in beta-glucan masking.*

Answer: We thank the reviewer for providing us such valuable information. Consistent with the major findings from the two published literatures that the reviewer mentioned above (PMCID: PMC6106086 and PMID: PMC8265655), we also found that compared to *C. albicans*, *C. auris* is less immunoinflammatory, as evidenced by lower rates in neutrophil and macrophage uptake, internalization and killing (**Fig. 2f-m; Supplementary Fig. 6a-c, g-i**). Further mechanistic studies revealed that the distinct mannan structure of *C. auris* cell wall appears to be crucial for innate immune evasion strategy of this fungus, as we reproducibly observed that genetic disruption of both O- and N-linked mannosylation pathways (mutant lacking each of the three major mannosyltransferases-encoding genes *PMR1*, *PMT1* and *OCH1*)

leads to β -glucan exposure followed by increased uptake, decreased viability and induction of proinflammatory cytokine/chemokine in BMDMs (**Fig.6; Supplementary Figs. 16-18**). Interestingly, a recent study by Horton *et al.* (PMCID: PMC8265655) identified that similar to our *in vitro* and *in vivo* observations, *C. auris* cell wall mannosylation, by shielding the exposure of β -glucan, contributes to the evasion of neutrophils *ex vivo* and in a zebrafish infection model. Surprisingly, unlike that of *C. auris*, disruption of the outer mannan layer in other *Candida* spp. (e.g., deleting *PMR1* or *VAN1* in *Candida albicans* and *Candida glabrata*) had no impact on neutrophil responses such as neutrophil engagement. These findings suggest that the mannosylation pathways are required for *C. auris* neutrophil evasion, however, it is not the case in *C. albicans* and *C. glabrata*. The divergence could be due to differences in neutrophil receptors that recognize cell wall components among *C. auris*, *C. albicans* and *C. glabrata*.

In comparison, the conclusions drawn by McKenzie *et al.* (PMCID: PMC2849426) were different when they investigated the role of mannan layer of *C. albicans* on macrophage-mediated immune evasion. They found that alterations in mannosylation of *C. albicans* cell wall (e.g., deleting *PMR1*, *MNT1-5* or *MNS2*) significantly affect the rate of recognition and phagocytosis by macrophages. For example, the absence of *O*- or *N*-linked mannans significantly increased the phagocytosis of *C. albicans*. This study supported that in *C. albicans*, the cell wall mannans mask recognition of underlying β -glucan in the cell wall and are required for phagocytosis and macrophage killing once taken up in macrophage phagosomes. Actually, we obtained the same results that similar to that of *C. albicans*, the outer mannan layer of *C. auris* cell wall acts to shield the exposure of β -glucan and contributes to macrophage-mediated innate immune evasion (**Fig. 6**).

Taken together, we proposed in the revised manuscript that other *Candida* species like *C. albicans* or *C. glabrata*, may behave similarly or differently as *C. auris* to induce innate immunity when the outer mannan layer was disrupted, arguing that the role of cell wall mannan in innate immune evasion strategies might be similar or divergent in different *Candida* species, depending on the type of immune cells interacting with the fungus. **(Lines 230-258 on pages 13-14; Lines 601-606 on page 31)**

References

1. Johnson *et al.* (2018) Emerging fungal pathogen *Candida auris* evades neutrophil attack. *mBio* 9(4): e01403-18 (PMC6106086)
2. Horton *et al.* (2021) *Candida auris* cell wall mannosylation contributes to neutrophil evasion through pathways divergent from *Candida albicans* and *Candida glabrata*. *mSphere* 6(3): e0040621 (PMC8265655)
3. McKenzie *et al.* (2010) Contribution of *Candida albicans* cell wall components to recognition by and escape from murine macrophages. *Infect and Immun* 78(4): 1650-8. (PMC2849426)

Q2. *The strong antifungal response to C. albicans, but not C. auris, could be due to the formation of C. albicans hyphae. Neutrophil responses to C. albicans hyphae is much stronger than to C. albicans yeast cells as determined by the activation of Syk kinase. Syk phosphorylation levels are similar between fixed C. albicans yeast and C. auris, but much higher in response to fixed hyphae, based on this publication (Spleen Tyrosine Kinase Is a Critical Regulator of Neutrophil Responses to Candida Species. PMID: PMC7218286). Since this study exclusively used live yeast of C. albicans SC5314, which rapidly develops into hyphae upon interaction with macrophages or neutrophils in their in vitro assays, most of differences in the*

immune responses between *C. albicans* and *C. auris* presented in Fig. 1 and Fig. 2 may come from *C. albicans* hyphae. In fact, Bruno et al. (2020 Nature Microbiology) used both live and thimerosal-killed *C. auris* and *C. albicans* cells when assaying phagocytic index by BMDM (Fig. 2). No significant differences in *C. auris* and *C. albicans* phagocytosis (percentage uptake) were observed for fixed *Candida*.

Answer: We thank the reviewer for raising this crucial point. Indeed, *C. albicans* switches between the yeast cells and hyphae during infection and morphological changes occurs upon interaction with the host. Studies have shown that the fungal morphotype (e.g., yeast and hyphae of *C. albicans*) may be important determinant of host response and different immune cells respond differently to yeast and hyphae. We carried out experiments to investigate the possible contribution of morphological change to different host responses observed in *C. albicans* and *C. auris*.

1) The innate immune system acts to sense the fungal pathogen mainly through Syk-coupled C-type lectin receptors (CLRs) and activate downstream signaling regulators by phosphorylation, including the classical mitogen-activated protein kinases (MAPKs). We evaluated levels of the active (phosphorylated) forms of ERK, JNK and p38, as well as the total protein levels, in BMDMs treated with live *C. auris* or *C. albicans*. It is noteworthy that treating BMDMs with *C. albicans* yeasts for only 15 mins, a time point when *C. albicans* cells are still uniformly yeast form (**Supplementary Fig. 4e**), is sufficient to induce peak phosphorylation of ERK, JNK and p38 MAPK in BMDMs without significantly affecting the total level of each protein, however, phosphorylation of these signaling factors were unaltered in *C. auris*-treated macrophages, indicating MAPK suppression (**Fig. 1g**). These results suggest that unlike *C. albicans*, *C. auris* may temper the activation of MAPK signaling pathway in controlling

the expression of proinflammatory cytokines and chemokines in macrophages. In other words, different from *C. auris*, *C. albicans* cells of unicellular budding yeast are able to induce strong MAPK activation in macrophages like BMDMs, suggesting that there already exist different innate responses between *C. albicans* and *C. auris* when they are yeast forms; **(Lines 198-202 on page 11)**

- 2) Similarly, we examined the abilities of different immune cells, including murine macrophages and neutrophils, to phagocytose live *C. albicans* SC5314 or *C. auris* BJCA001 after 15 min of coculture (Note: both strains are yeast forms at this time). We consistently observed that in both murine macrophages and neutrophils, *C. albicans* yeasts showed significantly higher rates of uptake compared to *C. auris* and the phagocytic index of *C. albicans* yeasts was also more than that of *C. auris*, further supporting that the yeast form of *C. albicans* is able to trigger stronger innate immune recognition than *C. auris*;

Fig.1: The yeast form of *C. albicans* is capable of triggering a stronger innate immune recognition than *C. auris*. **a** Representative snapshots were taken from live cell videos after 15 min co-incubation of murine BMDMs with live *C. albicans* SC5314 and *C. auris* BJCA001 cells (MOI=1). Numbers in the upper left corner of each image represent the time of phagocytic events, arrows indicate the fungal cells engulfed by BMDMs (Scale bar = 10 μ m). **b** and **c** Percentage uptake and phagocytic index for BMDMs ingesting *C. albicans* and *C. auris*. Macrophages that have taken up at least one fungal cell were manually tracked to allow a quantitative analysis of percentage uptake during 15 min co-incubation (**b**). The number of fungal cells ingested (phagocytic index) per 100 macrophages was manually counted during 15 min co-incubation (**c**). **d-f** Assays were carried out exactly the same as **b-c**, except that murine neutrophils were used.

3) When yeast-form cells of *C. albicans* or *C. auris* were inactivated by 4% paraformaldehyde (PFA-killed) and treated with BMDMs, we also found that *C. albicans* was able to stimulate higher levels of cytokine/chemokine

gene expression (**Supplementary Fig. 3**), minimizing a possible effect of hyphal induction during coculture. (**Lines 148-152 on page 9**).

- 4) Recent studies have indicated that *C. auris* cells, like the isolate (BJCA001) used in our study, were able to undergo morphological switches among three distinct cell types, including the typical yeast (Y), filamentation-competent (FC) yeast, and filamentous cells (F). The switch between the typical yeast and the FC/filamentous phenotype turns out to be heritable and triggered by passage through the mammalian body. We carried out *in vitro* and *in vivo* assays and compare the innate immune responses among the different cell types. *in vitro* assays using different mammalian cells indicated that all three cell types of *C. auris* displayed no differences in cell adherence and macrophage cytotoxicity (**Supplementary Fig. 9a, b**). Moreover, all three cell types of *C. auris* also failed to induce p38 MAPK activation and cytokine production (**Fig. 4a, Supplementary Fig. 9c, d**). Consistently, when the mice were challenged with either Y or FC form of *C. auris* in two different inoculum sizes, we observed that neither low (10^6 CFU/mice; L) nor high (2×10^7 CFU/mice; H) dose inoculum shows significant differences in fungal loads, innate immune cell populations and cytokine production (**Fig. 4b-d, Supplementary Fig. 9e-h**). Taken together, our data highly suggest that morphological changes have no impact on innate immune evasion of *C. auris*; (**Lines 327-350 on pages 18-19**)

- 5) In *C. albicans*, yeast and hyphae interact differently with the innate immune system and it is very interesting to test whether morphological differences between yeast and hyphae alone lead to differences in the interaction with host immune cells. Dectin-1 is the classical C-type lectin receptor that is widely expressed on phagocytes including macrophages and dendritic cells and contributes to the immunological response to β -glucans. Dectin-1

activation strongly promotes proinflammatory responses in macrophages. A previous study by Gantner *et al.* (PMCID: PMC556398) demonstrated that the *C. albicans* yeast cell wall β -glucan is largely shielded from Dectin-1 recognition by outer wall components like mannans, and importantly, the basic process of yeast budding growth and cell separation creates permanent scars which exposes sufficient β -glucan to trigger innate immune responses (e.g., phagocytosis and ROS production by macrophages) through specific binding of the innate immune recognition receptor Dectin-1. In contrast, the authors found that during hyphal growth, no β -glucan exposure occurs and *C. albicans* filaments fail to bind and activate Dectin-1. Collectively, the data from this study highly suggest that yeast cells of *C. albicans*, by binding to Dectin-1 receptor through β -glucan exposure, trigger antifungal inflammatory responses in macrophages, however, the hyphae fail to activate Dectin-1-mediated defenses;

- 6) A previous study by Moyes *et al.* (PMCID: PMC2991069) found that oral epithelial cells coordinate an innate immune response to *C. albicans* via NF- κ B and a biphasic MAPK response. Interestingly, activation of NF- κ B and the first MAPK phase, constituting c-Jun activation, was found to be morphology-independent and likely due to recognition of general fungal cell wall structures (mannan, β -glucan and chitin). Of course, the author found that both MKP1 and c-Fos activation were dependent upon hyphal formation. This study proposed epithelial mechanisms that discriminate between yeast and hyphal forms of *C. albicans* and further supported that the yeast-form cells of *C. albicans* are able to trigger strong innate immune responses. In comparison, *C. auris* failed to activate the pro-inflammatory responses including MAPK activation;
- 7) Another previous study by McKenzie *et al.* (PMCID: PMC2849426) analyzed the contribution of distinct *C. albicans* cell wall components and

yeast-hyphae morphogenesis to phagocytosis by and escape from macrophages. One of the most important findings from this study is that *C. albicans* phagocytosis by macrophages is dependent on the glycosylation status of the cell wall, but not morphogenic switching from yeast to hyphal forms. The lack of O- and N-linked mannans (*mns1* Δ/Δ single and *mnt1* Δ/Δ *mn2* Δ/Δ double mutant strains), more likely due to β -glucan unmasking, resulted in a significant increase in recognition and phagocytosis of *C. albicans* by macrophages. Interestingly, the authors found that *C. albicans* yeast-locked mutants (e.g., *clb2* Δ/Δ , *hgc1* Δ/Δ , *efg1* Δ/Δ and *cph1* Δ/Δ strains) were unaffected in the rate of uptake by macrophages compared to wild-type controls at 1 h and 3 h. The results clearly demonstrate that cell wall glycosylation, but not hyphal formation, is critically important for the recognition and phagocytosis of *C. albicans* by macrophages. We consistently observed that disruption of the mannan layer of *C. auris* (*pmr1* Δ , *pmt1* Δ , and *Tetoff-OCH1*) could sufficiently activate innate immune responses in *C. auris* mutants (**Fig. 6**);

Taken together, our data strongly suggest that the outer mannan layer acts to mask the inner layer of β -glucan from exposure and detection by innate immune cells, and therefore plays a key role in protecting *C. auris* against the host innate immune clearance. Comparatively, the basic process of yeast budding growth and cell separation in *C. albicans* creates permanent scars which exposes sufficient β -glucan and is sufficient to induce innate immune responses through specific binding of Dectin-1 receptor. Based on our and other groups' studies, it is very plausible that starting from the initial recognition of innate immune cells like macrophages and neutrophils, *C. albicans* yeast cells are able to trigger much stronger proinflammatory response than *C. auris*. Of course, upon the progression of host-fungal interaction, the filaments of *C. albicans* may enhance the immune activation, possibly through recognition of other host receptors.

References

1. Gantner *et al.* (2005) Dectin-1 mediated macrophage recognition of *Candida albicans* yeast but not filaments. **EMBO J** 24(6): 1277-86. (PMC556398)
2. Moyes *et al.* (2010) A biphasic innate immune MAPK response discriminates between the yeast and hyphal forms of *Candida albicans* in epithelial cells. **Cell Host Microbe** 8(3): 225-35 (PMC2991069)
3. McKenzie *et al.* (2010) Contribution of *Candida albicans* cell wall components to recognition by and escape from murine macrophages. **Infect and Immun** 78(4): 1650-8. (PMC2849426)

Q3. Fixed cells should be used to visualize beta-glucan exposure on *Candida* yeast cells, with either anti-beta-glucan antibody or Fc-Dectin-1. This is to prevent changes in cell wall structure during the experiments. Heat inactivated cells should be used as controls for fully exposed beta-glucan.

Answer: We appreciate the reviewer's suggestions. PFA-killed yeast cells of *C. albicans* or *C. auris* (treated with 4% paraformaldehyde) were stained with Fc-Dectin-1 and results revealed that compared to *C. auris*, *C. albicans* yeast cells display enhanced staining around the cell wall periphery (**Fig. 5f**, **Supplementary Fig. 13a**), suggestive of higher levels of β -glucan exposure. To prevent any changes in cell wall structure during the experiments, we also compared β -glucan exposure between PFA- and heat-killed yeast cells, using both Fc-Dectin-1 and β -glucan antibody. As shown in **Supplementary Fig. 13c**, same patterns of β -glucan exposure were shown in both staining methods. Moreover, we expectedly observed higher fluorescent intensity in PFA-killed *C. albicans* than in PFA-killed *C. auris*. Compared to PFA, heat treatment disrupted the cell wall structure and markedly increased β -glucan unmasking in

both species and interestingly, its level in *C. albicans* was still higher than *C. auris*. (Lines 438-445 on page 23)

Together, our data indicate that a dense outer layer of mannan in *C. auris* cell wall is able to mask β -glucan from immune recognition by Dectin-1.

Q4. *In the revision, localizations of beta-glucan or chitin on C. albicans yeast cells in Fig. 5 do not look correct. Also the levels are higher than expected based on published images for wild-type C. albican yeast cells (same image is use for both Fig. 5 and supplemental figure). Levels of beta-glucan in C. albicans are almost similar to some of the C. auris strains, which then do not correlate with their immune responses in Fig. 4. Anti-beta-glucan antibodies and Fc-Dectin-1 should give similar results. The opposite results provided in the two versions of manuscripts may indicate other factors (growth stage, growth media) affected the results.*

Answer: We very much appreciate the careful reading of our manuscript and valuable suggestions of the reviewer. We double checked our imaging procedures and found technical issues for those fluorescence images shown in our last version of manuscript (Fig.5). First, the images were taken using a regular fluorescence microscope (Olympus IX73) and the relatively low resolution may affect the effectiveness of pixel-based visualization, given the small size of *C. auris* cells. Second, during the process of chitin staining, we did not select a proper exposure time and the saturated fluorescence intensities reflect over-exposure images, especially those stained for chitin on *C. albicans* yeast cells. Third, we did not optimize the concentration of Fc-Dectin-1 before its application in our staining procedures. The clarity of β -glucan fluorescence on *C. albicans* cells was affected possibly due to incorrect use of Fc-Dectin-1 concentration.

We read a number of published literatures, carefully optimized our staining protocol and repeated the assays. The major changes include: 1) All images were taken using the high resolution confocal fluorescence microscope (Olympus FV-1200); 2) For chitin staining, we optimized the exposure time; 3) For β -glucan staining, we used both anti- β -glucan antibodies and Fc-Dectin-1 and applied to the two strains with the right concentrations after several rounds of trials. Note: the concentration of anti- β -glucan antibodies is 1 μ g/ml and the concentration of Fc-Dectin-1 was adjusted from 20 μ g/ml to 0.2 μ g/ml. The new staining results were shown in revised **Fig. 5f** and **Supplementary Fig. 13a**. We are confident that the concerns from the reviewer have been solved and the image qualities now support our conclusions.

We also compared β -glucan exposure between PFA- and heat-killed yeast cells, using both Fc-Dectin-1 and β -glucan antibody. As shown in **Supplementary Fig. 13c**, we observed that both anti- β -glucan antibodies and Fc-Dectin-1 staining yield similar result, as same patterns of β -glucan exposure were shown in both staining methods. Moreover, we expectedly observed higher fluorescent intensity in PFA-killed *C. albicans* than in PFA-killed *C. auris*. Compared to PFA, heat treatment disrupted the cell wall structure and markedly increased β -glucan unmasking in both species and interestingly, its level in *C. albicans* was still higher than *C. auris*. **(Lines 438-445 on page 23)**

We are sorry that the same image of *C. albicans* cell staining was mistakenly used in both **Fig. 5f** and **Supplementary Fig. 13a**. This obvious mistake has been corrected.

Reviewer #2 (Remarks to the Author):

Q. As indicated previously, this paper describes an impressive and interesting dataset relating to the immunogenicity and virulence of the emerging fungal pathogen, Candida auris. The resubmission contains a significant body of additional data that address most of the concerns of the Reviewers. However, significant editing is required to remove potentially antagonistic phraseology and to sharpen the text and increase its accuracy.

Answer: We are grateful to the reviewer for these constructive suggestions. We performed a comprehensive edit in the revised manuscript and tried to do our very best to remove all the ambiguities which might have occurred. In addition to the questions listed below, we also corrected a number of other potential errors in the manuscript (all modifications were highlighted in red) to increase accuracy and readability.

Q1: Line 8: Delete “with limited and somewhat controversial results”. These words are not necessary and potentially antagonistic.

Answer: The words have been deleted in the revised manuscript, according to the reviewer’s suggestion. **(Lines 7-8 on page 2)**

Q2: Line 79: Change to “... showed differing effects ...”

Answer: Corrected. **(Lines 76-77 on page 5)**

Q3: Lines 10, 564, 652: Remove “silent invader” and use more scientific language (e.g. “is less immunoinflammatory”)

Answer: We agree with the reviewer’s suggestion and have corrected the words in the revised manuscript. **(Lines 9, 596, 670 and 688 on pages 2, 30, 34 and 35)**

Q4: Lines 9-12: Rephrase because this infers that mannan does not mask beta-glucan in *C. albicans*. It does!

Answer: We apologize for this misunderstanding. The sentence has been rephrased according to the reviewer's comments. **(Lines 9-11 on page 2)**

Q5: Line 13: This study does NOT examine how "*C. auris* evolves rapidly".

Answer: We agree with the reviewer's comment and have deleted the words in the revised manuscript. **(Line 12 on page 2)**

Q6: Line 20: Which fungal pathogen (singular) is being referred to here?

Answer: We apologize for this obvious mistake. Pathogens (plural) should be used here. **(Line 19 on page 3)**

Q7: Line 61: Again, remove potentially antagonistic terminology. These aren't necessarily "discrepancies". They probably reflect interesting genetic differences between strains.

Answer: We agree with the reviewer's suggestion and have replaced "discrepancies" with "differences" in the revised manuscript. **(Lines 60, 611, 616 and 663 on pages 5, 31 and 34)**

Q8: Line 90: Phenotypic switch does NOT "mimic" yeast-hypha morphogenesis. These are different phenomena.

Answer: We agree with the reviewer's comments and rephrased the sentence in the revised manuscript, which was described as "a morphological change similar to the classical yeast-to-hyphae switch that was identified in most *Candida* species like *C. albicans*". **(Lines 87-89 on page 6)**

Q9: Lines 98-99: *This paper does NOT present data indicate that C. auris “develops an effective cell wall based-immune evasion strategy when it is confronted with host innate immune defense”. This could be a constitutive phenotype.*

Answer: We are grateful to the reviewer for pointing out this misunderstanding. We have rephrased the sentence in the revised manuscript, which was described as “like most fungi, this emerging fungal pathogen develops very effective immune-evasion systems through the distinct cell wall structure when it is confronted with host innate immune defense”. **(Lines 96-98 on page 6)**

Q10: Lines 266-268: *This sentence does not appear to be compatible with the authors hypothesis that the immune attack is less potent against C. auris.*

Answer: We agree with the reviewer’s comment and have corrected the sentence as the reviewer suggested, which was described as “These results suggest that *C. auris* may act differently from *C. albicans* and elicit less potent innate immune responses “ in the revised manuscript. **(Lines 282-283 on page 15)**

Q11: Line 307-309: *C. auris did NOT fail to induce changes. Rather the responses were much weaker.*

Answer: We agree with the reviewer’s comment and have corrected the sentence as the reviewer suggested. **(Lines 323-326 on page 17)**

Q12: Line 356: *Change “contrary” to “different”.*

Answer: Corrected. **(Line 374 on page 20)**

Q13: Lines 365-369: *Delete “We did find ... only a few tests. For example ...” & replace with “However, ... “*

Answer: We rephased the sentence in the revised manuscript according to the reviewer's suggestions. **(Lines 383-384 on page 20)**

Q14: *General: Remove anthropomorphic terms such as "ignored (line 234).*

Answer: We have removed any anthropomorphic terms in the revised manuscript, according to the reviewer's suggestions. For example, the words like "ignored, discrepancies". **(Lines 246 on page 14; Lines 60, 611, 616 and 663 on pages 5, 31 and 34)**

Q15: *Line 398: HPLC analysis of the cell wall does NOT reveal "exposure" of cell wall components. Rather is measures TOTAL mannan, glucan, chitin.*

Answer: The mistake has been corrected. **(Line 412 on page 22)**

Q16: *Line 554: Change "switch off" to "significantly reduce"*

Answer: Corrected as suggested. **(Lines 576-577 on pages 29-30)**

Q17: *Line 570: Change to "... our data differ from the recent in some respects"*

Answer: The sentence has been rephrased according to the reviewer's suggestions. **(Lines 607-608 on page 31)**

Q18: *Line 574-576: Change to "We sought to identify the possible bases for the differences between these two independent studies, which are summarised below..."*

Answer: The sentence has been rephrased according to the reviewer's suggestions. **(Lines 610-612 on page 31)**

Q19: *Line 628: Change "discrepancies in" to "differences between"*

Answer: Corrected as suggested. (Lines 663-664 on page 34)

Q20: Line 634: *There is no evidence to suggest that C. auris acts as an “immune silencer”. Rather it is less immunoinflammatory.*

Answer: We agree with the reviewer’s comment and have replaced the “immune silencer” with “less immunoinflammatory” in the revised manuscript. (Line 670 on page 34)

Q21: Line 647: *Again, “ignorance” is the wrong term. See previous point.*

Answer: Corrected as suggested. (Line 682 on page 35)

Reviewer #3 (Remarks to the Author):

1. Major Comments

Q1. *The overall findings from the manuscript is that, in this study, C. auris has an extensive outer mannan layer (very impressive TEM images) which conceals the underlying beta-glucan shielding the fungus from the actions on the innate immune system. The conclusions are backed up with extensive data, but overall the novelty is low as the role of mannan in shielding fungi from the immune system is already well established. This study is in contradiction to previous published reports, where C. auris has been shown to induce innate immune responses. In the discussion, the authors argue many reasons for these discrepancies (growth conditions, cell types, strains etc) but do not actually experimentally address the issue.*

Answer: We thank the reviewer for the valuable comments. In the discussion, we summarized possible explanations for the differences between our work and the one done by Bruno *et al.* (PMID: PMC32839538), and provided experimental evidence to support our claims. Of course, we were unable to perform parallel analyses for factors contributing to differences of the two studies because the strains used in Bruno's paper were unavailable. Here we list the experiments we have done to explain the major differences in these two studies.

1) Cell types. We mainly used murine BMDMs and neutrophils to compare the effects of *C. albicans* and *C. auris* on host innate immune activation whereas Bruno *et al.* mainly used human PBMCs for tackling similar questions. To further support the conclusion that *C. auris* is less immunoinflammatory than *C. albicans*, we carried out more experiments using human macrophages and neutrophils. These *in vitro* cell culture assays, together with *in vivo* animal models, provide strong evidence that

compared to *Candida albicans*, *C. auris* appears to be less immunoinflammatory and importantly, a thick and structurally distinct mannan layer of cell wall, which masks β -glucan exposure, contributes to the evasion of innate immunity **(Figs. 2 and 6; Supplementary Figs. 6 and 7; Supplementary videos 11-18); (Lines 8-11 on page 2)**

2) *Candida* strains. Due to the Covid-19 pandemic and also the very restrictive regulation policy on international shipping of microbial pathogens in our country, there is no way for us to obtain the same isolates used in Bruno's work. Thus, we couldn't perform a parallel analysis to directly compare the innate immune responses among the strains used in the two studies. In the revised manuscript, we added 5 more clinical isolates of *C. auris* and a total of 16 strains, who were isolated from different geographical areas and belonged to 4 different clades (Clade I, II, III and IV), were tested for the kinetics of infection-induced innate immune responses in macrophages. All *C. auris* strains failed to elicit meaningful proinflammatory response in BMDMs, as evidenced by unaltered p38 MAPK phosphorylation and decreased expression and secretion of cytokines/chemokines **(Fig. 4e-g and Supplementary Fig. 10b-e)**. In contrast, the recognition of *C. albicans* by BMDMs results in the release of inflammatory and chemotactic cytokines, aiding further immune cell recruitment and infection resolution; **(Lines 355-373 on pages 19-20)**

3) MOIs. Different MOIs were used in the two studies. However, we found that the failure of *C. auris* to elicit innate immune responses in the macrophages and neutrophils of human and mouse is highly conserved in all tested isolates and independent of MOIs **(Fig. 4g and Supplementary Fig. 10e)**. Moreover, we found that the same inocula (1×10^6 CFU/mice) did not change serum cytokine profiles, minimizing a possible impact of inoculum level **(Supplementary Fig. 8e); (Lines 315-317 on page 17; Lines 369-371 on page 20)**

- 4) Experimental time points. When we extended the incubation time to 24 h (the time point used by Bruno *et al.*), patterns of cytokine induction by *C. albicans* or *C. auris* remain largely unchanged, suggesting that the influence of incubation time could be minor (**Fig. 4g and Supplementary Fig. 10e**); (**Lines 369-371 on page 20**)
- 5) Growth conditions and experimental protocols. In Bruno's work, both *C. albicans* and *C. auris* strains were prepared by growing cells for 24 h in Sabouraud medium for 30 °C. In our experiments, the saturated overnight culture of indicated yeast strain was diluted and incubated to logarithmic growth phase at 30 °C. All strains were routinely grown in YPD medium. It is possibly that growth condition will affect the structure of PAMPs. Moreover, differences of the results could be due to the modified experimental protocols used in the two studies. Indeed, we did observe that growth condition affects the levels of three cell wall components (mannan, β -glucan and chitin) in *C. albicans* and *C. auris*. For example, *C. albicans* yeasts in the logarithmic growth phase appear to have more β -glucan and mannan but less chitin than cells in the stationary phase. Comparatively, *C. auris* cells in the logarithmic growth phase have slightly higher level of mannan, similar level of β -glucan and lower level of chitin than cells in the stationary phase.

Fig.2: The impact of cell growth condition on the levels of three major cell wall components. The mannan, β -glucan or chitin level of *C. auris* or *C. albicans* cell wall was quantified by flow cytometry. The logarithmic growth phase and stationary phase fungal cells were harvested from YPD medium and stained with ConA-FITC to detect mannan,

Fc-Dectin-1 to detect β -glucan, and calcofluor white (CFW) to detect chitin. Data are representative of three independent and reproducible experiments.

In summary, although we were unable to do tests one-by-one for all possibilities, we did carry out some major experiments to provide plausible explanations for the differences. Current evidence is still incomplete and more analyses are needed to draw firm conclusions.

Q2. *There is also some concern over the ethics of the in vivo work as mice actually died in these studies rather than being humanely terminated when infection had progressed to a point at which the mice started to suffer.*

Answer: We really appreciate the reviewer for raising this critical concern. We apologize for the lack of clarity and improper descriptions about the ethics of our *in vivo* work. We fully accepted the reviewer's comments and have corrected this in the revised manuscript. Mice were humanely sacrificed when infection had progressed to the humane endpoint at which the mice started to suffer. The humane endpoint in our mouse model of systemic candidiasis was defined as signs of severe illness: the mouse has reached a permissible percentage loss of body weight (about 20% to 25%) or look moribund. **(Lines 870-874 on page 44)**

Moreover, we comprehensively described the results and relative criteria for assessment of clinical scores during our animal studies **(Supplementary Fig. 7c; Supplementary Table 8)**, based on parameters that were described previously (PMCID: PMC4088949). **(Lines 862-870 on page 44)**

Reference

Conti *et al.* (2014) Animal models for candidiasis. *Curr Protoc Immunol* 105: 19.6.1-19.6.17. (PMC4088949)

Q3. *Did you actually request some of the strains from the Bruno study to see if they gave you the same results in your assay set up to confirm whether the differences are due to the strain or growth conditions?*

Answer: A very interesting question. As explained in Q1, we indeed planned a parallel analysis to directly compare innate immune responses among the strains used in both studies. Unfortunately, due to the Covid-19 pandemic, our country's customs have very restrictive regulation policy on international shipping of microbial pathogens. Current application procedures are very complicated and may take years. There is no way for us to obtain exactly the same strains used in Bruno's paper. As a backup, we obtained another 5 *C. auris* clinical isolates from our domestic collaborators, including clade III and IV. A total of 16 *C. auris* clinical isolates covering all four clades were used to examine the kinetics of infection-induced innate immune responses in macrophages. Our results demonstrate that all *C. auris* strains failed to elicit meaningful proinflammatory response in BMDMs, as evidenced by unaltered p38 MAPK phosphorylation and decreased expression and secretion of cytokines/chemokines (**Fig. 4e-g; Supplementary Fig. 10b-e**). In contrast, the recognition of *C. albicans* by BMDMs results in the release of inflammatory and chemotactic cytokines, aiding further immune cell recruitment and infection resolution. (**Lines 365-373 on pages 19-20**)

Q4. *The authors conclude that C. auris is not good at adhering to human skin based on the ability to bind cell lines in monoculture. Single cell culture is very different to actual skin. Therefore, you may not want to be so bold with your statements here, unless you measure adhesion to skin biopsies/3D models that actually mimic skin*

Answer: We are grateful to the reviewer for this constructive suggestion. We accept the reviewer's comments and measured fungal adhesion using both the commercial three-dimensional reconstructed epidermis-EpiSkin (PMCID:

PMC4797543) and the murine skin topical exposure model (PMCID: PMC7878403). As described in the revised manuscript, no statistically significant difference was observed in *C. auris* adhesion between skin-derived HaCat and gut-derived Caco-2 (**Supplementary Fig. 5a**) and an *in vitro* *Candida* infection model using the commercial three-dimensional reconstructed epidermis-EpiSkin also verified the weak adherence ability of *C. auris* (**Supplementary Fig. 5b, c**). However, using the murine skin topical exposure model described by Huang *et al.*, we obtained the same results that *C. auris* established persistent colonization on skin surface (**Supplementary Fig. 5d, e**). The exact reason for this difference (*in vitro* vs *in vivo*) is unclear at present, possibly related to the structural variations between monoculture and actual skin, as well as the differential growth response of *C. auris* under these two conditions. (**Lines 218-230 on pages 12-13**)

References

1. Liang *et al.* (2016) A *Trichophyton Rubrum* infection model based on the reconstructed human epidermis-Episkin. ***Chin Med J*** 129(1): 54-8. (PMC4797543)
2. Huang *et al.* (2020) Murine model of colonization with fungal pathogen *Candida auris* to explore skin tropism, host risk factors and therapeutic strategies. ***Cell Host Microbe*** 29(2): 210-221. (PMC7878403)

Q5. *How sick are the mice when infected with C. auris, if there is not a strong immune response do the mice succumb to the infection?*

Answer: We really appreciate the reviewer's valuable comments. As described in the revised manuscript, all mice infected with *C. auris* cells survived and appeared to be clinically normal, although we did observe abnormal symptoms, including weight loss, head bobbing and body spinning,

in some mice receiving a high inoculum (2×10^7 CFU/mice) (**Supplementary Fig. 7**). (Lines 271-275 on page 15)

Q6. Was *C. auris* endocytosed by the epithelial cells?

Answer: We thank the reviewer for such a valuable suggestion. Endocytosis of *C. auris* or *C. albicans* yeast cells by endothelial cells was determined by the differential fluorescence assay and live video microscopy described previously (PMCID: PMC1802757; PMC4601288). Our results suggest that similar to its behavior in innate immune cells, *C. auris* failed to be endocytosed by human Caco-2 cells (**Supplementary Fig. 5f, g; Supplementary Videos 1,2**) and some other epithelial or endothelial cells (**Supplementary Videos 3-10**). (Lines 227-230 on page 13)

References

1. Phan *et al.* (2007) Als3 is a *Candida albicans* invasion that binds to cadherins and induces endocytosis by host cells. *PLOS Biol* 5(3): e64. (PMC1802757)
2. Glass *et al.* (2015) Protection of *Candida parapsilosis* from neutrophil killing through internalization by human endothelial cells. *Virulence* 6(5): 504-14. (PMC4601288)

Q7. I am surprised by you images the SC5314 cell wall when stained for mannan, and glucan (Fig S11). Usually there is much brighter staining of the *C. albicans* cell wall with ConA, and a lot less Fc-Dectin-1 staining. This is also true for the TEM images of SC5314 where the outer mannan layer is not uniform like in other published images. Therefore, some of the explanation for your results may purely be down to the growth conditions. Did you try growing the *Candida* in YPD or YNB to match other published studies?

Answer: Actually, the reviewer #1 asked the same question. We very much appreciate the careful reading of our manuscript and valuable suggestions of both reviewers. We double checked our imaging procedures and found technical issues for those fluorescence images shown in our last version of manuscript (Fig.5). First, the images were taken using a regular fluorescence microscope (Olympus IX73) and the relatively low resolution may affect the effectiveness of pixel-based visualization, given the small size of *C. auris* cells. Second, during the process of chitin staining, we did not select a proper exposure time and the saturated fluorescence intensities reflect over-exposure images, especially those stained for chitin on *C. albicans* yeast cells. Third, we did not optimize the concentration of Fc-Dectin-1 before its application in our staining procedures. The clarity of β -glucan fluorescence on *C. albicans* cells was affected possibly due to incorrect use of Fc-Dectin-1 concentration.

We read a number of published literatures, carefully optimized our staining protocol and repeated the assays. The major changes include: 1) All images were taken using the high resolution confocal fluorescence microscope (Olympus FV-1200); 2) For chitin staining, we optimized the exposure time; 3) For β -glucan staining, we used both anti- β -glucan antibodies and Fc-Dectin-1 and applied to the two strains with the right concentrations after several rounds of trials. Note: the concentration of anti- β -glucan antibodies is 1 μ g/ml and the concentration of Fc-Dectin-1 was adjusted from 20 μ g/ml to 0.2 μ g/ml. The new staining results were shown in revised **Fig. 5f** and **Supplementary Fig. 13a**. We are confident that the concerns from the reviewer have been solved and the image qualities now support our conclusions.

The uniform mannan layer observed in the TEM image of *C. albicans* SC5314 might be accidentally caused by inappropriate manipulations during the process of TEM combined with freeze substitution. We repeated the assay and added new image in the revised manuscript (**Fig. 5c**).

In our assays, *Candida* cells were routinely grown in YPD medium, unless specifically indicated.

Q8. *Why did you put Och1 under the Dox repressor rather than making a gene deletion?*

Answer: As described in the revised manuscript, repeated attempts to delete the *C. auris* homologue of *OCH1* gene failed, indicating that this gene may be essential for the viability of this species. Then we took an alternative strategy by generating a Tetoff-*OCH1* mutant in which the endogenous *OCH1* promoter in *C. auris* BJCA001 was replaced with the tetracycline-regulatable promoter. **(Lines 567-569 on page 29)**

Interestingly, a recent study by Yadav *et al.* demonstrated that *OCH1* is a non-essential gene in some *Candida* species such as *C. albicans* and *C. dubliniensis* (mutant cells lacking *OCH1* are viable), however, may also be essential for the viability of *C. glabrata* (PMCID: PMC7750734).

Reference

Yadav *et al.* (2020) Differences in fungal immune recognition by monocytes and macrophages: N-mannan can be a shield or activator of immune recognition. **Cell Surf** 6: 100042. (PMC7750734)

Q9. *Have you looked at the numbers/expression of the mannosyltransferases to see if you can explain why C. auris has such an extensive outer mannan layer?*

Answer: As described in the revised manuscript, we searched the *Candida* genome database for the number of genes encoding mannosyltransferases and identified 51 in *C. albicans* and 49 in *C. auris*, respectively **(Supplementary Fig. 15b; Supplementary Table 5)**. Moreover, our

RT-qPCR analyses indicated that the transcript levels of genes encoding some of the major protein α - and *N*-mannosyltransferases were significantly increased in *C. auris* when compared to their levels in *C. albicans* (**Supplementary Fig. 15c**), which may explain the observed phenotype that *C. auris* has an extensive outer mannan layer. **(Lines 504-510 on page 26)**

Q10. *The TEM shows a much greater level of outer mannans, but the biochemical analysis shows mild increases in mannan levels (10% by HPLC), how do you explain this, a comment should be made in the text.*

Answer: An explanation was added to the revised manuscript. Interestingly, unlike the results from TEM, HPLC analyses of the outer mannan levels only showed a mild increase (10%) in *C. auris*, possibly due to its smaller cell size than *C. albicans*. **(Lines 427-429 on page 22)**

Q11. *In Fig S11 the FACS does not show ConA staining for SC5314. ConA stains SC5314 well and so I am not sure why the authors are not detecting it here. There is something odd going on.*

Answer: We apologize for our accidental mislabeling in Fig. S11 of previous version. The line representing the ConA staining (chitin) of SC5314 should be black color (not red color), which is exactly the same color in **Fig. 5f**. We have corrected this mislabeled line in the revised manuscript (**Supplementary Fig. 13a**).

Minor comments

Q1. *Line 7: begun should be beginning*

Answer: Corrected. **(Line 8 on page 2)**

Q2. *Line 62: replace clear with understood*

Answer: Corrected. (Line 60 on page 5)

Q3. Line 83: replace women with female

Answer: Corrected. (Line 81 on page 6)

Q4. Line 103: the sentence is quite complicated to read

Answer: We rephrased the sentence in the revised manuscript, which was changed as “Accumulating evidence suggests that the increasing global prevalence of *C. auris* infection could be influenced by host immune status”.

(Lines 102-103 on page 7)

Q5. Line 113: measured daily by CFU determination

Answer: We rephrased the sentence in the revised manuscript, according to the reviewer’s suggestion. (Lines 112-113 on page 7)

Q6. Line 303: remove this sentence from the main manuscript as this will question the ethics of your study. Mice should not die in these studies but should be terminated in a humane manner

Answer: We have deleted all improper descriptions about the ethics of *in vivo* study in the revised manuscript. (Lines 268-269 on page 15; Line 318 on page 17)

Q7. Line 324: not sure apparently is the right word here

Answer: We rephrased the sentence by removing the word “apparently” in the revised manuscript. (Line 339 on page 18)

Q8. Line 360: remove whether

Answer: Corrected. (Line 378 on page 20)

Q9. Fig1A what do you mean by fold on the Y axes of the graphs?

Answer: We thank the reviewer for raising this question. To avoid misunderstanding, we replaced “fold” with “mRNA (fold)” on the Y axes of the graphs, including all qPCR results in the revised manuscript (**Fig. 1a; Fig. 4e; Fig. 6h, n; Supplementary Fig. 3; Supplementary Fig. 9d; Supplementary Fig. 10c; Supplementary Fig. 14a, b; Supplementary Fig. 15c**). The results were normalized to the expression level of the control gene GAPDH and presented relative to the negative control (set as 1). Actually, similar term can be found in at least two published literatures (PMCID: PMC4439382; PMC6399272).

References

1. Deng *et al.* (2015) Tyrosine phosphatase SHP-2 mediates C-type lectin receptor-induced activation of the kinase Syk and anti-fungal TH17 responses. *Nat Immunol* 16(6): 642-52. (PMC4439382)
2. Wang *et al.* (2019) A small secreted protein triggers a TLR2/4-dependent inflammatory response during invasive *Candida albicans* infection. *Nat Commun* 10(1): 1015 (PMC6399272)

Reviewers' Comments:

Reviewer #1:

Remarks to the Author:

The revision did not address my concerns experimentally. Major conclusions are still not well supported. Overall, the conclusions are vague in this version, which reduced the significance as well. Since many of the *in vitro* and *in vivo* effects of *C. auris* on immune responses are already reported in a number of publications, it is not clear what are the major new findings in this study.

Major points:

The difference between *C. auris* and *C. albicans* *in vitro* and *in vivo* in this study could be all due to the use of SC5314, which is more immunogenic than other *C. albicans* strains. Including the *C. albicans* strain used in Bruno et al or a yeast-locked *C. albicans* mutant for experiments in Fig1, 2, 3 is necessary to determine if the differences in this study are coming from *C. albicans* strains or between *C. auris* and *C. albicans* yeast form.

In support of the above point, all immune responses produced by *C. auris*, either heat-inactivated or the *pmr1* and *pmt1* mutants in Fig. 5 and Fig. 6 are very low, compared to that of *C. albicans* SC5314 in Fig. 1. So, increasing β -glucan exposure of *C. auris* can't account the difference seen between *C. auris* and SC5314.

Minor points:

Cell images in this manuscript are hard to see. Dectin-1 binding pattern in SC5314 is still not consistent with published.

Reviewer #3:

Remarks to the Author:

The revised manuscript is much improved and it is clear that the authors have spent considerable time and effort in addressing the reviewer comments. From the data presented, and *C. auris* strains tested, it is clear that in the authors hands *C. auris* induces a less potent pro-inflammatory innate immune response, and that this could lead to the persistent colonisation/infection in patients. I have some minor comments:

The phagocytosis data for *C. auris* is far more variable than any of the other data, do the authors know why this might be?

In the discussion (line 695) the author's comment that the immune evasion promotes the spread of infection. However, none of the experiments directly experimentally address the spread of infection. Therefore, this sentence should be removed.

Line 696 the phrase "smart behaviour" should be replaced, and there is no evidence in the manuscript that this evasion strategy aids the spread of infections in hospitals. Surely this is more to do with the ability of the fungus to remain alive on surfaces and medical equipment than immune evasion strategies?

In line with the above comments, if *C. auris* effectively evades the immune system, why does this mean *C. auris* causes infections mainly in ICU patients which have suppressed immune systems (line 697)?

Line 966 how does this work link to the development of control measures to reduce risk of spread (linking above comments)?

Some of the figures have many panels which makes individual panels very small. It might be worth the authors removing some of the data to supplementary figures (i.e. only including the most relevant time point in main figures).

The manuscript still requires some minor edits as some of the sentences are rather complex and

on occasion missing the odd word here and there.

Response to the reviewers

Reviewer #1 (Remarks to the Author):

Q. *The revision did not address my concerns experimentally. Major conclusions are still not well supported. Overall, the conclusions are vague in this version, which reduced the significance as well. Since many of the in vitro and in vivo effects of C. auris on immune responses are already reported in a number of publications, it is not clear what are the major new findings in this study.*

Answer: In this study, we compare the innate immune responses induced by *C. auris* BJCA001 and *Candida albicans* SC5314 *in vitro* and *in vivo*. Our results indicate that *C. auris* BJCA001 appears to be less immunoinflammatory than *C. albicans* SC5314, and this differential response correlated with structural features of the cell wall. Several major findings in the study are summarized here below:

- 1) Previous studies showed differing results on the ability of *C. auris* to induce innate immune responses. We provide sufficient *in vitro* and *in vivo* evidence to support that *C. auris* isolates tend to induce a less potent pro-inflammatory innate immune response than *C. albicans*. Our work addressed an important question of whether innate immune response could be activated or suppressed upon *C. auris* infection;
- 2) Although considerable progress has increased our understanding of the biological and clinical aspects of *C. auris*, its interaction with host immune system is only now beginning to be investigated in-depth. So far, only a few studies have compared the innate immune responses following the challenge with *C. albicans* and *C. auris*. Our work adds new information and enhances our understanding of how *C. auris*, a pathogen that is rapidly gaining clinical importance, interacts with the host defense system during

its infection. And the observations, of course, will shed lights on future development of infection prevention and control measures;

- 3) We found that the outer cell wall layer of *C. auris* showing high density of mannan and low structural complexity (due to the lack of phosphomannan and acid-labile mannan), contributes to a reduced innate immune recognition, compared with *C. albicans*. A relatively simple structure of cell wall may generate less variability in antigen surface exposure and therefore more easily shield the fungal cells from being recognized by host innate immune system;
- 4) Decreased innate immune recognition and activation during *C. auris* infection are able to effectively protect the pathogen from immune clearance and benefit the colonization and invasion inside its host. Our work provided insights into a deeper understanding of the mechanisms associated with pathogenicity of this emerging human fungal pathogen;
- 5) Based on our observations and other groups' studies, it is very likely that stimulation with *C. auris* may yield different cytokine responses, depending on the type of immune cell being tested, and this behavior may reflect a flexible adaptation of this fungus to the changing microenvironments in the host. Moreover, our work further illustrated that different *Candida* species may behave similarly or differently as *C. auris* to induce innate immune responses when the outer mannan layer was disrupted, arguing that the role of cell wall mannan in innate immune responses might be similar or divergent in different *Candida* species, depending on the type of immune cells interacting with the fungus.

Major issues

Q1. *The difference between C. auris and C. albicans in vitro and in vivo in this study could be all due to the use of SC5314, which is more immunogenic than*

other *C. albicans* strains. Including the *C. albicans* strain used in Bruno et al or a yeast-locked *C. albicans* mutant for experiments in Fig1, 2, 3 is necessary to determine if the differences in this study are coming from *C. albicans* strains or between *C. auris* and *C. albicans* yeast form.

Answer: We thank the reviewer for providing us such valuable information. As explained in our previous response letter, currently there is no way to obtain the same isolates (*C. auris* and *C. albicans*) used in Bruno's work. The main reason is that our country has very restrictive regulation policy on international shipping of microbial pathogens, especially during the Covid-19 pandemic. Thus, we couldn't perform a parallel analysis to directly compare the innate immune responses among the strains used in the two studies. To rule out the possibility that the different innate immune responses between *C. auris* and *C. albicans* could be all due to the use of SC5314, which is more immunogenic than other *C. albicans* strains, we followed the reviewer's suggestion and performed comparative experiments using some other *C. albicans* strains available in the lab (see the Table below). It has to be mentioned that our city is currently experiencing a Covid-19 outbreak and a city-wide lockdown has a huge impact on research. We overcame the difficulty and evaluated the innate immune responses against challenge by several other strains of *C. albicans*. First, we used five strains of *C. albicans* (yCB215, yCB216, ATCC90028, BJ1097 and WO-1) and examined the mRNA levels of IL-1 β , IL-6, TNF- α , CXCL1 in BMDMs that were infected without (PBS; negative control) or with live *C. auris* or *C. albicans* yeast cells (MOI=5) for 3 h (SC5314 was used as a control). Second, we performed ELISA analysis to examine the production of TNF- α in the culture supernatants of BMDMs that were infected without or with these *C. albicans* cells (MOI=5) for 6 h. Third, we used more *C. albicans* strains (a total of 9 strains) available in the lab and carried out immunoblot analysis for p38 MAPK activation using lysates from BMDMs that were infected without or with live *C. auris* or *C. albicans* yeast cells (MOI=5) for 15 min. Notably, we included a

yeast-locked mutant strain of *C. albicans* (*efg1Δ/Δ*). The results were shown below. Similar to SC5314, each of these *C. albicans* strains also induced a more potent pro-inflammatory innate immune response than *C. auris* BJCA001, minimizing a possible effect of strain origin or morphology during coculture. Because our city is still experiencing Covid-19 lockdown and ordering/shipping a large number of mice for *in vivo* animal studies is extremely difficult, we can only perform *in vitro* cell culture assay now.

***C. albicans* strains used in the study**

Strain name	Description	Resource
SC5314	Standard strain	Fonzi and Irwin, 1993
CAI4	MTLa/α ura3::imm434/ura3::imm434 (derived from SC5314)	Huang et al. , 2006
BWP17	MTLa/α ura3::imm434/ura3::imm434 his1::hisG/his1::hisG arg4::hisG/arg4::hisG (derived from SC5314)	Huang et al. , 2009
ATCC90028	Standard strain from ATCC	Liao et al. , 2001
BJ1097	Clinical isolate	Tao et al. , 2014
WO-1	Standard strain (MTLa/α)	Huang et al. , 2006
yCB215	Clinical isolate	Lab collection
yCB216	Clinical isolate	Lab collection

1. Fonzi, W.A. & Irwin, M.Y. Isogenic strain construction and gene mapping in *Candida albicans*. *Genetics* 134, 717-28 (1993).
2. Huang, G. *et al.* Bistable expression of WOR1, a master regulator of white-opaque switching in *Candida albicans*. *Proc Natl Acad Sci U S A* 103, 12813-8 (2006).
3. Huang, G., Srikantha, T., Sahni, N., Yi, S. & Soll, D.R. CO(2) regulates white-to-opaque switching in *Candida albicans*. *Curr Biol* 19, 330-4 (2009).
4. Liao, R.S., Rennie, R.P. & Talbot, J.A. Novel fluorescent broth microdilution method for fluconazole susceptibility testing of *Candida albicans*. *J Clin Microbiol* 39, 2708-12 (2001).

5. Tao, L. *et al.* Discovery of a "white-gray-opaque" tristable phenotypic switching system in *Candida albicans*: roles of non-genetic diversity in host adaptation. *PLoS Biol* 12, e1001830 (2014).

Stimulation of BMDMs with different isolates of *C. albicans* showed conserved features of host innate immune response. **a-b** The innate immune response against various *C. albicans* strains. **(a)** Expression of IL-1 β , IL-6, TNF- α and CXCL1 were analyzed by real-time RT-qPCR. BMDMs were stimulated without (PBS control) or with each of *C. albicans* strains or *C. auris* BJCA001 at MOI=5 for 3 h. Results were normalized to the expression of the control gene GAPDH and are presented relative to those of negative control, set as 1. (n=3; from left to right, IL-1 β : $p=0.0013, 0.0177, 0.0445, 0.0008$; IL-6: $p=0.0073, 0.0337, 0.0058, 0.0002$; TNF- α : $p=0.0045$). **(b)** Production of TNF α was analyzed by ELISA after BMDMs were stimulated with indicated strains at MOI=5 for 6 h. **c.** Immunoblot analysis for detection of p38 MAPKs activation, using lysates from BMDMs that were stimulated with each of *C. albicans* strains at MOI=5 for 15 min. Data were expressed as mean \pm SD and were representative of three independent experiments. ns, no significance; * $p < 0.05$; ** $p < 0.01$; *** $p < 0.001$; **** $p < 0.0001$; by two-way ANOVA with Tukey's test (**a-b**).

On the other hand, we used a total of 16 clinical isolates of *C. auris*, who were isolated from different geographical areas and belonged to 4 different clades (Clade I, II, III and IV), and tested for the kinetics of infection-induced innate immune responses in macrophages. All *C. auris* isolates induced a less potent

pro-inflammatory innate immune response than *C. albicans* SC5314. (**Fig. 4e-g and Supplementary Fig. 10b-e**)

Moreover, we provided a number of experimental evidence (**Fig. 1g; Supplementary Fig. 3; Supplementary Fig. 4e; Fig. 4a-d; Supplementary Fig. 9; Fig. 6**), together with a list of published literatures (PMC556398; PMC2991069; PMC2849426), in our previous responses to the reviewer, to support our claim that starting from the initial recognition of innate immune cells like macrophages and neutrophils, *C. albicans* yeast-form cells are able to trigger much stronger proinflammatory response than *C. auris*. Of course, upon the progression of host-fungal interaction, the filaments of *C. albicans* may enhance the immune activation, possibly through recognition of other host receptors. For example, McKenzie *et al.* found that *C. albicans* yeast-locked mutants (*e.g.*, *clb2* Δ/Δ , *hgc1* Δ/Δ , *efg1* Δ/Δ and *cph1* Δ/Δ strains) were unaffected in the rate of uptake by macrophages compared to wild-type controls, suggesting that cell wall glycosylation, but not hyphal formation, is critically important for the recognition and phagocytosis of *C. albicans* by macrophages (PMC2849426).

Taken together, our study indicates that *C. auris* is less immunoinflammatory than *C. albicans* SC5314, and persistently resides in the body of infected animals. We also argued that stimulation with *C. auris* may yield different cytokine responses, depending on the type of immune cell being tested.

References

1. Gantner *et al.* (2005) Dectin-1 mediated macrophage recognition of *Candida albicans* yeast but not filaments. **EMBO J** 24(6): 1277-86. (PMC556398)

2. Moyes *et al.* (2010) A biphasic innate immune MAPK response discriminates between the yeast and hyphal forms of *Candida albicans* in epithelial cells. ***Cell Host Microbe*** 8(3): 225-35 (PMC2991069)
3. McKenzie *et al.* (2010) Contribution of *Candida albicans* cell wall components to recognition by and escape from murine macrophages. ***Infect and Immun*** 78(4): 1650-8. (PMC2849426)

Q2. *In support of the above point, all immune responses produced by C. auris, either heat-inactivated or the pmr1 and pmt1 mutants in Fig. 5 and Fig. 6 are very low, compared to that of C. albicans SC5314 in Fig. 1. So, increasing β -glucan exposure of C. auris can't account the difference seen between C. auris and SC5314.*

Answer: We thank the reviewer for raising this crucial point. Indeed, we observed that increasing β -glucan exposure of *C. auris* did induce the innate activation, however, with a level that is less than *C. albicans*. We reasoned that heat-inactivation or disruption of mannosyltransferase-encoding genes (e.g., *PMR1*, *PMT1*, *OCH1*) possibly affects other, as-yet-undiscovered, factors and mechanisms involved in recognition of *C. auris* as invading pathogens by cells of the innate immune system, and somehow interfered with the effect of β -glucan unmasking. More decent studies are needed to test this possibility.

Minor issues

Q. *Cell images in this manuscript are hard to see. Dectin-1 binding pattern in SC5314 is still not consistent with published.*

Answer: We thank the reviewer for the careful reading of our manuscript. All cell images were taken using the high resolution confocal fluorescence microscope (Olympus FV-1200) located at our core facility. This microscope is well suited for taking fluorescence images, but recently we found that it does

not perform very well on bright field images (our core facility had requested to repair or replace the microscope lens, however, the engineer could not make it due to the city-wide Covid-19 lockdown during the past two months in our city). To make better visualization, we used ImageJ software to adjust the brightness and contrast of the images (**Fig. 5f; Fig. 6d,j; Supplementary Fig. 13a**).

C. auris is able to diminish its detection by innate immune cells such as macrophages through masking of β -glucan in the inner cell wall with an outer layer of heavily glycosylated mannoproteins. It is very difficult to clearly visualize the β -glucan exposure of *C. auris* yeast cells under microscope. To improve visualization of the image, we adjusted the fluorescence turn-on ratio in order to obtain a significant increase of fluorescence signal presence/intensity. Since we used the same microscope setting to check Dectin-1 binding patterns in both *C. auris* and *C. albicans* yeast cells, it is likely that the images showing the distribution pattern of β -glucan in *C. albicans* may slightly differ from those seen in published literatures. Of course, our experimental setting definitely did not change our conclusions.

Reviewer #3 (Remarks to the Author):

Q1: *The phagocytosis data for C. auris is far more variable than any of the other data, do the authors know why this might be?*

Answer: The variation could be due to different total numbers of cells counted in microscopic fields. In our assays, we referred to a statistic method described by Godoy *et al* (PMC9016338). For each assay, all immune cells (macrophages or neutrophils), including those with and without fungal cells internalized, were counted by combining cells from 6 microscopic fields (taken from 6 videos). In each field, fungal cells engulfed by immune cells were counted at indicated time points, and the phagocytic index (the total number of fungal cells taken up per field) was calculated with a formula [(number of fungal cells at time point/number of immune cells) *100].

For example, when neutrophils were treated with either *C. auris* or *C. albicans* yeast cells for 30 min, the phagocytosis index was calculated based on procedures described below.

- 1) First, we chose six different microscopic fields from the movies and count the total number of neutrophils, as well as fungal cells engulfed by neutrophils. The phagocytosis index of each field was calculated using the formula [Phagocytosis index = (number of fungal cells at time point/number of immune cells) *100];

C. auris BJCA001

Field 1		Field 2		Field 3		Field 4		Field 5		Field 6	
Neutrop hills	# of engulfed fungal cells	Neutrop hills	# of engulfed fungal cells	Neutrop hills	# of engulfed fungal cells	Neutrop hills	# of engulfed fungal cells	Neutrop hills	# of engulfed fungal cells	Neutrop hills	# of engulfed fungal cells
Cell 1	0	Cell 1	0	Cell 1	0	Cell 1	0	Cell 1	0	Cell 1	0
Cell 2	1	Cell 2	0	Cell 2	0	Cell 2	0	Cell 2	0	Cell 2	0
Cell 3	0	Cell 3	0	Cell 3	0	Cell 3	0	Cell 3	0	Cell 3	0
Cell 4	0	Cell 4	0	Cell 4	0	Cell 4	0	Cell 4	0	Cell 4	0
Cell 5	0	Cell 5	0	Cell 5	0	Cell 5	0	Cell 5	0	Cell 5	0
Cell 6	0	Cell 6	0	Cell 6	0	Cell 6	0	Cell 6	0	Cell 6	0
Cell 7	0	Cell 7	0	Cell 7	0	Cell 7	0	Cell 7	0	Cell 7	0
Cell 8	0	Cell 8	0	Cell 8	0	Cell 8	0	Cell 8	0	Cell 8	0
Cell 9	0	Cell 9	0	Cell 9	0	Cell 9	0	Cell 9	0	Cell 9	0
Cell 10	0	Cell 10	0	Cell 10	0	Cell 10	0	Cell 10	0	Cell 10	0
Cell 11	0	Cell 11	0	Cell 11	0	Cell 11	0	Cell 11	0	Cell 11	0
Cell 12	0	Cell 12	0	Cell 12	0	Cell 12	0	Cell 12	0	Cell 12	0
Cell 13	0	Cell 13	0	Cell 13	0	Cell 13	0	Cell 13	0	Cell 13	0
Cell 14	0	Cell 14	0	Cell 14	0	Cell 14	0	Cell 14	0	Cell 14	0
Cell 15	0	Cell 15	0	Cell 15	0	Cell 15	0	Cell 15	0	Cell 15	0
Cell 16	0	Cell 16	0	Cell 16	0	Cell 16	0	Cell 16	0	Cell 16	0
Cell 17	0	Cell 17	0	Cell 17	0	Cell 17	0	Cell 17	0	Cell 17	0
Total # of fungus	1	Total # of fungus	0	Total # of fungus	0	Total # of fungus	0	Total # of fungus	0	Total # of fungus	0
Phagocytosis index	5.882353	Phagocytosis index	0	Phagocytosis index	0	Phagocytosis index	0	Phagocytosis index	0	Phagocytosis index	0

C. albicans SC5314

Field 1		Field 2		Field 3		Field 4		Field 5		Field 6	
Neutrop hills	# of engulfed fungal cells	Neutrop hills	# of engulfed fungal cells	Neutrop hills	# of engulfed fungal cells	Neutrop hills	# of engulfed fungal cells	Neutrop hills	# of engulfed fungal cells	Neutrop hills	# of engulfed fungal cells
Cell 1	0	Cell 1	0	Cell 1	1	Cell 1	0	Cell 1	0	Cell 1	1
Cell 2	1	Cell 2	1	Cell 2	1	Cell 2	0	Cell 2	0	Cell 2	1
Cell 3	0	Cell 3	1	Cell 3	2	Cell 3	0	Cell 3	0	Cell 3	2
Cell 4	0	Cell 4	1	Cell 4	0	Cell 4	0	Cell 4	0	Cell 4	0
Cell 5	0	Cell 5	1	Cell 5	0	Cell 5	0	Cell 5	0	Cell 5	0
Cell 6	2	Cell 6	2	Cell 6	1	Cell 6	0	Cell 6	0	Cell 6	0
Cell 7	0	Cell 7	1	Cell 7	1	Cell 7	0	Cell 7	0	Cell 7	1
Cell 8	0	Cell 8	1	Cell 8	1	Cell 8	0	Cell 8	0	Cell 8	1
Cell 9	4	Cell 9	1	Cell 9	0	Cell 9	1	Cell 9	2	Cell 9	0
Cell 10	1	Cell 10	1	Cell 10	0	Cell 10	0	Cell 10	0	Cell 10	0
Cell 11	0	Cell 11	0	Cell 11	2	Cell 11	0	Cell 11	0	Cell 11	0
Cell 12	1	Cell 12	1	Cell 12	0	Cell 12	0	Cell 12	0	Cell 12	0
Cell 13	0	Cell 13	0	Cell 13	0	Cell 13	0	Cell 13	0	Cell 13	0
Total # of fungus	9	Cell 14	1	Total # of fungus	9	Total # of fungus	1	Total # of fungus	4	Total # of fungus	6
Phagocytosis index	69.23077	Cell 15	0	Phagocytosis index	69.23077	Phagocytosis index	7.692308	Phagocytosis index	30.76923	Phagocytosis index	46.15385
		Cell 16	0								
		Cell 17	1								
		Total # of fungus	13								
		Phagocytosis index	76.47059								

2) Second, we generated a table to summarize the data above.

Phagocytosis index

Field #	C. auris BJCA001	C. albicans SC5314
1	5.882535	69.23077
2	0	76.47059
3	0	69.23077
4	0	7.692308
5	0	30.76923
6	0	46.15385

3) Finally, the data were plotted using GraphPAD software.

We found that the number of engulfed fungal cells from different fields sometimes may differ significantly from one another, and this may yield variations.

Reference

Godoy, P., Darlington, P.J. & Whiteway, M. (2022) Genetic Screening of *Candida albicans* Inactivation Mutants Identifies New Genes Involved in Macrophage-Fungal Cell Interactions. *Front Microbiol* 13, 833655. (PMC9016338)

Q2: *In the discussion (line 695) the author's comment that the immune evasion promotes the spread of infection. However, none of the experiments directly experimentally address the spread of infection. Therefore, this sentence should be removed.*

Answer: We agree with the reviewer's comment and have deleted the sentence in the revised manuscript. **(Lines 694-695 on page 35)**

Q3: *Line 696 the phrase "smart behaviour" should be replaced, and there is no evidence in the manuscript that this evasion strategy aids the spread of infections in hospitals. Surely this is more to do with the ability of the fungus to remain alive on surfaces and medical equipment than immune evasion strategies?*

Answer: We agree with the reviewer's suggestion and have corrected the description in the revised manuscript. **(Lines 694-695 on page 35)**

Q4: *In line with the above comments, if C. auris effectively evades the immune system, why does this mean C. auris causes infections mainly in ICU patients which have suppressed immune systems (line 697)?*

Answer: We thank the reviewer for valuable suggestions, the sentence containing misleading information has been deleted in the revised manuscript. **(Lines 694-695 on page 35)**

Q5: *Line 966 how does this work link to the development of control measures to reduce risk of spread (linking above comments)?*

Answer: We agree with the reviewer's comment and have deleted the sentence in the revised manuscript, in order to avoid misleading. **(Lines 694-695 on page 35)**

Q6: *Some of the figures have many panels which makes individual panels very small. It might be worth the authors removing some of the data to supplementary figures (i.e. only including the most relevant time point in main figures).*

Answer: In the revised manuscript, all items, including 7 major figures, 19 supplementary figures and a number of tables and movies, were arranged to fit together into a coherent whole.

Q7: *The manuscript still requires some minor edits as some of the sentences are rather complex and on occasion missing the odd word here and there.*

Answer: We are grateful to the reviewer for these constructive suggestions. We made key edits in the revised manuscript and tried to do our very best to remove all the ambiguities which might have occurred.